# CRADLE: Empowering Foundation Agents Towards General Computer Control

**Weihao Tan**[1][*] **Wentao Zhang**[1][*] **Xinrun Xu**[2][*] **Haochong Xia**[1][†] **Ziluo Ding**[3][†] **Boyu Li**[3][†] **Bohan Zhou**[4][†]
**Junpeng Yue**[4][†] **Jiechuan Jiang**[4][†] **Yewen Li**[1][†] **Ruyi An**[1][†] **Molei Qin**[1][†] **Chuqiao Zong**[1][†] **Longtao Zheng**[1][†]
**Yujie Wu**[5][†] **Xiaoqiang Chai**[5][†] **Yifei Bi**[3] **Tianbao Xie**[6] **Pengjie Gu**[1] **Xiyun Li**[3] **Ceyao Zhang**[7] **Long Tian**[5]
**Chaojie Wang**[5] **Xinrun Wang**[1][‡] **Börje F. Karlsson**[3][‡][†] **Bo An**[1][5] **Shuicheng Yan**[8][5] **Zongqing Lu**[4][3]

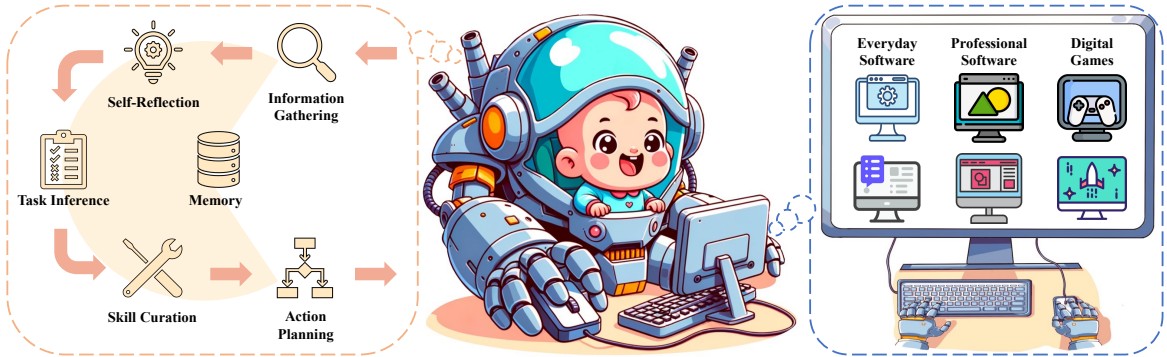

Figure 1: The **CRADLE** framework empowers nascent foundation models to perform complex computer tasks via the same unified interface humans use, *i.e.,* screenshots as input and keyboard & mouse operations as output.

## Abstract

Despite the success in specific scenarios, existing foundation agents still struggle to generalize across various virtual scenarios, mainly due to the dramatically different encapsulations of environments with manually designed observation and action spaces. To handle this issue, we propose the **General Computer Control** (GCC) setting to restrict foundation agents to interact with software through the most unified and standardized interface, *i.e.,* using screenshots as input and keyboard and mouse actions as output. Then we introduce **CRADLE**, a modular and flexible LMM-powered framework, as a preliminary attempt towards GCC. Enhanced by six key modules: Information Gathering, Self-Reflection, Task Inference, Skill Curation, Action Planning, and Memory, **CRADLE** is able to understand input screenshots and output executable code for low-level keyboard and mouse control after high-level planning, so that **CRADLE** can interact with any software and complete long-horizon complex tasks without relying on any built-in APIs. Experimental results show that **CRADLE** exhibits remarkable generalizability and impressive performance across four previously unexplored commercial video games (Red Dead Redemption 2, Cities: Skylines, Stardew Valley and Dealer's Life 2), five software applications (Chrome, Outlook, Feishu, Meitu and CapCut), and a comprehensive benchmark, OSWorld. With a unified interface to interact with any software, **CRADLE** greatly extends the reach of foundation agents thus paving the way for generalist agents. Video demos and code can be found at https://baai-agents.github.io/Cradle.

[*]Equal contribution [1]Nanyang Technological University, Singapore [2]Institute of Software, Chinese Academy of Sciences [3]Beijing Academy of Artificial Intelligence [4]Peking University [5]Skywork AI [6]The University of Hong Kong [7]The Chinese University of Hong Kong, Shenzhen [8]National University of Singapore. Correspondence to: Weihao Tan <weihao001@ntu.edu.sg>, Bo An <boan@ntu.edu.sg>, Shuicheng Yan <yansc@nus.edu.sg>, Zongqing Lu <zongqing.lu@pku.edu.cn>.

*Proceedings of the $42^{nd}$ International Conference on Machine Learning*, Vancouver, Canada. PMLR 267, 2025. Copyright 2025 by the author(s).

## 1. Introduction

Artificial General Intelligence (AGI) has long been a north-star goal for the AI community (Morris et al., 2023). The recent success of foundation agents, *i.e.,* agents empowered by large multimodal models (LMMs) and advanced tools, in various environments, *e.g.,* web browsing (Zhou

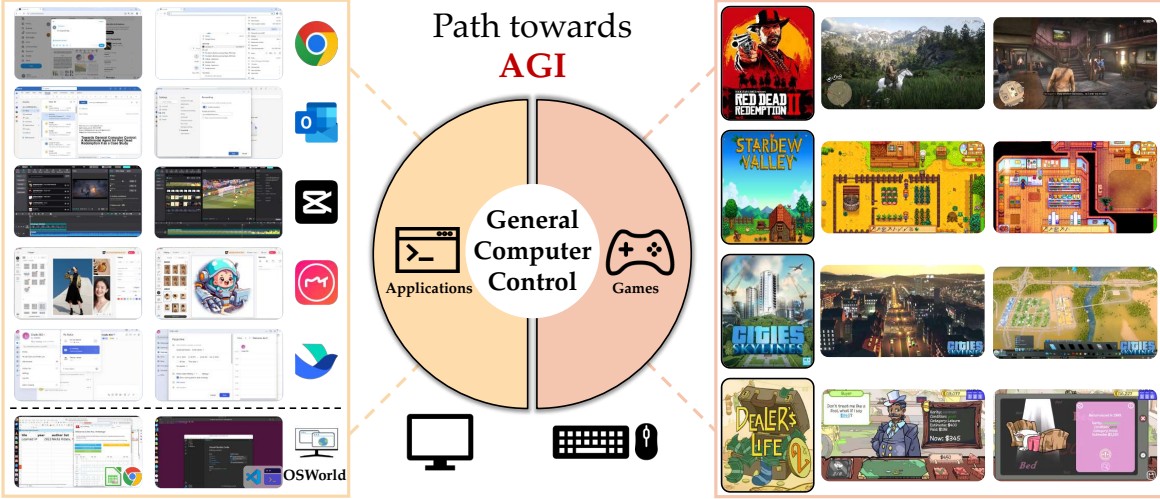

Figure 2: Taxonomy of GCC and the games and software investigated in this work.

et al., 2023; Deng et al., 2023; Gur et al., 2023; Zheng et al., 2024b;a; He et al., 2024), operating mobile applications (Yang et al., 2023b; Wang et al., 2024b) and desktop software (Zhang et al., 2024; Wu et al., 2024), crafting and exploration in Minecraft (Wang et al., 2023b; 2024a; 2023a), and some robotics scenarios (Huang et al., 2022; Brohan et al., 2023b; Driess et al., 2023; Brohan et al., 2023a), have shown promise. However, current foundation agents still struggle to generalize across different scenarios, primarily due to the dramatic differences in the encapsulation of environments with human-designed observation and action space. Therefore, developing foundation agents applicable to various environments remains extremely challenging.

Computers, as the most important and universal interface that connects humans and the increasing digital world, provide countless rich software, including applications and realistic video games for agents to interact with, while avoiding the challenges of robots in reality, such as hardware requirements, constraints of practicability, and possible catastrophic failures (Raad et al., 2024). Mastering these virtual environments is a promising path for foundation agents to achieve generalizability. Therefore, we propose the **General Computer Control** (GCC) setting [1]:

> *Building **foundation agents** that can master **ANY** computer task via the universal human-style interface by receiving input from screens and audio and outputting keyboard and mouse actions.*

There are many challenges to achieving GCC: i) good alignment across multi-modalities for better understanding and decision-making; ii) precise control of keyboard and mouse to interact with the computer, which has a large hybrid action space, including not only which key to press and where the mouse to move, but also the duration of the press and

the speed of the mouse movement; iii) long-horizontal reasoning due to the partial observability of complex GCC tasks, which also leads to the demand for long-term memory to maintain past useful experiences; and iv) efficient exploration to discover better strategies and solutions autonomously, *i.e.,* self-improving, which can allow agents to generalize across the various tasks in the digital world.

As shown in Figure 1, we introduce **CRADLE**, a novel modular LMM-powered framework towards GCC. **CRADLE** consists of six key modules: 1) information gathering, to extract the relevant information from multimodal observations; 2) self-reflection, to rethink past experiences about whether the actions and tasks are successfully completed and reasons for possible failures; 3) task inference, to determine whether to continue current tasks or propose a new task given the current situation; 4) skill curation, to generate, update, and retrieve useful skills for the current task; 5) action planning, to generate specific executable operations for keyboard and mouse control via skills; and 6) memory, for storage, summary, and retrieval of past experiences.

As illustrated in Figure 2, tasks in GCC can be broadly divided into two categories: video game playing and software application manipulation. Compared to Software applications, video games provide more comprehensive and challenging testbeds to evaluate and improve agents' various abilities including but not limited to classical UI manipulation and embodied control. In this work, we conduct extensive experiments to demonstrate the generalizability of **CRADLE** in such complex environments, while also mastering diverse everyday software applications in distinct domains. We managed to prove that commercial software is out-of-box testbeds under our framework. The four selected representative games are: epic AAA 3D role-playing game, **RDR2**, 2D pixel-art farming simulation game, **Stardew Valley**, pawn shop simulation game, Dealer's Life 2, and 3D, top-down view, city-building game, **Cities: Skylines**.

---

[1]This setting can be seamlessly extended to other digital devices, *i.e.,* mobile phones, game controllers, and virtual reality headsets with standard input and output.

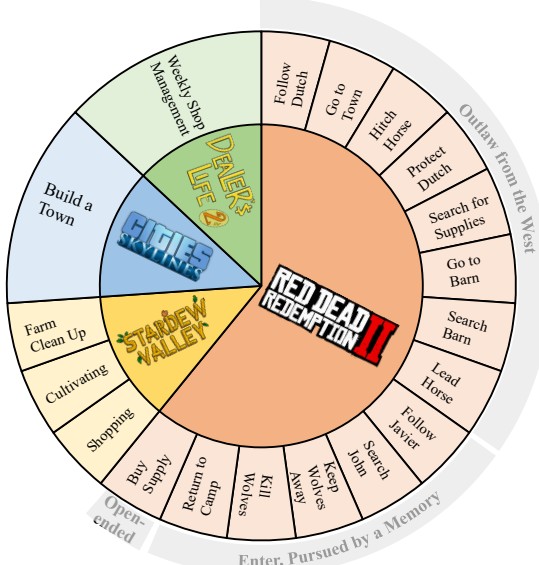 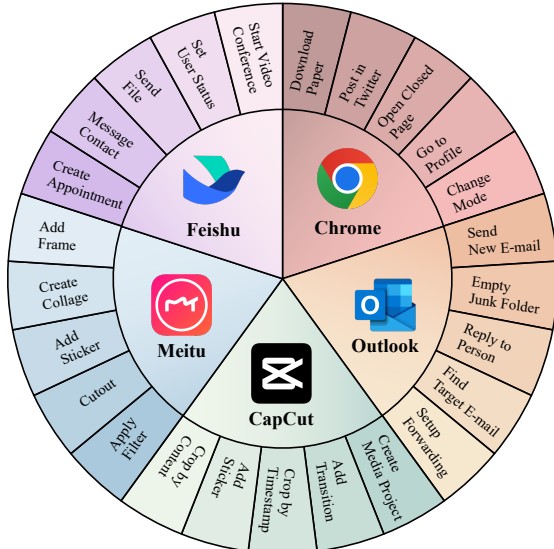

Figure 3: Overview of all game tasks (left) in RDR2, Stardew Valley, Cities: Skylines, and Dealer's Life 2 and application tasks (right) in Chrome, Outlook, CapCut, Meitu, and Feishu.

The target set of diverse software applications for evaluation includes: **Chrome**, **Outlook**, **CapCut**, **Meitu**, and **Feishu**, as well as one comprehensive software benchmark, **OSWorld** (Xie et al., 2024). We provide a brief introduction to these games in Appendix A.

As shown in Figure 3, for each game and software application, representative tasks are designed to measure the various abilities of the agent comprehensively. Experimental results show that CRADLE exhibits remarkable generalization ability and impressive performance across the four previously unexplored commercial video games, the five target software applications, and the comprehensive contemporaneous OSWorld benchmark. To our best knowledge, CRADLE is the first to enable LMM-based agents to follow the main storyline and complete one-hour-long real missions in a complex AAA game, RDR2. CRADLE also manages to create a city with nearly a thousand people in Cities: Skylines, farm and harvest parsnips in Stardew Valley, trade and bargain with a maximal weekly total profit of 87% in Dealer's Life 2. Besides, CRADLE can not only operate daily software, like Chrome and Outlook, but also edit images and videos using Meitu and CapCut, and perform office tasks in Feishu. The success of CRADLE demonstrates the feasibility of challenging GCC setting. Being able to interact with various software in a unified manner, CRADLE greatly extends the reach of AI agents by making it easy to convert any software, especially complex games, into benchmarks and testbeds to evaluate agents' various abilities and facilitate further data collection, paving the way for generalist agents. We hope CRADLE can accelerate the development of more powerful foundation agents, thereby advancing the path towards AGI.

## 2. Related Work

**Agents for Software Applications.** While previous LLM-based web agents (Deng et al., 2023; Zhou et al., 2023; Gur et al., 2023; Zheng et al., 2024b) show some promising results in effectively interacting with content on webpages, they usually use raw HTML code and DOM tree as input and interact with the available element IDs, ignoring the rich visual patterns with key information, like icons, images, and spatial relations. Multimodal web agents (Hong et al., 2023; Furuta et al., 2023; Yan et al., 2023; He et al., 2024; Zheng et al., 2024a) and mobile app agents (Yang et al., 2023b; Wang et al., 2024b) have also been explored. Though using screenshots as input, they still need to use built-in APIs to get the available interactive element IDs to execute corresponding actions. Several recent works (Cheng et al., 2024; Wu et al., 2024; Kapoor et al., 2024) aim to apply web agents to more applications by using keyboard and mouse for control. However, they primarily focus on the static websites and lack the generalizability to other domains.

**Agents for Video Games.** Several attempts try to develop foundation agents for complex video games, such as Minecraft (Wang et al., 2023b;a; 2024a), Starcraft II (Ma et al., 2023) and Civilization-like game (Qi et al., 2024) with textual observations obtained from internal APIs and pre-defined semantic actions. Although JARVIS-1 (Wang et al., 2023a) claims to interact with the environment in a human-like manner with the screenshots as input and mouse and keyboard for control, its action space is predefined as a hybrid space composed of keyboard, mouse, and API. The game-specific observation and action spaces prohibit the generalization of them to other novel games. SIMA(Raad

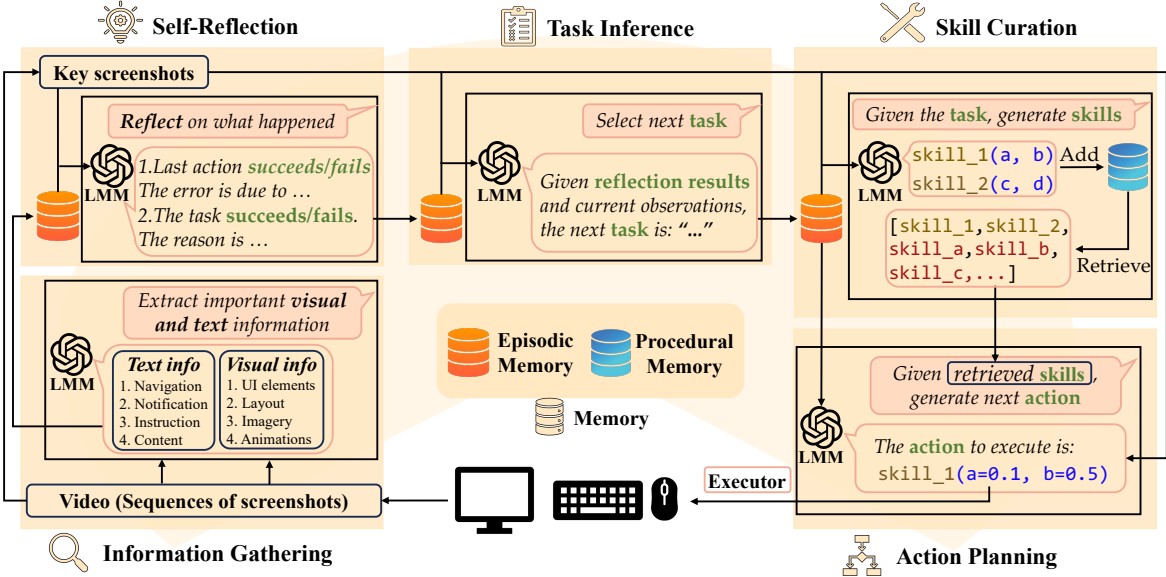

Figure 4: An overview of the **CRADLE** framework, demonstrating the input, output and functionality of each module. **CRADLE** takes video from the computer screen as input and outputs computer keyboard and mouse control determined through inner reasoning.

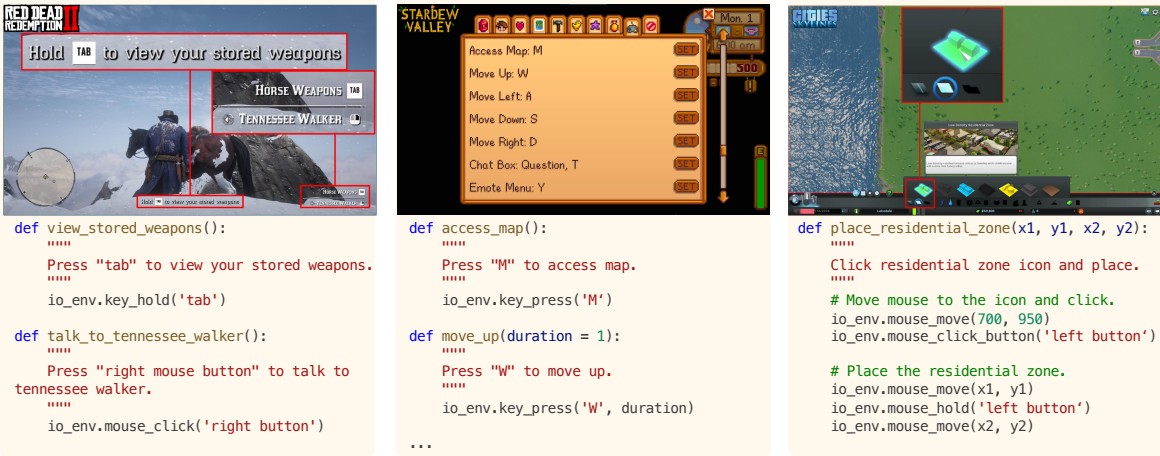

Figure 5: Examples for skill generation according to in-game guidance in RDR2 (left), in-game manual in Stardew Valley (middle), self-exploration in Cities: Skylines (right). Code and comments are shown in brevity.

et al., 2024) trained embodied agents to complete 10-second-long basic tasks over ten 3D video games, and the results are promising to be scaled up.

Due to the space limitation, we provide a detailed discussion of the related work in Appendix B.

## 3. The CRADLE Framework

To pursue GCC, we propose **CRADLE**, illustrated in Figure 4, a modular and flexible LMM-powered framework that can properly handle the challenges GCC presents. The framework should have the ability to understand and interpret computer screens and dynamic changes between consecutive frames from arbitrary software and be able to generate reasonable computer control actions for precise execution. This suggests that a multimodal model with powerful vision and reasoning capabilities, in addition to rich

knowledge of computer UI and control, is a requirement. In this work, we leverage GPT-4o (OpenAI, 2024b) as the framework's backbone model.

### 3.1. Environment IO

**Observation and Action Space.** **CRADLE** only takes a video clip, recording the execution of the last action, as input and outputs keyboard and mouse operations to interact with environments. The observation space is made up of complete screen videos with different lengths. For the action space, it includes all possible keyboard and mouse operations, including `key_press`, `key_hold`, `key_release`, `mouse_move`, and `wheel_scroll`, where keys include both keyboard keys and mouse buttons. These operations can be combined in various ways to form combos and shortcuts, execute rapid key sequences, or co-ordinate timings. Python code is applied to simulate these

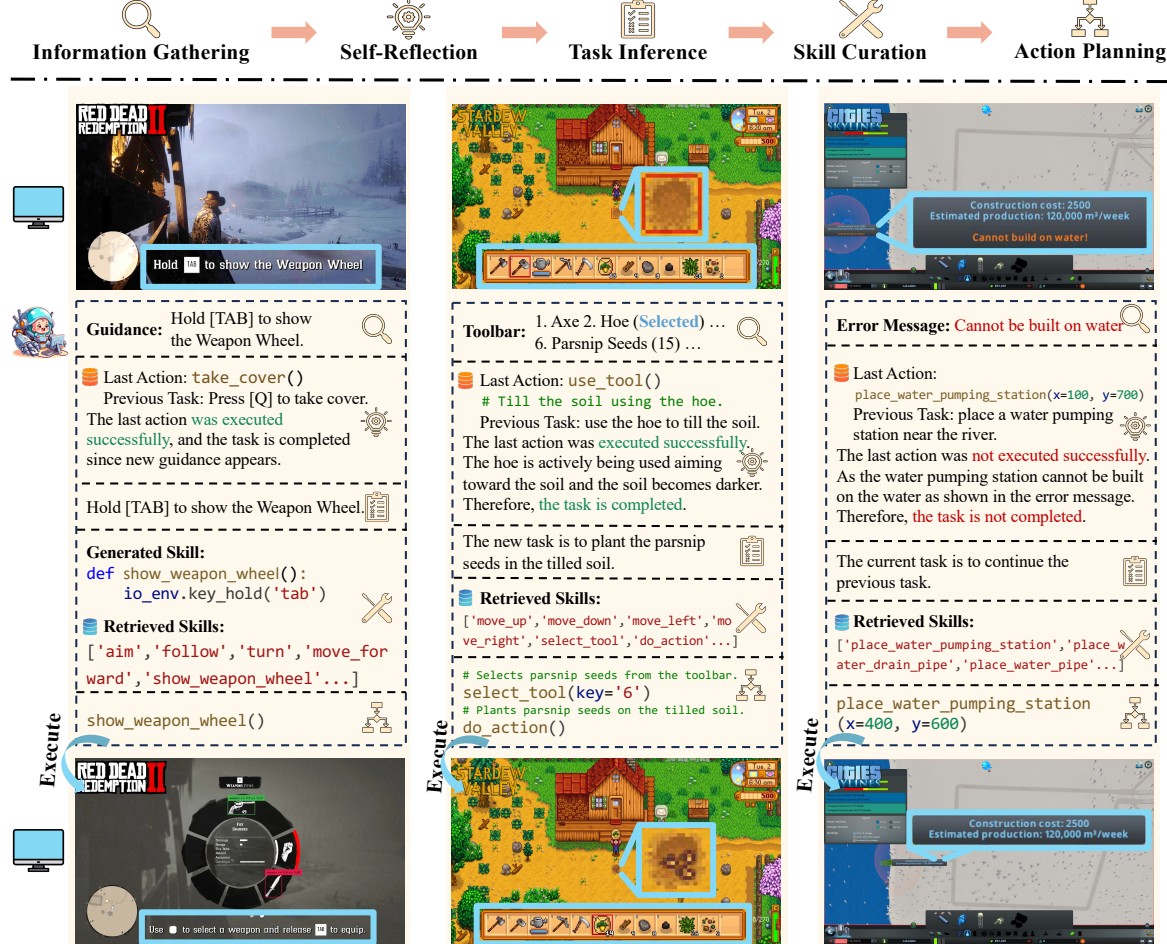

Figure 6: Illustrative examples of **CRADLE**'s complete workflow in RDR2 (left), Stardew Valley (middle) and Cities: Skylines (right). Prompts are shown partially for brevity.

operations and encapsulate them into an `io_env` class.

**Information Gathering.** Provided with a video clip as input, it is critical for **CRADLE** to capture and extract all useful visual and textual information to understand the recent situation and perform further reasoning. Visual information includes layout, imagery, animations, and UI elements which pose high spatial perception and visual understanding requirements for LMM models. Moreover, we depend on their OCR capabilities to extract textual information in images, which usually includes content (headings and paragraphs), navigation labels (menus and links), notifications, and instructions to convey messages and guide users. For each environment, different tools, such as template matching (Brunelli, 2009), Grounding DINO (Liu et al., 2023), and SAM (Kirillov et al., 2023), can be applied to enhance LMMs' abilities to provide additional grounding for object detection and localization.

**Skill and Action Generation** As shown in Figure 5, to bridge the gap between semantic actions generated by LMMs and OS-level executable actions, **CRADLE** uses LMMs to generate code functions as semantic-level skills,

which encapsulate lower-level keyboard and mouse control. Similar to how humans improve while playing, these skills can be developed from scratch according to in-game tutorials and guidance, game manuals and settings, or through self-exploration as the game progresses. These skills can also be pre-defined or composited to solve more complex tasks. An action usually consists of a single or multiple skills instantiated with any necessary parametric aspects, such as duration and position. An *Executor* will be triggered to map these semantic actions to the OS-level keyboard and mouse commands to interact with the environment.

### 3.2. Memory

**CRADLE** stores and maintains all the useful information from the environment or outputted by each module through a memory mechanism, consisting of episodic memory and procedural memory.

**Episodic Memory.** Episodic memory is used to maintain current and past experiences, including key screenshots from each video observation, and everything useful outputted by LMMs and advanced tools, *e.g.,* textual and visual infor-

mation, actions, tasks, and reasoning from each module. To facilitate retrieval and storage, periodical summarization is conducted to abstract recently added multimodal information into long-term summaries. The incorporation of episodic memory enables **CRADLE** to effectively retain crucial information over extended periods.

**Procedural Memory.** This memory is specific to storing and retrieving skills in code form, which can be learned from scratch as shown in Figure 5, or pre-defined in procedural memory. Skills can be added, updated, or composed to the procedural memory in the skill curation module. Same as Voyager (Wang et al., 2024a), skills are retrieved according to the similarities between their corresponding embedding and task description.

### 3.3. Reasoning

Based on the extracted information from observations and memory, **CRADLE** conducts high-level reasoning and then makes the decision. This process is analogous to "**reflect on the past, summarize the present, and plan for the future**", which is broken down into the following modules.

**Self-Reflection.** The reflection module initially evaluates whether the last executed action was successfully carried out and whether the task was completed. Sequential key screenshots from the last video observation, along with the previous context for action planning and task inference are fed to the LMM for reasoning. Additionally, we also request the LMM to provide an analysis of any failure. This valuable information enables **CRADLE** to remedy inappropriate decisions or less-than-ideal actions. Furthermore, reflection can also be leveraged to inform re-planning of the task and bring the agent closer to target task completion, better understand the factors that led to previous successes, or suggest how to update or improve specific skills.

**Task Inference**. After reflecting on the outcome of the last executed action, **CRADLE** needs to analyze the current situation to infer the most suitable task for the current moment. We let LMMs determine the highest priority task to perform and when to stop an ongoing task and start a new one.

**Skill Curation.** As the task is specified, **CRADLE** needs to prepare the tactics to accomplish it, by retrieving useful skills from the procedural memory, updating existing skills, or generating new ones. The new skill will be stored in the procedural memory for future utilization.

**Action Planning.** **CRADLE** needs to select the appropriate skills from the curated skill set and instantiate these skills into a sequence of executable actions by specifying any necessary parametric aspects (*e.g.,* duration, position, and target) according to the current task and history information. The generated action is then fed to the *Executor* for interaction with the environment.

## 4. Empirical Studies

In this section, we present empirical results of deploying **CRADLE** across various challenging environments with GCC setting to demonstrate its comprehensive capabilities.

**Experimental Settings.** If not specifically mentioned, all experiments are conducted in five runs under a maximum step limit, using OpenAI's model, *gpt-4o-2024-05-13* (OpenAI, 2024b). For each video game, we hired five human players, who never played the corresponding game before, to do the evaluation. Before they start the experiments, they will read the prompts used by **CRADLE** agents for fair comparison. Every player played the task once. Similar to SIMA (Raad et al., 2024), we apply human evaluation to all tasks across software and games, except for OSWorld (Xie et al., 2024), which provides automatic evaluation scripts. Due to the dynamism of the RDR2 and Stardew Valley and the LMM inference and communication latency, we need to pause those game environments while waiting for backbone model's responses. Other environments execute continuously. All software and games can be run on regular Windows 10 machines, except for RDR2, which is tested on a machine with an NVIDIA RTX-4090 GPU. We provide estimated experimental cost of the time and API usage in Appendix C, implementation details, environment introduction, task description, case study and prompts in Appendices E to K.

**Task Introduction.** As shown in Figure 7 and 8, for **RDR2**, we mainly focus on evaluating agents on the first two complete missions of the main storyline in Chapter I, which can be divided into 13 tasks according to the in-game checkpoints, including but not limited to navigation, NPC interaction, inventory management, house exploration, and combat. It usually takes a human player about an hour to complete these missions. Few previous studies tackle such long-duration tasks and rich semantic environments. It is an ideal scenario to emulate a novice player learning to play the game from scratch according to the rich in-game tutorials and hints. For **Stardew Valley**, we propose three essential tasks at the stage of the game, *i.e., Farm Clearup*: Clear the obstacles on the farm, such as weeds, stones, and trees, as much as possible to prepare for farming; 2) *Cultivation*: Plant the parsnip seed, water every day and harvest at least one mutual parsnip; 3) *Shopping*: Go to the general store in the town, which is out of the scope of the current map, to buy more seeds and return home. For **Dealer's Life**, the agent is tasked with managing a pawn shop for a week, appraising item values and haggling with the customers to secure deals. For **Cities: Skylines**, the task is to build a reasonable city ending in as much population as possible, with the initial starting funds of ℂ70,000, and basic road, water and electricity facilities. Moreover, we define five representative domain-specific tasks for each of the five

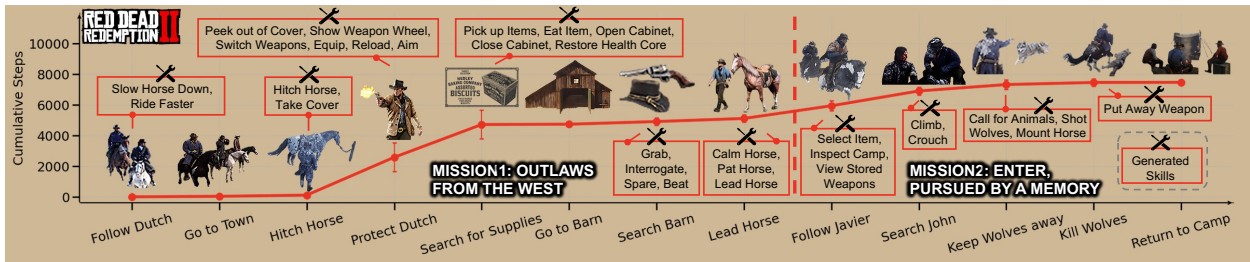

Figure 7: Demonstration of cumulative steps CRADLE takes to complete 13 sequential tasks in the two main storyline missions, starting from the beginning of the game. If a task fails, CRADLE can select the 'retry checkpoint' option to retry the task. Skills generated during the task completion are also illustrated in the figure. We only provide key skills for brevity. Error bars represent the standard deviation of steps needed to complete each task separately.

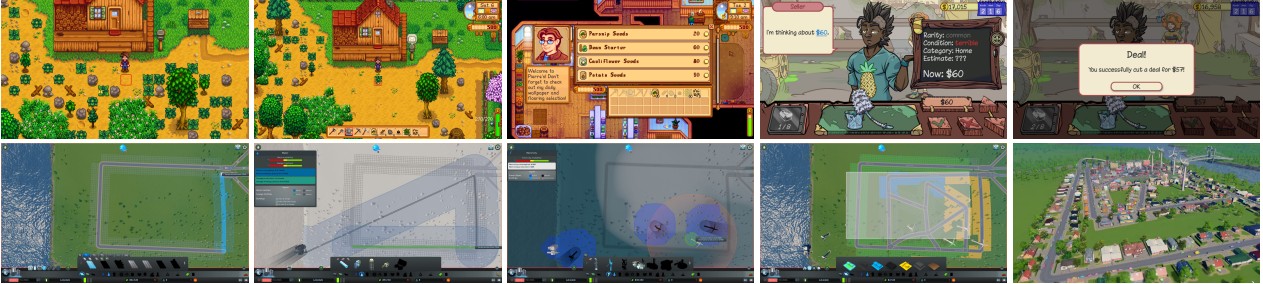

Figure 8: First row shows farm clearup, cultivation and shopping in **Stardew Valley** and haggling and dealing in **Dealer's Life 2**. Second row shows road construction, water pipe laying, wind turbine building, zoning, and the resulting city in **Cities: Skylines**.

**Software Applications** in our diverse target set.

### 4.1. Performance across Environments

**Red Dead Red Redemption 2.** Figure 7 shows that CRADLE can efficiently complete simple navigation tasks with a few steps like following an NPC or going to specific locations on the ground (*e.g., Follow Dutch*, *Go to Town* and *Go to Barn*). Another following task, *Follow Javier*, and the searching task, *Search John*, are dangerous for the rugged and winding path up to the snow mountain with cliffs. Note that CRADLE is able to retry the checkpoint automatically according to the game guidance if the task fails. Therefore, CRADLE takes more steps for retrying the task in these dangerous areas. In addition, CRADLE spends about one-fourth of the total steps in the task of *Protect Dutch*, which is a long-horizontal task with nighttime combat. Many key skills are generated in this task for weapon management and shooting movement. The visibility is poor due to the snow falling in the dark, preventing GPT-4o from accurately recognizing and locating enemies or objects and precisely timing decisions, even equipped with Grounding DINO as an additional detection tool. More times of retry, combined with the need for frequent interactions during combat and the long horizon of the task, lead to this task requiring a large number of steps to complete. The success rate of the combat has significantly improved during the day with much fewer steps for completion, as shown by tasks like *Keep Wolves away*. Additionally, indoor tasks like *Search for Supplies* are also challenging due to GPT4-o's limited spatial perception, which finds it difficult to locate target objects and

ends up circling aimlessly around the house. Moreover, the room contains numerous interactive items unrelated to the task, resulting in much more steps for the agent to complete the task. Overall, CRADLE requires approximately 8,000 steps to complete both missions, taking around 98 minutes of in-game time, compared to the average of 67 minutes for human players. It is the first time for LMM-powered AI agents to exhibit comparable performance in AAA games.

**Stardew Valley.** As shown in Table 1, we surprisingly find that GPT-4o struggles with accurately recognizing and locating objects near the player in this pixel-art game. This leads to difficulties for the agent to interact with objects or people, as it requires the player to stand precisely in front of them in the grid (*e.g.,* when entering doors, using a pickaxe to break stones). It explains the inefficiency in the farming task though the agent manages to clear up most of the obstacles in front of the house within 100 steps and poor performance in the shopping task. On the other hand, relying on episodic summarization and task inference, CRADLE manages to obtain the parsnip by watering the seed for four days and harvesting. Given GPT-4's limited visual capabilities in this game, there is still room for improvement in narrowing the gap between CRADLE and human players.

**Dealer's Life 2.** Table 1 shows that CRADLE demonstrates robust performance and efficient profit-making on the *Weekly Shop Management* task, successfully finalizing 93.6% of potential transactions, with an average of 2 negotiation rounds per customer, and generally aiming for a profit rate of over 50% at the initial offer. It consistently generates profit across all runs, maintaining a total profit

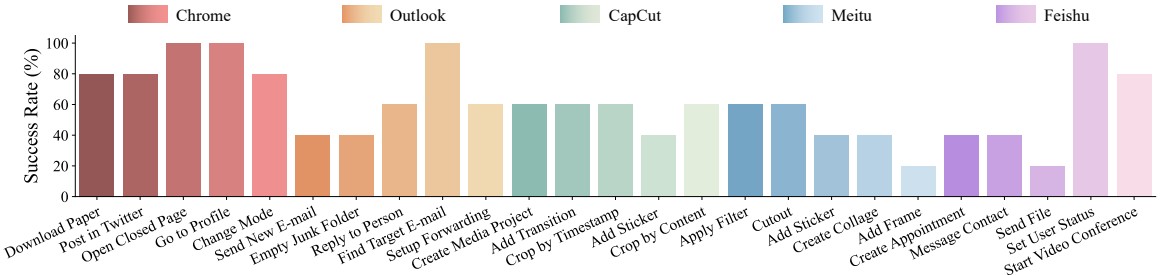

Figure 9: CRADLE's performance in software applications. Each task is run for 5 trials.

Table 1: CRADLE's and human players' performance in Stardew Valley, Dealer's Life 2 and Cites: Skylines with each trial at most 100, 500, 1000 steps respectively. $1/5$ indicates one successful run out of five runs. $x \pm y$ indicate mean and standard deviation.

| Stardew Valley | | |
|---|---|---|
| Task | **CRADLE** | Human |
| Farm Clearup Grids | $14.8 \pm 5.0$ | $\mathbf{35.2 \pm 14.5}$ |
| Cultivation | $4/5$ | $5/5$ |
| Shopping | $1/5$ | $5/5$ |
| **Dealer's Life 2** | | |
| Metrics | **CRADLE** | Human |
| Avg. Haggling Count | $\mathbf{1.95} \pm 0.43$ | $1.63 \pm 0.53$ |
| Turnover Rate (%) | $\mathbf{93.6} \pm 6.9$ | $68.4 \pm 22.2$ |
| Item Profit Rate (%) | $\mathbf{37.8} \pm 19.1$ | $21.1 \pm 13.6$ |
| Total Profit Rate (%) | $\mathbf{39.6} \pm 27.3$ | $17.3 \pm 15.1$ |
| **Cities: Skylines** | | |
| Metrics | **CRADLE** | Human |
| Smooth Closed-loop Road | $4/5$ | $5/5$ |
| Sufficient Water Supply | $1/5$ | $3/5$ |
| Sufficient Power Supply | $5/5$ | $5/5$ |
| High Zoning Coverage | $4/5$ | $4/5$ |
| Population | $\mathbf{450} \pm 224$ | $415 \pm 416$ |

rate of +39.6%, peaking at +87.4% in a single run. In this game, CRADLE significantly outperforms human players. The achievements are mainly attributed to its cautious strategy, by bargaining within a smaller range of price variation but haggling more frequently, resulting in a significantly higher turnover rate. In contrast, human players usually fail the deal due to their aggressive strategy by proposing an unreasonable price and sometimes confusing buy and sell.

**Cities: Skylines.** Table 1 shows that CRADLE is able to complete most of the city design with the averaged maximal population of 450 and the highest single population exceeding 860. CRADLE manages to build the roads in a closed loop to ensure smooth traffic flow, place multiple wind turbines to provide sufficient electricity supply and cover more than 90% of available area with residential, commercial and industrial zones, but fails to provide sufficient water supply for all the regions reliably. The most common failure arises from the missing of water pipes. CRADLE often fail to connect them with each other to cover all zones, resulting in localized water shortages in the city, and preventing

new residents from moving in. The issue also arises from GPT-4o's limited visual understanding, making it difficult to accurately recognize which areas are already covered by the water pipes. We empirically observed that these mistakes usually could be fixed within three unit operations (building or removing a road/facility/a place of zones is counted as one unit operation). Then cities built by CRADLE can eventually reach a population of more than one thousand. We provide a detailed case study in Appendix I.5.2. Overall, as shown in Table 1, without the manual fixes, CRADLE still beats human players even though it suffers from local water storage. Human players typically pay insufficient attention to budget management and tend to allocate a disproportionate amount of funds to the construction of wind turbines for electricity, resulting in limited road construction and residential areas to attract residents.

**Software Applications.** Figure 9 shows CRADLE's performance across tasks on five applications. Multiple tasks remain challenging. Even with a well-known GUI, like Chrome and Outlook, GPT-4o still cannot recognize specific UI items to interact with and also struggles with visual context. For example, forgetting to press the Save button in an open dialog, or not distinguishing between a nearby enabled button vs. a distant and disabled one (*e.g.,* when posting on Twitter). The phenomenon is more severe in the UI with non-standard layouts, like CapCut, Meitu, and Feishu. Lacking prior knowledge by GPT-4o leads to the failure of task inference and selecting the correct skills.

### 4.2. Baseline Comparison

Since no existing methods are fully applicable to the GCC setting, we select several representative methods with necessary adaptions to make them applicable to GCC, labeling them as "like" in Table 2. Compared to CRADLE, React (Yao et al., 2023)-like method only has gather information, skill curation, action planning and procedural memory module, while Reflextion (Shinn et al., 2023)-like method adds a self-reflection and episodic memory, compared to React-like. To show the necessity of multimodal input without access to APIs, we let GPT-4o describe the image and then feed the textual description to Voyager (Wang et al., 2024a)-like as input. Additionally, experiments with GPT-4o and Claude 3 Opus (Anthropic, 2024) as backbone are

Table 2: Baseline comparison for five task in RDR2 and one task in Stardew Valley (*Cultivation*). Numbers before the brackets are average steps with standard deviation for completion. N/A indicates failure for all trials. Every task is run 5 times. Each trial is run for at most 500 steps in RDR2 and 100 steps in Stardew Valley.

| Method | Follow Dutch | Follow Micah | Hitch Horse | Protect Dutch | Search for Supplies | Cultivation |
|---|---|---|---|---|---|---|
| React-like (GPT-4o) | $15 \pm 2$ ($^5/_5$) | $74 \pm 0$ ($^1/_5$) | N/A | N/A | N/A | N/A |
| Reflextion-like (GPT-4o) | $19 \pm 4$ ($^5/_5$) | $58 \pm 14$ ($^2/_5$) | N/A | N/A | N/A | N/A |
| Voyager-like (GPT-4o) | $32 \pm 12$ ($^3/_5$) | N/A | N/A | N/A | N/A | N/A |
| CRADLE (Claude 3 Opus) | $30 \pm 7$ ($^5/_5$) | $52 \pm 17$ ($^4/_5$) | N/A | N/A | N/A | N/A |
| CRADLE (GPT-4o) (Ours) | $13 \pm 3$ ($^5/_5$) | $33 \pm 3$ ($^5/_5$) | $26 \pm 5$ ($^4/_5$) | $461 \pm 0$ ($^1/_5$) | $134 \pm 0$ ($^1/_5$) | $24 \pm 4$ ($^4/_5$) |

conducted. Due to the limitation of requests per minute, other prompting methods like self-consistency (Wang et al., 2022) and TOT (Yao et al., 2024) are not considered. Note that methods here refer to the agents initialized by the corresponding framework with game-specific implementations.

As shwon in Table 2, all the baseline methods can only complete simple and straightforward tasks without complex targets and time delays. Compared to React-like method, Reflection-like method has better performance in the task of *Follow Micah* and still fails to complete more complex tasks, emphasizing the importance of task inference and procedural memory. Voyager-like method that loses vision suffers to accomplish tasks and are the worst of all comparison methods. CRADLE with GPT-4o always has the best performance across all tasks. CRADLE with GPT-4o has the best performance, while Claude 3 Opus fails frequently due to unreliable OCR ability of the guidance, leading to incorrect skill generation and failures of complex tasks.

We provide more analysis in Appendix D, including OS-World results ( D.1), baseline comparison in Stardew Valley ( D.2) and a comprehensive ablation study ( D.3).

## 5. Limitations and Future Work

Despite CRADLE's encouraging performance across games and software, several limitations remain. i) Due to the limited ability of current LMM models, CRADLE struggles in recognizing out-of-distribution (OOD) icons and completing OOD tasks, such as games with non-realistic styles, *i.e.,* Stardew Valley. As LMMs evolve, they can further improve CRADLE's performance. ii) Another general bottleneck for LMM-based agents is the latency caused by the limited inference speed of LMMs, which can also be alleviated as LMMs evolve (*e.g.,* Realtime API (OpenAI, 2024a)). iii) Audio, as an important modality, often plays an important role in games and software; which has not been considered in this work. The future work will be enabling CRADLE to process the audio and graphical input simultaneously. iv) As the preliminary attempt towards GCC, most CRADLE's modules need to call LMM explicitly to process the input for best performance, resulting in frequent interactions with LMM and potentially high costs and long delays. The six

modules represent a problem-solving mindset; as LMM capabilities improve, some or even all of these modules may be combined into a single request. Exploring other potential GCC frameworks is also promising. v) In this work, we mainly focus on enabling foundation agents to interact with various software in a unified manner without taking training into consideration. As SIMA (Raad et al., 2024) has already shown promising results in a similar setting with trained agents, we will let CRADLE autonomously explore and improve over environments through RL (Tan et al., 2023) or collect expert demonstrations for supervised learning (Raad et al., 2024). vi) Due to the large scope of the experiments conducted in this work, the number of runs for each task and human participants are limited. A more comprehensive evaluation can be beneficial. vii) Though CRADLE is broadly applicable to any computer task, only a few selected tasks are investigated in this work. We plan to expand its application to a wider range of targets, delve deeper into complex games, and enhance its adaptability for users.

## 6. Conclusion

We introduce GCC, a general and challenging setting to control diverse video games and software with a unified and standard interface, paving the way towards general foundation agents across all digital world tasks. To properly address the challenges GCC presents, we propose a novel framework, CRADLE, which exhibits strong performance in reasoning and performing actions to accomplish various missions in a set of complex video games and common software applications. To the best of our knowledge, CRADLE is the first framework that enables foundation agents to succeed in such a diverse set of environments without relying on any built-in APIs. The success of CRADLE greatly extends the reach of foundation agents and demonstrates the feasibility of converting any software, especially complex games, into benchmarks to evaluate agents' general intelligence and facilitate further data collection for self-improvement. Although CRADLE still faces difficulties in certain tasks, it serves as a pioneering work to develop more powerful LMM-based agents towards GCC. We hope CRADLE could motivate further study on the challenging but general GCC setting and more applications in diverse software applications and commercial video games.

## Impact Statement

CRADLE holds great potential to improve effective general computer task completion and boost research and deployment of foundation agents. However, there are some risks associated with its misuse or unintended consequences. Below, we outline the potential challenges and safeguards necessary to ensure responsible deployment.

**Development of Game Cheats**. One of the risks associated with CRADLE is its potential misuse in the development of game cheats or bots that could undermine the integrity of online gaming environments. Such misuse could lead to unfair advantages for certain players, disrupt the gaming experience for others, and potentially harm the reputation of game developers. To mitigate this risk, safeguards such as anti-cheat mechanisms and ethical guidelines for the use of AI in gaming should be implemented.

**Incorrect Operations and Harmful Failures**. The autonomous nature of CRADLE means that there is a risk of incorrect or unintended operations, particularly in critical software applications. For example, a malfunctioning agent could inadvertently delete important files, send incorrect emails, or perform other actions with harmful consequences. To address this, robust error-handling mechanisms, human oversight, and fail-safes should be integrated into the system to ensure that it operates safely and reliably.

**Ethical and Privacy Concerns**. The deployment of CRADLE raises ethical and privacy concerns, particularly in scenarios where the agent interacts with sensitive data or performs tasks on behalf of users. For example, an agent operating in a healthcare setting might have access to confidential patient information, raising concerns about data security and privacy. To address these issues, CRADLE should be designed with strong data protection measures, including encryption, access controls, and compliance with relevant regulations.

**Negative Agent Behavior**. There is a risk that CRADLE could exhibit unintended or undesirable behavior, particularly in complex or unpredictable environments. For example, an agent might develop strategies that are effective but unethical, such as exploiting loopholes in software or engaging in deceptive practices. To mitigate this risk, CRADLE should be equipped with ethical guidelines and constraints that prevent harmful behavior. Additionally, ongoing monitoring and evaluation of the agent's actions should be conducted to ensure that it operates in alignment with human values.

## Acknowledgments

We thank Ye Wang and Jiangxing Wang for their time and effort in helping us test the framework. This research is supported by the National Research Foundation, Singapore under its Industry Alignment Fund – Pre-positioning (IAF-PP) Funding Initiative. Any opinions, findings and conclusions or recommendations expressed in this material are those of the author(s) and do not reflect the views of National Research Foundation, Singapore. This research is also supported by the Joint NTU-WeBank Research Centre on Fintech, Nanyang Technological University, Singapore. This work was also supported in part by NSFC under Grant 62450001. Xinrun Xu is advised by Dr. Zhiming Ding from the Institute of Software, Chinese Academy of Sciences, and is supported by the National Key R&D Program of China (No. 2022YFF0503900).

## Team Members and Contributions

### Roles

**Program Leads:** Zongqing Lu, Shuicheng Yan, and Bo An
**Team Lead:** Weihao Tan
**Framework Co-Leads:** Börje F. Karlsson and Weihao Tan
**General Advisors**: Xinrun Wang and Börje F. Karlsson
**Core Contributors**: Weihao Tan, Wentao Zhang, Xinrun Xu, Haochong Xia, Ziluo Ding, Boyu Li, Bohan Zhou, Junpeng Yue, Jiechuan Jiang, Yewen Li, Ruyi An, Molei Qin, Chuqiao Zong, Longtao Zheng, Yujie Wu, Xiaoqiang Chai, Xinrun Wang, and Börje F. Karlsson

### Detailed Contributions

**Framework Design.** Weihao Tan, Börje F. Karlsson, Wentao Zhang, Ziluo Ding, and Xinrun Wang.

**RDR2.** Implementation, experiments, and analysis by Weihao Tan, Börje F. Karlsson, Wentao Zhang, Ziluo Ding, Boyu Li, Bohan Zhou, Junpeng Yue, Haochong Xia, Jiechuan Jiang, Molei Qin, Longtao Zheng, Xinrun Xu, Yifei Bi, Pengjie Gu, and Yewen Li. Xinrun Wang provided insightful suggestions.

**Stardew Valley.** Implementation, experiments, and analysis by Weihao Tan, Wentao Zhang, Chuqiao Zong, Yujie Wu, Xiaoqiang Chai, Haochong Xia, Yewen Li, Ruyi An, and Molei Qin. Xinrun Wang, Long Tian, Chaojie Wang, and Börje F. Karlsson provided insightful suggestions.

**Cities: Skylines.** Implementation, experiments, and analysis by Weihao Tan, Wentao Zhang, Haochong Xia, and Ceyao Zhang. Xinrun Wang provided insightful suggestions.

**Dealer's Life 2.** Implementation, experiments, and analysis by Yewen Li, Ruyi An. Weihao Tan, Wentao Zhang, and Xinrun Wang provided insightful suggestions.

**Software Applications** (including **Chrome**, **Outlook**, **CapCut**, **Meitu**, and **Feishu**). Implementation, experiments,

and analysis by Xinrun Xu, Börje F. Karlsson, and Xiyun Li. Weihao Tan and Xinrun Wang provided insightful suggestions.

**OSWorld.** Implementation, experiments, and analysis by Haochong Xia, Tianbao Xie, Pengjie Gu. Weihao Tan, Xinrun Xu, Xinrun Wang, and Börje F. Karlsson provided insightful suggestions.

**Paper Writing.** Weihao Tan, Xinrun Wang, Börje F. Karlsson, Wentao Zhang, Xinrun Xu, Yewen Li, Ruyi An, and Chuqiao Zong. Haochong Xia, Molei Qin, and Ceyao Zhang further contributed with writers in appendix organization.

**Organization.** Zongqing Lu, Shuicheng Yan, and Bo An provided directional research advice and organizational support.

Weihao Tan's work was conducted during his internships at Skywork AI and BAAI. Longtao Zheng was also an intern at Skywork AI. Xinrun Xu, Bohan Zhou, and Junpeng Yue were interns at BAAI.

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

## A. Game & Task Introduction

The four selected representative games are:

- **Red Dead Redemption 2** (RDR2), an epic AAA 3D role-playing game (RPG) with rich storylines, realistic scenes, and an immersive open-ended world; where players can complete missions by following the instructions, freely explore the world, interact with non-player characters (NPCs) and engage in a variety of activities such as hunting and fishing, in a first- or third-person perspective. This game offers great challenges in 3D embodied navigation and interaction.
- **Stardew Valley**, a 2D pixel-art farming simulation game where players can restore and expand a farm through carefully planned activities such as planting crops, mining, fishing, and crafting. Players can build relationships with the villagers, participate in seasonal events, and uncover the mysteries of the valley. The game encourages strategic planning and time management, as each day brings new opportunities and challenges. Players have to balance their energy and resources to maximize their farm's productivity and profitability.
- **Dealer's Life 2**, a simulation game where players manage a pawn shop. They must assess the value of items, haggle with customers, and make strategic decisions to grow their business. The game offers a dynamic market influenced by trends, customer preferences, and random events, requiring players to adapt and refine their negotiation tactics.
- **Cities: Skylines**, a 3D, top-down view, city-building game where players take on the role of a city mayor, tasked with the development and management of a thriving metropolis, engaging in urban planning by controlling zoning, road placement, taxation, public services, and public transportation in an area. They must balance the needs and desires of the population with the city's budget, addressing issues such as traffic congestion, pollution, and citizen satisfaction. The game provides a sandbox environment where creativity and strategic thinking are key to building efficient and aesthetically pleasing urban landscapes. It also requires highly precise mouse control.

## B. Extended Related Work

### B.1. Environments and Benchmarks for Computer Control

**Environments and Benchmarks on Software Applications**. Simulated environments on computers have been popular benchmarks and testbeds for the research community. Earlier computer control environments primarily focused on web navigation tasks (Shi et al., 2017; Liu et al., 2018; Yao et al., 2022; Deng et al., 2023; Zhou et al., 2023; Koh et al., 2024). Recent benchmarks start to include various common software (Kapoor et al., 2024; Xie et al., 2024), aiming to develop a generalist agent in the digital world. However, none of them takes video games into consideration, missing a key component of computer control.

**Environments and Benchmarks on Video Games**. On the other side, many research environments are built on top of video games, significantly advancing the study of decision-making, especially, reinforcement learning (RL). Examples include but are not limited to Atari games (Bellemare et al., 2013), Super Mario Bros (Kauten, 2018), Google Research Football (Kurach et al., 2020), Minecraft (Johnson et al., 2016; Guss et al., 2019; Fan et al., 2022), Dota II (Berner et al., 2019), StarCraft II (Vinyals et al., 2019; Samvelyan et al., 2019; Ellis et al., 2023), Quake III (Jaderberg et al., 2019), Gran Turismo (Wurman et al., 2022), Diplomacy (Bakhtin et al., 2022) and Civilization (Qi et al., 2024). Additionally, many custom-built environments, especially grid world and embodied scenarios, are created from scratch in a game-like manner to facilitate agent development, such as BabyAI (Chevalier-Boisvert et al., 2019), Melting Pot (Leibo et al., 2021), Overcooked (Carroll et al., 2019; Wu et al., 2021; Xiao et al., 2022), VRKitchen (Gao et al., 2019), VirtualHome (Puig et al., 2018), iGibson (Shen et al., 2021; Li et al., 2021), ProcTHOR (Deitke et al., 2022), Habitat (Manolis Savva* et al., 2019; Szot et al., 2021; Puig et al., 2023), and Generative agents (Park et al., 2023).

Each of these environments highly relies on the accessibility of the open-source code or provided built-in APIs. Significant human efforts are required for implementation and encapsulation, enabling agent interaction. Therefore, despite the abundance of software and games available for human use, only a limited number are accessible to agents, especially for commercial closed-source games and software applications. Additionally, the lack of consensus on environment standards further complicates the interaction, as each environment has specific observation and action spaces, tailored to its unique requirements. This variation exacerbates the challenge of enabling agents to interact with diverse environments and collect data with a consistent level of fine-grained semantics to improve the agent's capabilities. Few agents can complete tasks across multiple environments so far.

Similar to OpenAI Universe (OpenAI, 2016) and SIMA (Raad et al., 2024), our goal is to explore a unified way that allows agents to interact for measuring and training agents' abilities across a wide range of games, websites, and other applications

without heavy human efforts needed. This approach aims to prove that diverse software applications and games can serve as out-of-the-box environments for AI development.

### B.2. LMM-based Agents for Computer Tasks

**Agents for Software Manipulation.** Agents for software applications are developed to complete tasks such as web navigation (Zhou et al., 2023; Deng et al., 2023; Mialon et al., 2023) and software application control (Rawles et al., 2023; Yang et al., 2023b; Kapoor et al., 2024). While previous LLM-based web agents (Deng et al., 2023; Zhou et al., 2023; Gur et al., 2023; Zheng et al., 2024b) show some promising results in effectively interacting with content on webpages, they usually use raw HTML code and DOM tree as input and interact with the available element IDs, ignoring the rich visual patterns with key information, like icons, images, and spatial relations. Recently, multimodal web agents (Yan et al., 2023; Gao et al., 2023; He et al., 2024; Zheng et al., 2024a; Niu et al., 2024; Zhang et al., 2024; Wu et al., 2024) and mobile app agents (Yang et al., 2023b; Wang et al., 2024b) have been explored. Though using screenshots as input, they still rely on built-in APIs and advanced tools to get internal information, like available interactive element IDs, to execute corresponding actions, which greatly limits their applicability. Other train-based agents (Hong et al., 2023; Furuta et al., 2023; Cheng et al., 2024) also suffer from generalizing to unseen software and tasks. Moreover, all of these works primarily focus on static websites and software, which greatly reduces the need for timeliness and simplifies the setting by ignoring the dynamics between adjacent screenshots, *i.e.,* animations, and incomplete action space without considering the duration of the key press and different mouse mode. It results in the failure of deployment to the tasks with rapid graphics changes, *e.g.,* game playing.

**Agents for Game Playing.** Several attempts try to develop foundation agents for complex video games, such as Minecraft (Wang et al., 2023b;a; 2024a), Starcraft II (Ma et al., 2023) and Civilization-like game (Qi et al., 2024) with textual observations obtained from internal APIs and pre-defined semantic actions. Although JARVIS-1 (Wang et al., 2023a) claims to interact with the environment in a human-like manner with the screenshots as input and mouse and keyboard for control, its action space is predefined as a hybrid space composed of keyboard, mouse, and API. The game-specific observation and action spaces prohibit the generalization of them to other novel games. Pre-trained with videos with action labels, VPT (Baker et al., 2022) manages to output mouse and keyboard control with raw screenshots as input without any additional information. However, collecting videos with action labels is time-consuming and costly, which is difficult to generalize to multiple environments. Another concurrent work, SIMA (Raad et al., 2024) trained embodied agents to complete 10-second-long tasks over ten 3D video games. Though their results are promising to scale up, they focus on behavior cloning with gameplay data from human experts, resulting in a high expense.

In both targeting complex video games and diverse software applications, CRADLE attempts to explore a new way to efficiently interact with different complex environments in a unified manner and facilitate further data collection. In a nutshell, to our best knowledge, there are currently no agents under the GCC setting, reported to show superior performance and generalization in complex video games and across computer tasks. In this work, we make a preliminary attempt to explore and benchmark diverse environments in this setting, applying our framework to diverse challenging environments under GCC and proposing an approach where any software can be used to benchmark agentic capabilities in it.

## C. Experimental Cost

Table 3: Financial and time-related costs of running all the tasks once in each environment or domain.

| | RDR2 | Cities: Skylines | Stardew Valley | Dealer's Life 2 | Software Apps | OSWorld | Total |
|---|---|---|---|---|---|---|---|
| Tasks Num. | 14 | 1 | 3 | 1 | 25 | 369 | - |
| Input Tokens | 600M | 150M | 60M | 25M | 45M | - | - |
| Output Toekns | 20M | 7.5M | 4M | 1M | 2.5M | - | - |
| Cost (USD) | $3300 | $862.5 | $345 | $140 | $262.5 | $500 | $5410 |
| Time | 240 hrs | 60 hrs | 30 hrs | 20 hrs | 50 hrs | 240 hrs | 640 hrs |

Table 3 shows the approximate cost of experiments in Section 4.1 with gpt-4o-2024-05-13. Baselines comparison and ablation studies are not included. Since all the tasks were run 5 times except for OSWorld once, the total cost of getting all the results shown in Section 4.1 is approximately 5400 USD. claude-3-opus-20240229 will roughly use 3X more money and

2X more time compared to gpt-4o-2024-05-13, due to its higher price and longer latency. We also want to note that with the latest model, gpt-4o-2024-08-06, the cost will be halved. We estimate that costs will decrease by one or two orders of magnitude in the coming few years. Then the cost will be affordable to every researcher and developer.

## D. Additional Results

### D.1. Results in OSWorld

Table 4 shows that CRADLE achieves the overall highest success rate in OSWorld, compared to the baselines without relying on any internal APIs to provide extra grounding labels, *e.g.*, Set-of-Mark (SoM) (Yang et al., 2023a). The information gathering module improves grounding for more precise action execution, increasing the performance. The self-reflection module enables CRADLE to predict infeasible tasks and subsequently fix mistakes, shown in the Professional domain results, where it achieves a $20.41\%$ success rate, significantly surpassing the baselines.

Table 4: Success rates (%) of different methods in OSWorld.

| Method | Office (117) | OS (24) | Daily (78) | Workfl-ow(101) | Professi-onal (49) | All (369) |
|---|---|---|---|---|---|---|
| GPT-4o | **3.58** | 8.33 | 6.07 | **5.58** | 4.08 | 5.03 |
| GPT-4o+SoM | **3.58** | **20.83** | 3.99 | 3.60 | 2.04 | 4.59 |
| CRADLE | **3.58** | 16.67 | **6.55** | 5.48 | **20.41** | **7.81** |

### D.2. Baseline Comparison in Stardew Valley

Figure 10 provides the detailed performance of each baseline method in the *Cultivation* task in Stardew Valley. Without task inference and episodic memory for summarization, even React-like and Reflexion-like methods sometimes managed to get the parsnip to sprout from the ground, they failed to harvest it because GPT-4o failed to recognize the mature parsnip. Episodic memory can help CRADLE record the days of watering and know when the crop can be harvested. Voyager-like method struggles with getting out of the house and returning home due to the lack of visual input. Claude 3 Opus also has difficulties in localizing the position of the character and the crop. Moreover, it prefers moving characters much more frequently than GPT-4, resulting in the failure to position the character in front of the crop.

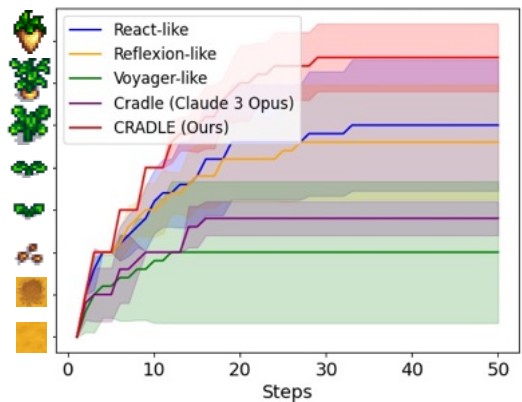

Figure 10: Performance of each method in task *Cultivation*. The Y-axis shows the stage of parsnip. Only if the mutual parsnip (shown on the top of the y-axis) is obtained will this trial be counted as a success.

### D.3. Ablation Study

Besides comparing with other baseline methods, we provide a complete ablation study by systematically removing each module of CRADLE to show the effectiveness in Table 5. We mainly show the results of 6 consecutive subtasks at the beginning of the main storyline, separated from the tasks of *Follow Micah*, *Hitch Horse* and *Protect Dutch* in RDR2. Note that the combination of skill curation, action planning and procedural memory is the minimal unit of our framework. Without

any of them, the agent cannot generate and execute valid actions successfully. So these modules are not ablated.

The most significant decline in agent capabilities arises from the absence of the information gathering module. Without this module, the agent is unable to extract key information in the observation, which is critical for all other modules to function effectively. The second largest impact comes from the lack of the self-reflection module, which is instrumental in correcting mistakes and recognizing when the agent is stuck, such as in the subtask of *Go to Shed*. Third, the task inference module is vital for tasks that require strict adherence to guidance, like *Switch Weapon*. In these cases, the in-game instructions appear only at the beginning of the task, as seen in *Follow Micah* and *Go to Shed*. Lastly, episodic memory becomes increasingly important as tasks grow more complex, requiring more steps to complete, such as in *Go to Shed* and *Combat*, which involve far more steps than other subtasks. Overall, each module plays a crucial and distinct role in the CRADLE framework. Removing or isolating any of them significantly reduces the agent's effectiveness, underscoring the importance of their integrated function.

Table 5: Success rates of each variant by systematically removing CRADLE's module on six consecutive subtasks in RDR2. Every subtask is run 5 times. Each of subtasks are run for at most 500 steps.

| Subtask | w/o Information Gathering | w/o Self-Reflection | w/o Task Inference | w/o Episodic Memory | CRADLE |
|---|---|---|---|---|---|
| Follow Micah | 0% | 0% | 40% | 80% | **100%** |
| Hitch Horse | 0% | **100%** | **100%** | **100%** | **100%** |
| Go to Shed | 0% | 20% | 40% | 20% | **80%** |
| Peek out of Cover | 60% | **100%** | 80% | **100%** | **100%** |
| Switch Weapon | 0% | 80% | 60% | 80% | **100%** |
| Combat | 0% | 0% | 0% | 0% | **20%** |

## E. General Implementation

Here we introduce the general implementation details of CRADLE. For specialized implementations addressing issues unique to their own environment, please refer to the corresponding section.

**Hardware.** All software and games can be run on regular Windows 10 machines, except for RDR2, which is tested on machines with RTX-4090 GPU separately.

**Backbone Model.** We employ GPT-4o (OpenAI, 2024b), currently one of the most capable LMM models, as the framework's backbone model. If not mentioned explicitly, all the experiments are done with *gpt-4o-2024-05-13*. Temperature is set to 0 to lower the variance of the text generation. Same as Voyager (Wang et al., 2024a), we use OpenAI's *text-embedding-ada-002 model* (OpenAI, 2022) to generate embeddings for each skill, stored in the procedural memory and retrieved according to the similarities.

**Evaluation Methods.** Unlike conventional research benchmarks, which usually provide grounding signals for evaluation, it is difficult to have a unified and general method to determine whether a task is completed automatically in diverse software, especially in video games. Similarly to SIMA (Raad et al., 2024), we apply human evaluation to all tasks across application software and games. Moreover, to provide more quantitative results and a comparison baseline, we provide results for the OSWorld (Xie et al., 2024) benchmark, a contemporaneous benchmark that provides evaluation scripts for at least one solution per task.

**Observation Space.** CRADLE only takes a video clip, which records the progress of execution of the last action, as input. To lower the frequency of interaction with backbone models and reduce the strain on the computer, video is recorded at 2 fps (a screenshot every 0.5 seconds), which proves to be sufficient in most cases for information gathering without missing any important information. It is important to note that, due to the dynamism of the RDR2 and Stardew Valley and the LMM inference and communication latency, we must pause those game environments while waiting for backbone model responses. Other environments execute continuously.

**Action Space.** For the action space, it includes all possible keyboard and mouse operations, including `key_press`, `key_hold`, `key_release`, `mouse_move`, `mouse_click`, `mouse_hold`, `mouse_release`, and `wheel_scroll`, which can be combined in different ways to form combos and shortcuts, use keys in fast sequence, or coordinate timings. We choose to use Python code to simulate these operations and encapsulate them into an `io_env` class. Skill code needs to be generated by the agent in order to utilize such functions and affordances so executed actions take effect. Table 6 illustrates CRADLE's action space.

Table 6: Action space in the CRADLE framework, including action attributes. Coordinate system is either *absolute* or *relative*. Actions with durations can be either *synchronous* or *asynchronous*.

| Type | Action | Attributes |
|------|--------|-----------|
| **Keyboard** | Key Press | Key name (string),
Key press duration (seconds:float) |
| | Key Hold | Key name (string) |
| | Key Release | Key name (string) |
| | Key Combo | Key names (strings),
Key combo duration (seconds:float),
Wait behaviour (sync/async) |
| | Hotkey | Key names (strings),
Hotkey sequence duration (seconds:float),
Wait behaviour (sync/async) |
| | Text Type | String to type (string),
Typing duration (seconds:float) |
| **Mouse** | Button Click | Mouse button (left/middle/right),
Button click duration (seconds:float) |
| | Button Hold | Mouse button (left/middle/right) |
| | Button Release | Mouse button (left/middle/right) |
| | Move | Mouse position (width:int, height:int),
Mouse speed (seconds:float),
Coordinate system (relative/absolute),
Tween mode (enum) [2] |
| | Scroll | Orientation (vertical),
Distance (pixels:int),
Duration (seconds:float) |
| **Wait** | Noop | - |

It is important to note that, while some works (*e.g.,* AssistantGUI (Gao et al., 2023), OmniACT (Kapoor et al., 2024) and OSWorld (Xie et al., 2024)) use *PyAutoGUI* [3] for keyboard and mouse control, this approach does not work in all applications, particularly in modern video games using DirectX [4]. Moreover, such work chooses to expose a subset of the library functionality in its action space, ignoring dimensions like press duration and movement speed, which are critical in many scenarios (*e.g.,* RDR2, for opening the weapon wheel and changing view).

To ensure wide game and software compatibility and accommodate different operating systems, in our current implementation we use the similar *PyDirectInput* library [5] and *PyAutoGUI* for keyboard control, utilize *AHK* [6] and write our own abstraction (using the *ctypes* library [7]) to send low-level mouse commands to the operating system for mouse control. For increased portability and ease of maintenance, all keyboard and mouse control is encapsulated in a class, called *IO_env*.

Notably, our low-level control wrapper is adapted for both MacOS and Windows systems, making the OS transparent to us. At the software window level, we implemented automatic switching between the target software window and the window running the agent (using Python *ctypes* for Windows and *AppleScript* for MacOS [8]).

---

[3] Python library that provides a cross-platform GUI automation module - https://github.com/asweigart/pyautogui

[4] Microsoft DirectX graphics provides a set of APIs for high-performance multimedia apps - https://learn.microsoft.com/en-us/windows/win32/directx

[5] Python library encapsulating Microsoft's *DirectInput* calls for convenience manipulating keyboard keys - https://github.com/learncodebygaming/pydirectinput

[6] A fully typed Python wrapper around AutoHotkey to keyboard and mouse control - https://github.com/spyoungtech/ahk

[7] Python library that provides C compatible data types, and allows calling functions in DLL/.so binaries - https://docs.python.org/3/library/ctypes.html

[8] AppleScript is a scripting language created by Apple, which allows users to directly control scriptable applications, as well as parts of MacOS - https://developer.apple.com/library/archive/documentation/AppleScript/Conceptual/

**Procedure Memory.** This memory stores pre-defined basic skills and the generated skills captured from the *Skill Curation*. However, as we continuously obtain new skills during game playing, the number of skills in procedural memory keeps increasing, and it is hard for GPT-4o to precisely select the most suitable skill from the large memory. Thus, similar to Voyager (Wang et al., 2024a), we use OpenAI's *text-embedding-ada-002 model* (OpenAI, 2022) to generate embeddings for each skill and store pre-defined basic skills and any generated skills captured from *Skill Curation*, along with their embeddings in a procedural memory. We retrieve a subset of skills, that are relevant to the given task, and then let GPT-4o select the most suitable one from the subset. In the skill retrieval, we pre-compute the embeddings of the documentations (code, comments and descriptions) of skill functions, which describe the skill functionality, and compute the embedding of the given task. Then we compute the cosine similarities between the skill documentation embeddings and the task embedding. The higher similarity means that the skill's functionality is more relevant to the given task. We select the top K skills with the highest similarities as the subset. Using similarity matching to select a small candidate set simplifies the process of choosing skills.

**Episodic Memory.** This memory stores all the useful information provided by the environment and LMM, which consists of short-term memory and long-term summary.

The short-term memory stores the screenshots within the recent k interactions in game playing and the corresponding information from other modules, *e.g.,* screenshot descriptions, task guidance, actions, and reasoning. We set k to five, and it can be regarded as the memory length. Information stored over k interactions ago will be forgotten from direct short-term memory. Empirically, we found that recent information is crucial for decision-making, while a too-long memory length would cause hallucinations. In addition, other modules continuously retrieve recent information from short-term memory and update the short-term memory by storing the newest information.

For some long-horizon tasks, short-term memory is not enough. This is because the completion of a long-horizon task might require historical information from a long steps ago. For example, the agent might do a series of short-horizon tasks during a long-horizon task, which makes the original long-horizon task forgotten in short-term memory. To maintain the long-term valuable information while avoiding the long-token burden of GPT-4o, we propose a recurrent information summary as long-term memory, which is the text summarization of experiences in game playing, including the ongoing task, the past entities that the player met, and the past behaviors of the player and NPCs.

In more detail, we provide GPT-4o with the summarization before the current screenshot and the recent screenshots with corresponding descriptions, and GPT-4o will make a new summarization by organizing the tasks, entities, and behaviors in the time order with sentence number restriction. Then we update the summarization to be the newly generated one, which includes the information in the current screenshot. The recurrent summarization update, inspired by RNN, achieves linear-time inference by preserving a hidden state that encapsulates historical input. This method ensures the compactness of summarization token lengths and recent input data. Furthermore, the incorporation of long-term memory enables the agent to effectively retain crucial information over extended periods, thereby enhancing decision-making capabilities.

**Information Gathering.** Given the video clip as input, we mainly depend on GPT-4o's OCR capabilities to extract textual information in the keyframes, which usually contain critical guidance and notifications for the current situation. We also rely on GPT-4o's visual understanding to analyze the visual information in the frames. Besides, we augment LMMs' visual understanding via some tools, like template matching (Brunelli, 2009), Grounding DINO (Liu et al., 2023), and SAM (Kirillov et al., 2023), to provide additional grounding for object detection and segmentation. Some visual prompting tricks, like drawing axes and colorful directional bands, are also applied to enhance the GPT-4o's visual ability.

**Task Inference.** After reflecting on the outcome of the last executed action, We let GPT-4o analyze the current situation to infer the most suitable task for the current moment and estimate the highest priority task to perform and when to stop an ongoing task and start a new one.

**Skill Curation.** GPT-4o is required to strictly follow the provided interfaces and examples to generate the corresponding code for new skills. Moreover, GPT-4o is required to include documentation/comments within the generated code, delineating the functionality of each skill. *Procedural Memory* where skills are stored will then check whether the code is valid, whether the format of documentation is right, and whether any skill with the same name already exists. If all conditions are passed, the newly generated skill is persisted for future utilization.

**Action Planning.** GPT-4o needs to select the appropriate skills from the curated skill set and instantiate these skills into a

`AppleScriptLangGuide/introduction/ASLR_intro.html`

sequence of executable actions by specifying any necessary parametric aspects (*e.g.,* duration, position, and target) according to the current task and history information. The generated action is then fed to the ***Executor*** for interaction with the environment.

**Prompts.** Prompts used by each module are initialized by the corresponding templates in Markdown-style format. These prompt templates provide a minimal workflow with basic rules for the module to run and use placeholders of each key for input and output. CRADLE automatically retrieves the corresponding value for each key in the input from the episodic memory and forms valid requests to query LMMs with the values and templates. After receiving responses from LMMs, CRADLE automatically extracts the keys in the output and stores them in the episodic memory. Users can freely customize their own prompts without writing any code.

**Apply to new environments.** Theoretically, CRADLE can be directly deployed to new video games or other software applications with the default prompt templates and empty procedural memory. Due to the limited ability of current LMMs and the complexity of challenging environments and tasks, prompt engineering may need to be applied to every module to enhance LMMs' reasoning ability and introduce domain knowledge. Additional tools can also be applied to provide extra grounding and domain knowledge as part of the prompt input. Procedural memory can be initialized with hand-craft skills to mitigate the incomplete tutorials provided by the software and the complexity of tasks. Users may need to analyze the task-specific issue and choose a suitable solution. We provide all the implementation details for each video game and software in Appendices E to K.

## F. Red Dead Redemption II

### F.1. Introduction to RDR2

Red Dead Redemption II (RDR2) is an epic AAA Western-themed action-adventure game by Rockstar Games. As one of the most famous and highest-selling games in the world, it is widely acknowledged for its movie-like realistic scenes, rich storylines, and immersive open-ended world. The game applies a typical role-playing game (RPG) control system, played from a first- or third-person perspective, which uses WASD for movement, mouse control for view changing, first- or third-person shooting for combat, and inventory and manipulation.

For most of the game, players need to control the main character, Arthur Morgan, upon choosing to complete mission scenarios following the main storyline. Otherwise, they can freely explore the interactive world, such as going hunting, fishing, chatting with non-player characters (NPCs), training horses, witnessing or partaking in random events, and participating in side quests. As the main storyline progresses, different skills are gradually unlocked. As a close-source commercial game, no APIs are available for obtaining additional game-internal information nor pre-defined automation actions. Following its characteristics, this game serves as a fitting and challenging environment for the GCC setting and a comprehensive benchmark for embodiment.

### F.2. Objectives

In Chapter 1 of RDR2, the first two missions of the main storyline are *Outlaws from the West* and *Enter, Pursued by a Memory*. These missions serve as the tutorial content for RDR2, guiding players step-by-step into the role of Arthur. They immerse the player in the story's development while teaching the game's controls and mechanics.

We divided Mission 1 and Mission 2 into 8 and 5 tasks respectively based on the checkpoints within each mission. Each checkpoint may present failure scenarios. For example, in Mission 1, there are six failure scenarios: i) Assaults, kills, or abandons Dutch or Micah; ii) Allows Dutch or Micah to be killed; iii) Abandons the homestead; iv) Assaults, kills, or abandons their horse; v) Assaults, kills, or abandons the horse in the barn; vi) Dies. We categorized each sub-task as either "Easy" or "Hard" based on the likelihood of failure at each checkpoint and the need to retry the checkpoint.

To evaluate CRADLE's capabilities in an open-world environment, Mission 3 is designed as a hard open-ended task. Unlike the first two tutorial missions, it does not include any checkpoints. Consequently, the entire Mission 3 is treated as a single, comprehensive task. Although we do not subdivide Mission 3 into finer tasks, we aim to identify key points to facilitate a clearer understanding of Mission 3 for the reader.

Tables 7 and 8 provide a brief introduction of each task in the first two missions of the main storyline and an open-ended mission, along with approximate estimates of their difficulty. Due to GPT-4o's poor performance in spatial understanding and fine-manipulation skills, it can be challenging for our agent to perform certain actions, like entering or leaving a building,

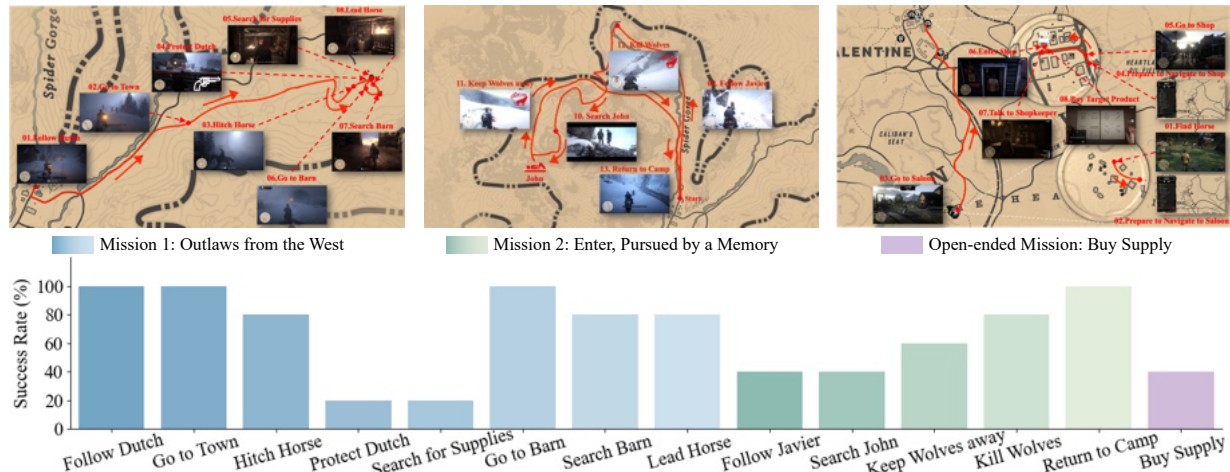

Figure 11: Trajectory and success rates of 13 main storyline tasks and 1 open-world task in RDR2. Each task is run 5 times and each trial is run for at most 500 steps. Long horizontal and challenging tasks like *Protect Dutch* and *Search for Supplies* usually need several times of retry to complete, resulting in the demand for more steps. It explains the low success rate of these tasks within 500 steps.

or going to precise indoor locations to retrieve specific items. Additionally, the high latency of GPT-4o's responses also makes it harder for an agent to deal with time-sensitive events, *e.g.,* during combat.

### F.3. Implementation Details

Our experiments are based on the latest version of RDR2, 'Build 1491.50'. As shown in Figure 15, strictly following the GCC setting, our agent takes the video of the screen as input and outputs keyboard and mouse operations to interact with the computer and the game. An observation thread is responsible for the collection of video frames from the screen and each video clip records the whole in-game process since executing the last action.

**Information Gathering.** To extract keyframes from the video observation, we utilize the VideoSubFinder tool [9], a professional subtitle discovery and extraction tool. These keyframes usually contain rich meaningful textual information in the game, which are highly relevant to the completion of tasks and missions (such as character status, location, dialogues, in-game prompts and tips, etc.) We use GPT-4o to extract and categorize all the meaningful contexts in these keyframes and perform OCR, and call this processing "gathering text information". Then, to save interactions with GPT-4o, we only let GPT-4o provide a detailed description of the last frame of the video.

While GPT-4o exhibits impressive visual understanding abilities across various CV tasks, we find that it struggles with spatial reasoning and recognizing some game-specific icons. To address these limitations, we add a visual augmentation sub-module within our *Information Gathering* module. This augmentation step serves two main purposes: i) utilize Grounding DINO (Liu et al., 2023), an open-set object detector, to output precise bounding boxes of possible targets in an image and serve as spatial clues for GPT-4o; and ii) perform template matching (Brunelli, 2009) to provide icon recognition grounding truth for GPT-4o when interpreting instructions or menus shown on screen. As LMM capabilities mature, it should be possible to disable such augmentation.

**Self-Reflection.** The reflection module mainly serves to evaluate whether the previously executed action was successfully carried out and whether the current executing task is finished. To achieve this, we uniformly sample at most 8 sequential frames from the video observation since the execution of the last action and use GPT-4o to estimate the success of its execution. Additionally, we expect GPT-4o can also provide analysis for any failure of the last action (*e.g.,* the move-forward action failed and the cause could be the agent was blocked by an obstacle). With such valuable information as input for *Action Planning*, including the failure/success of the last action and the corresponding analysis, the agent is capable of attempting to remedy an inappropriate decision or action execution.

Moreover, some actions require prolonged durations, such as holding down specific keys, which can coexist or interfere with other actions decided by subsequent decisions. Consequently, the reflection module must also decide whether an ongoing

---
[9]VideoSubFinder standalone tool - https://sourceforge.net/projects/videosubfinder/

Table 7: Tasks in the first two missions of RDR2. In the tutorial guide, the prompt text *Start Dialogue* signifies the end of the previous checkpoint and the beginning of the current checkpoint. *Difficulty* refers to how hard to accomplish the corresponding tasks. Figures 12 and 13 showcase snapshots of each task (specific sub-figures marked in parenthesis in the table). The maximal number of steps (agent takes one action) for each task is 500.

| Mission 1: Outlaws from the West | Description | Start Dialogue | Difficulty |
|---|---|---|---|
| Follow Dutch (Fig. 12a) | Arthur follows Dutch on horseback into the snow to find their scouting gang members. | Use [W] to Follow Dutch | Easy |
| Go to Town (Fig. 12b) | Arthur rides his horse, following Micah to the vicinity of a little homestead Micah discovered. | Hold [W] to match speed with Dutch and Micah | Easy |
| Hitch Horse (Fig. 12c) | Arthur hitches the horse to the hitching post, then goes to the old shed and takes cover. | Hold [E] to hitch your horse | Easy |
| Protect Dutch (Fig. 12d) | Arthur uses his gun to shoot all of the O'Driscolls inhabiting the house and protect Dutch. | Use [W] to peak out of cover | Hard |
| Search for Supplies (Fig. 12e) | Arthur follows Dutch to the house to search for supplies. | Hold [R] near items to pick the up while searching house. | Hard |
| Go to Barn (Fig. 12f) | Arthur follows Dutch's directions and goes to the barn to see if there's anything inside. | Dutch: Micah, Arthur, keep looking for stuff | Easy |
| Search Barn (Fig. 12g) | Arthur searches the barn and defeats the O'Driscoll hiding inside. | [F] Attack the O'Driscoll | Hard |
| Lead Horse (Fig. 12h) | Arthur calms the horse and takes it out of the barn. | Hold [Right Mouse Button] to focus on the horse | Easy |
| Mission 2: Enter, Pursued by a Memory | Description | Start Dialogue | Difficulty |
| Follow Javier (Fig. 13a) | Arthur rides his horse following Javier up the mountain through the blizzard searching for John's trail. | Follow Javier | Hard |
| Search John (Fig. 13b) | After dismounting, Arthur followed Javier over slopes and ledges to find John and carry him away. | Javier: Down this way | Hard |
| Keep Wolves away (Fig. 13c) | Arthur manages to shoot all of the wolves before they can attack Javier and John. | Keep the wolves away from Javier and John | Hard |
| Kill Wolves (Fig. 13d) | Three people ride horses down the mountain. Arthur eliminate the wolves, protecting Javier and John ahead. | Javier: Come on, let's get back to the others | Hard |
| Return to Camp (Fig. 13e) | Arthur followed Javier on horseback back to camp. | Yea...c'mon. Let's push hard and get back | Easy |

action should continue to be executed. Furthermore, self-reflection can be leveraged to dissect why the last action failed to bring the agent close to the target task completion, better understand the factors that led to the successful completion of the preceding task, and so on.

Besides, we observe that instead of providing GPT-4o with sequential high-resolution images for self-reflection, low-resolution images make it easier for GPT-4o to understand the relation among the sequential screenshots and capture dynamic changes, resulting in a significantly higher success rate of detecting whether the action is executed successfully and take any effect. We hypothesize that since a high-resolution image can cost as many as 2000 tokens, too many high-resolution images make GPT-4o fail to capture the overall changes across screenshots and be caught up in the local details.

**Task Inference.** During gameplay, we let GPT-4o propose the current task to perform whenever it believes it is time to start a new task. GPT-4o also outputs whether the task is a long- or short-horizon task when proposing a new task. Long-horizon tasks, such as traveling to a location, typically require multiple iterations, whereas short-horizon tasks, like picking up an item or conversing with someone, involve fewer iterations. The agent will follow the newly generated task for the next 3 interactions. After 3 interactions, the agent returns to the last long-horizon task in the stack. Deciding on a binary task horizon is much easier and more robust for GPT-4o, than re-planning at every iteration. Since a long-horizon task frequently includes multiple short-horizon sub-tasks, this implementation also helps avoid forgetting the long-horizon tasks under execution.

**Skill Curation.** As shown in Figure 17, during gameplay, instructions often appear on the screen, such as "press [Q] to take

Table 8: Key points in the open-ended mission, *Buy Supply* in RDR2. Figure 14 showcases snapshots of key points (specific sub-figures marked in parenthesis in the table).

| Mission 3: Buy Supply | Description |
| --- | --- |
| Find Horse (Fig. 14a) | Find and mount the horse in the camp. |
| Prepare to Navigate to Saloon (Fig. 14b) | Open map, find the saloon and create waypoint. |
| Go to Saloon (Fig. 14c) | Ride horse to the saloon. |
| Prepare to Navigate to Shop (Fig. 14d) | Open map, find the general store and create waypoint. |
| Go to Shop (Fig. 14e) | Ride horse to the shop. |
| Enter Shop (Fig. 14f) | Dismount the horse and enter the shop. |
| Talk to Shopkeeper(Fig. 14g) | Approach the shopkeeper and talk. |
| Buy Target Product (Fig. 14h) | Open the menu, find and buy the target product. |

over" and "hold [TAB] to view your stored weapons", which serve as essential directives for completing current and future tasks proficiently. To save interactions with GPT-4o, we implement a simple version of this module inside *Information Gathering* to reduce interactions with GPT-4o. When GPT-4o detects and classifies some instructional text in the recent observation, which usually contains key and button hints, it will directly generate the corresponding code and description.

**Action Planning.** Upon execution of this module, we first retrieve the top k relevant skills for the task from procedural memory, alongside the newly generated skills. We then provide GPT-4o with the current task, the set of retrieved skills, and other information collected in *Information Gathering* that may be helpful for decision-making (*e.g.*, recent screenshots with corresponding descriptions, previous decisions, and examples) and let it suggest which skills should be executed. We also request that GPT-4o provide the reasons for choosing these skills, which increases the accuracy, stability, and explainability of skill selection and thus greatly improves framework performance. While GPT-4o sometimes may generate a sequence of actions, we currently only execute the first one, and perform *Self-Reflection*, since we observe a tendency for the second action to usually suffer from severe hallucinations.

**Action Execution**. Unlike the conventional mouse operation in standard software, where the cursor is restricted to a 2D grid and remains visible on the screen to navigate and interact with elements, the utilization of the mouse in 3D games like RDR2 introduces a varied control scheme. In menu screens, the mouse behaves traditionally, offering familiar point-and-click functionality. However, during gameplay, the mouse cursor disappears, requiring players to move the mouse according to specific action semantics. For example, to alter the character's viewpoint, the player needs to map the actual mouse movement to in-game direction angle changes, which differ in magnitude in the X and Y axes. Another special transition applies to shooting mode, where the front sight is fixed at the center of the screen, and players must maneuver the mouse to align the sight with target enemies. This nuanced approach to mouse control in different contexts adds an extra layer of challenge to general computer handling, showcasing the adaptability required in game environments, compared to regular software applications.

**Procedural Memory**. In our target setting, We intend to let the agent learn all skills from scratch, to the extent possible for the main storyline missions. The procedural memory is initialized with only preliminary skills for basic movement, which are not clearly provided by the in-game tutorial and guidance.

- *turn(degree)*, *move_forward(duration)*: Since the game does not precisely introduce how to move in the world through in-game instructions, we provide these two basic actions in advance, so GPT-4o can perform basic mobility, while greatly reducing the number of calls to the model.

- *shoot(x, y)*: RDR2 also does not provide detailed instructions on how to aim and shoot. Moreover, due to limitations with GPT-4o spatial reasoning and the need to sometimes augment images with object bounding boxes, we provide such basic skill for the agent to complete relevant tasks.

- *select_item_at(x, y)*: Similarly to *shoot()*, due to the lack of instructions, we provide such skill for the agent to move the mouse to a certain place to select a given item.

Beyond these basic atomic low-level actions, we introduce a few composite skills to facilitate the game playing progress.

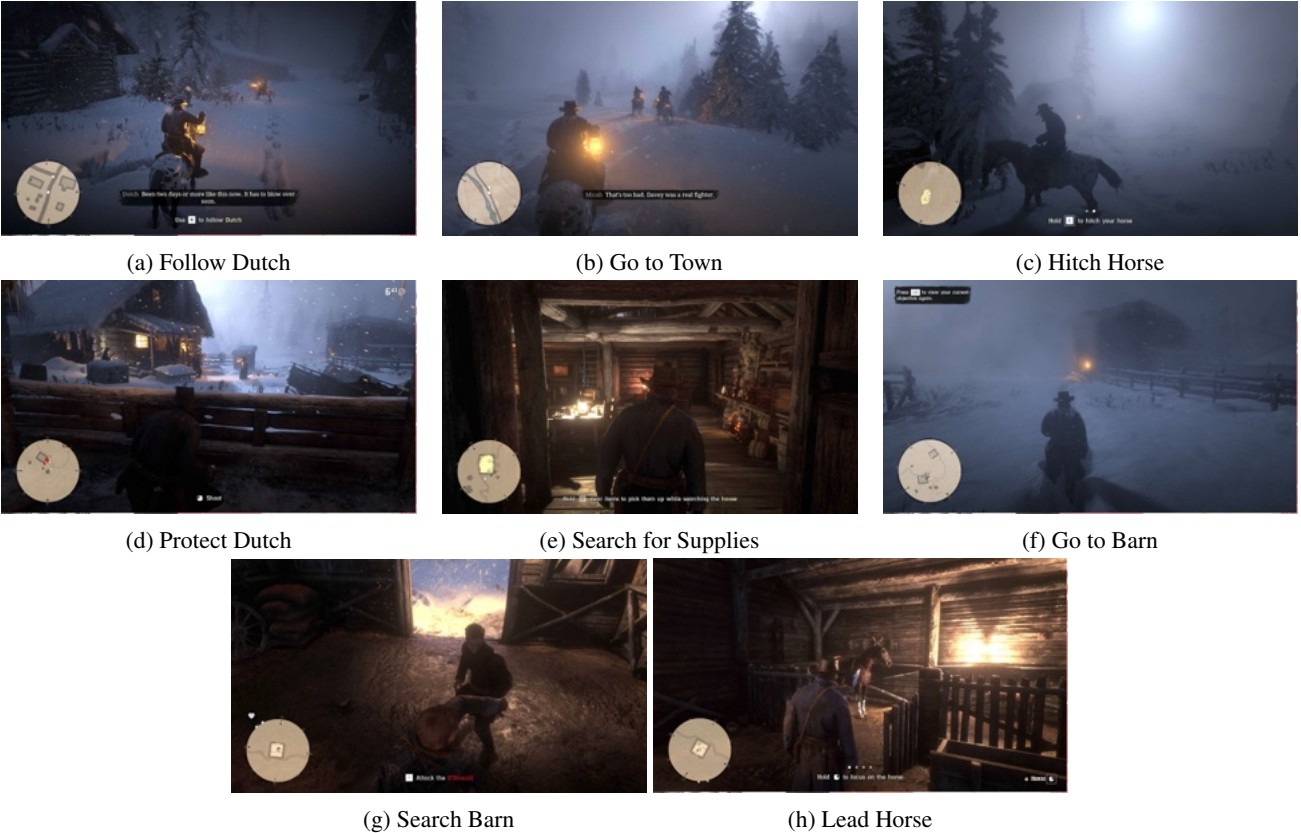

(a) Follow Dutch       (b) Go to Town       (c) Hitch Horse

(d) Protect Dutch       (e) Search for Supplies       (f) Go to Barn

(g) Search Barn       (h) Lead Horse

Figure 12: Image examples of tasks in the first mission of *Outlaws from the West*. (The picture has been brightened for easier reading.)

The agent should be able to complete tasks using only the basic skills above and the skills it learns, but these composite skills streamline the process by greatly reducing calls to the backend model.

- *turn_and_move_forward(degree, duration)*: This skill is just a simple composition of *turn()* and *move_forward()* to save frequent calls to GPT-4o in a common sequence.

- *follow(duration)* and *navigate_path(duration)*: In RDR2, tasks often guide players to follow NPCs or generated paths (red lines) in the minimap to certain locations. This can be reliably accomplished via the basic movement skills, but requires numerous interactions with GPT-4o. To control both cost and time budgets involving GPT-4o's responses, we leverage the information shown in the minimap to implement a composite skill to follow target NPCs or red lines for a short set of game iterations. The default duration is 20 iterations. Increasing the duration can dramatically improve the performance in task *Follow Dutch*, *Follow Javier* and *Killing Wolves* but significantly decrease the success rate of *Search John* since this task requires frequent exchange of the skills between climbing and following.

- *fight()*: As output of an interaction with GPT-4o, the agent will only take one action per step. However, though the action is generated correctly, specifically in fight scenarios, the action frequency may not be high enough to defeat an opponent. In order to allow sub-second punches, we provide a pre-defined action that wraps this multi-action punching, which can be selected by GPT-4o to effectively win fights.

For the open-ended mission, since the agent skips all the tutorials in Chapter I, we provide all the necessary skills in the procedural memory at the beginning of the mission.

**Episodic Memory.** This module stores all the useful information, *e.g.,* input and output of GPT-4o. In each iteration, after the self-reflection, we will request GPT-4o to summary the event that happened in the last action and the past experiences.

**Game Pause.** To prevent in-game time from passing in real-time games like RDR2, we have to pause the game while waiting for LMMs' response. The time interval between two consecutive actions can be as long as one minute. In RDR2,

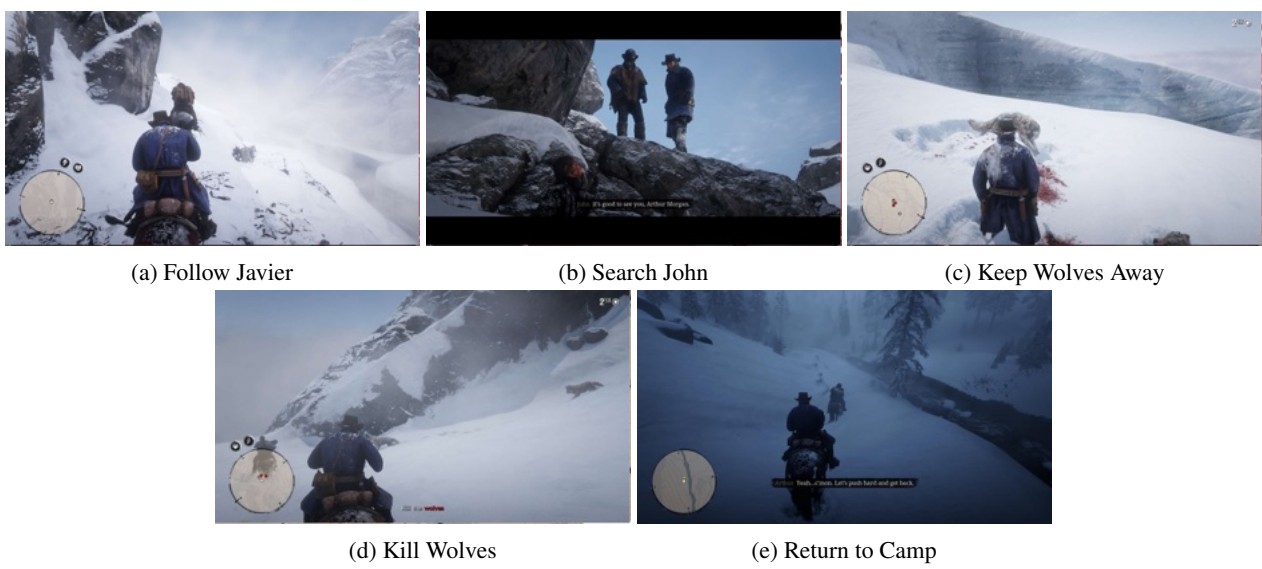

(a) Follow Javier      (b) Search John      (c) Keep Wolves Away

(d) Kill Wolves      (e) Return to Camp

Figure 13: Image examples of tasks in the second mission of *Enter, Pursued by a Memory*.

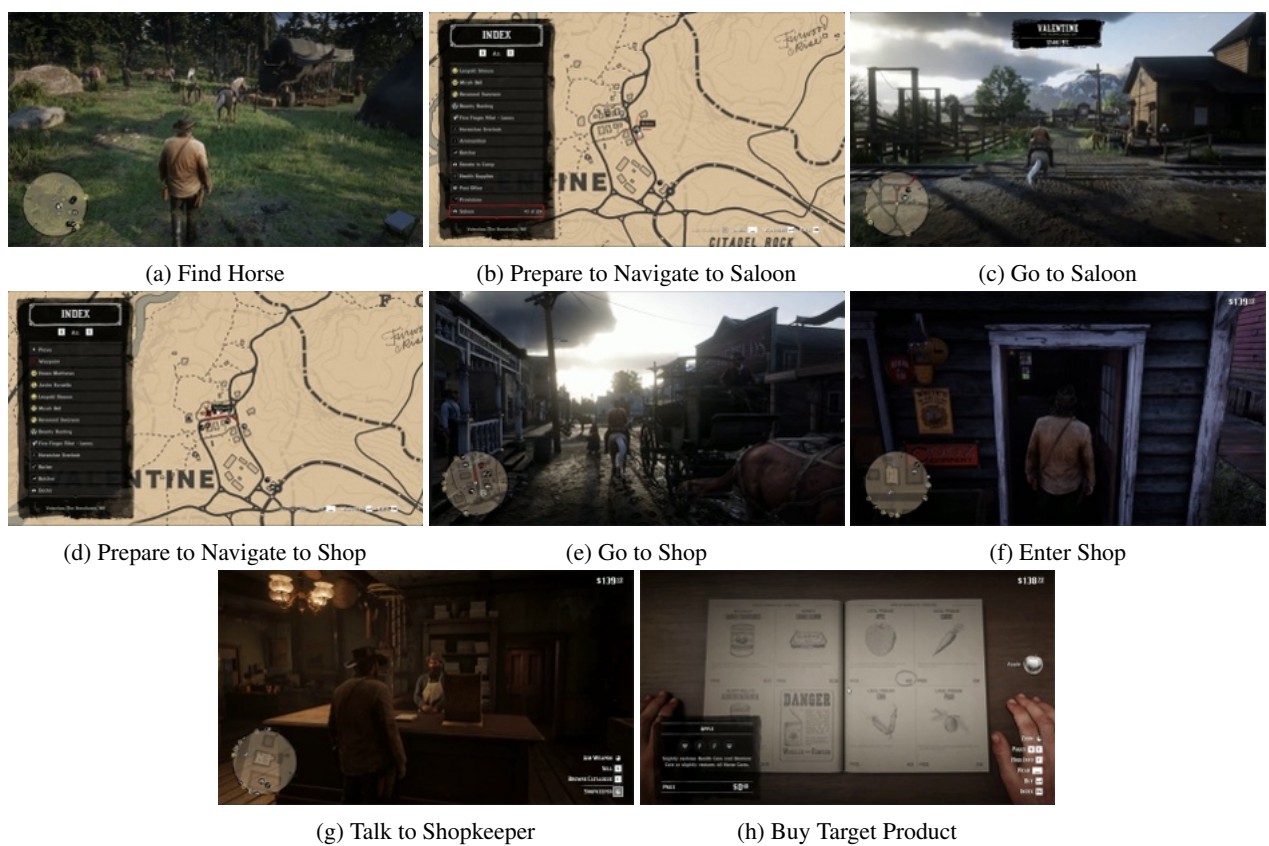

(a) Find Horse      (b) Prepare to Navigate to Saloon      (c) Go to Saloon

(d) Prepare to Navigate to Shop      (e) Go to Shop      (f) Enter Shop

(g) Talk to Shopkeeper      (h) Buy Target Product

Figure 14: Image examples of key points in the open-ended task of *Buy Supply*.

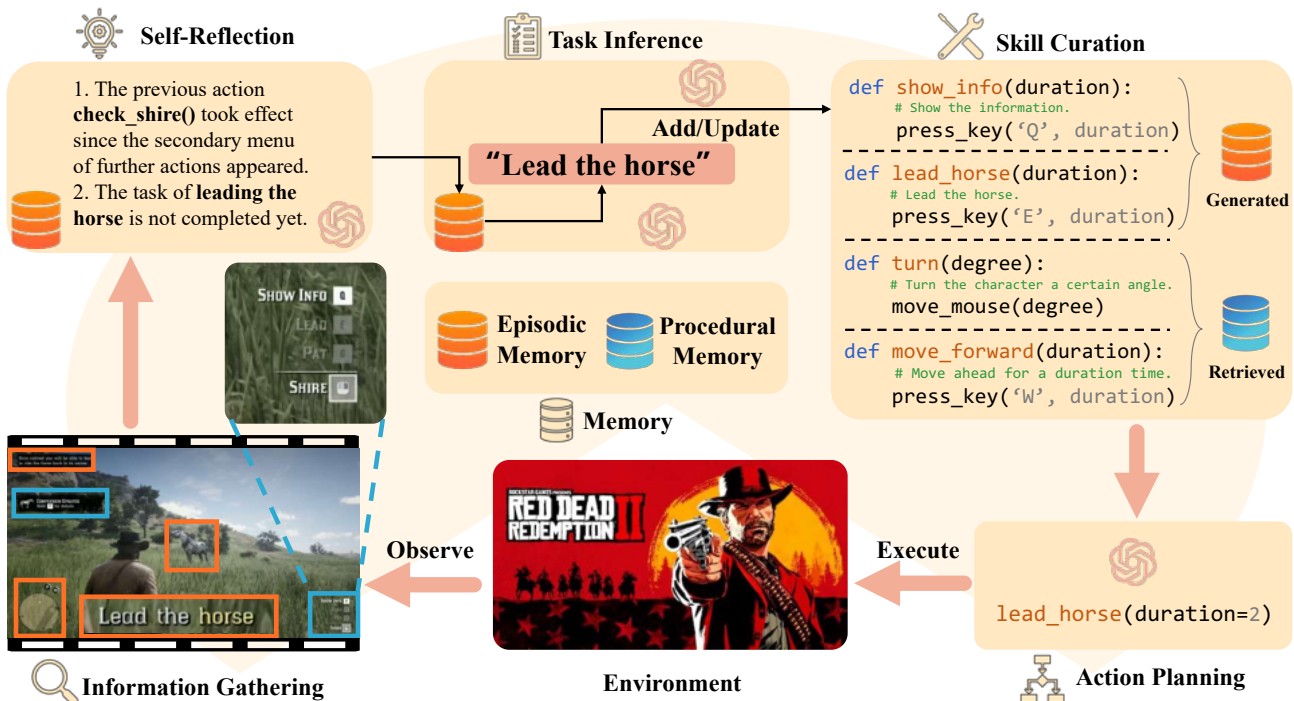

Figure 15: The detailed illustration of how CRADLE is instantiated as a game agent to play RDR2.

after the agent finishes executing outputted actions, *esc* will be automatically pressed to pause the game and when the agent determines the next action, *esc* will be automatically pressed again to unpause the game. Note that there will be an animation lasting up to 0.5 seconds for both pausing and unpausing. During this animation, we can not control the character, but the dynamics of the game world keep changing, *e.g.,* the wolves are still moving. It introduces additional challenges for the tasks that require precise timing, like combat.

## F.4. Case Studies

Here we present a few game-specific case studies for more in-depth discussion of the framework capabilities and the challenges of the GCC setting.

### F.4.1. SELF-REFLECTION

Self-reflection is an essential component in CRADLE as it allows our framework reasoning to correct previous mistakes or address ineffective actions taken in-game. Figure 16 provides an example of the self-reflection module. The task requires the agent to select a weapon to equip, in the context of the "Protect Dutch" task. Initially, the agent selects a knife as its weapon by chance, but since the game requires a gun to be chosen, this is incorrect and the game still prompts the player to re-open the weapon wheel. The self-reflection module is able to determine that the previous action was incorrect and on a subsequent iteration the agent successfully opts for the gun, correctly fulfilling the task requirement and advancing to the next stage in the story.

### F.4.2. SKILL CURATION

For skill curation, we first provide GPT-4o with examples of general mouse and keyboard control APIs, *e.g.*, io_env.key_press and io_env.mouse_click. Figure 17 shows that GPT-4o can capture and understand the prompts appearing on screenshots, *i.e.,* icons and text, and strictly follow the provided skill examples using our IO interface to generate correct skill code. Moreover, GPT-4o also generates comments in the code to demonstrate the functionality of this skill, which are essential for computing similarity and relevance with a given task during skill retrieval. The quality of the generated comment directly determines the results of skill retrieval, and further impacts reasoning to action planning. Curation can also re-generate code for a given skill, which is useful if GPT-4o wrongly recognized a key or mouse button in a previous iteration.

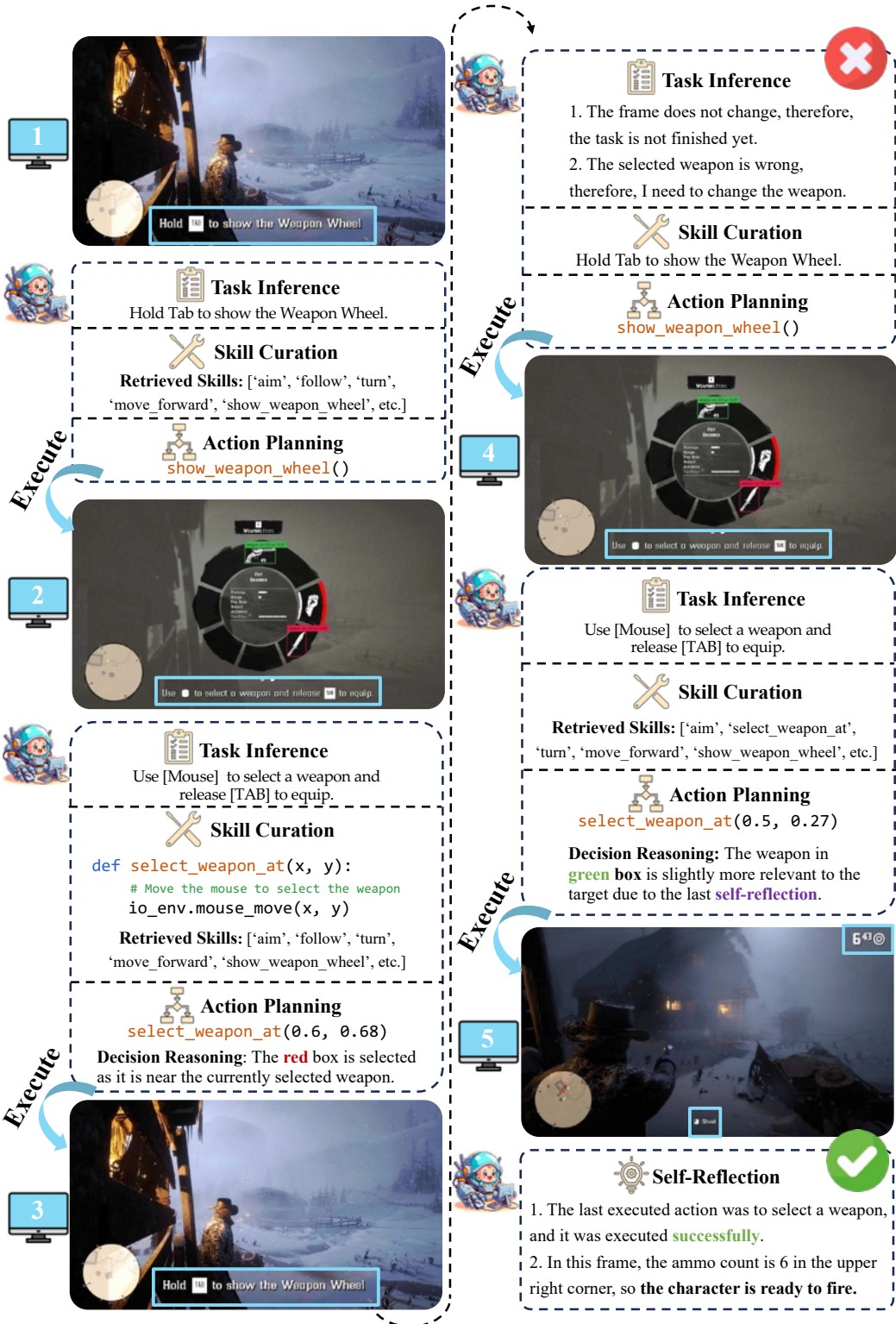

Figure 16: Case study of self-reflection on re-trying a failed task. Task instruction and context require the agent to equip the gun. A wrong weapon (knife) is first selected, but the agent equips the gun after self-reflection. Only relevant modules are shown for better readability, though all modules (Figure 4) are executed per iteration.

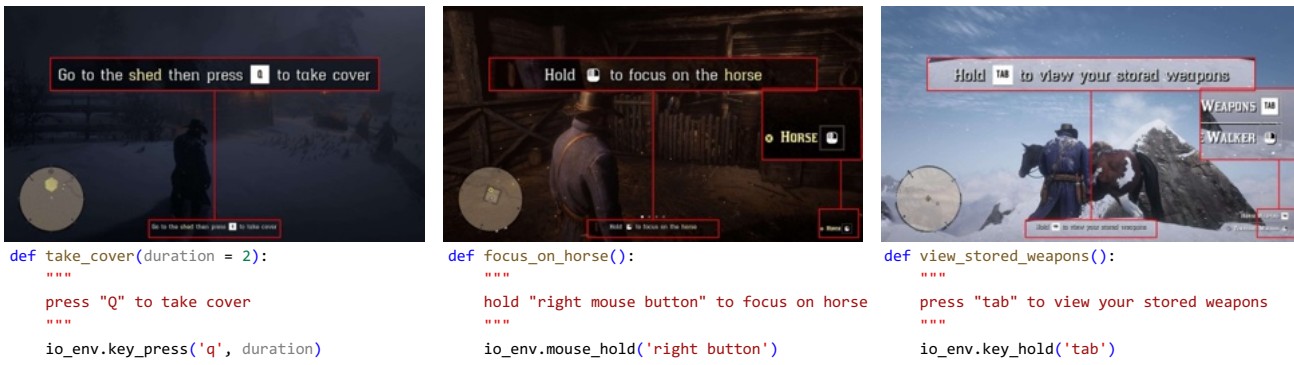

```
def take_cover(duration = 2):
    """
    press "Q" to take cover
    """
    io_env.key_press('q', duration)
```

```
def focus_on_horse():
    """
    hold "right mouse button" to focus on horse
    """
    io_env.mouse_hold('right button')
```

```
def view_stored_weapons():
    """
    press "tab" to view your stored weapons
    """
    io_env.key_hold('tab')
```

Figure 17: Skill code generation based on in-game instructions. As the storyline progresses, the game will continually provide prompts on how to use a new skill via keystrokes or utilizing the mouse.

### F.4.3. ACTION EXECUTION AND FEEDBACK

Proper reasoning about environment feedback is critical due to the generality of the GCC setting and the level of abstraction to interact with the complex game world. The semantic gaps between the execution of an action, its effects in the game world, and observing the relevant outcomes for further reasoning lead to several potential issues that CRADLE needs to deal with. Such issues can be categorized into four major cases:

**Lack of grounding feedback.** In many situations, due to the lack of precise information from the environment, it can be difficult for the system to deduce the applicability or outcome of a given action. For example, when picking an item from the floor, the action may fail due to the distance to the object not yet being close enough. Or, if within pick up range, the chosen action may not exactly apply due to other factors (*e.g.,* character's package is full).

Even if the right action is selected and executed successfully, the agent still needs to figure out its results from the partial visual observation of the game world. If the agent needs to pick or manipulate an object that is occluded from view, the action may execute correctly, but no outcome can be seen.

A representative example in RDR2 happens when the agent tries to pick up its gun from the floor after a fight. Getting to the right distance, without completely occluding the object, can lead to multiple re-trials. Figure 18a showcases a situation where, though the character is already standing near the gun (as seen in the minimap), it's still not possible to pick it up.

Previous efforts (Wang et al., 2023b; 2024a) that utilize in-game state APIs unreasonably bypass such issues by leveraging internal structured information from the game and the full semantics of responses (data) or failures (error messages).

**Imprecise timing in IO-level calls.** This issue is caused by the ambiguity in the game instructions or differences in specific in-game action behaviors, where even the execution of a correct action may fail due to minor timing mismatches. For example, when executing an action like 'open cabinet', which requires pressing the [R] key on the keyboard, if the press is too fast, no effect happens in the game world. However, as there is no visual change in the game nor other forms of feedback, it can be difficult for GPT-4o to figure out if an inappropriate action was chosen at this game state or if the minor timing factor was the problem. Pressing the key for longer triggers an animation around the button (only if the helper menu is on screen), but this is easily missed and any key release before the circle completes also results in no effect. Figure 18b illustrates the situation.

The same problem also manifests in other situations in the game, where pressing the same key for longer triggers a completely different action (*e.g.,* lightly pressing the [Left Alt] key vs. holding it for longer).

**Change in the semantics of key and button.** A somewhat similar situation occurs when the same keyboard key or mouse button gets attributed different semantics in different situations (or even in a multi-step action). GPT-4o may decide to execute a given skill, but the original semantics no longer hold. The lack of in-game effect parallels the previous situations. Worse yet, an undesired effect will confuse the system regarding the correct action being selected or not.

For example, when approaching a farm in the beginning of the game, the agent needs to hitch the horse to a pole to continue. The operation to perform the action consists of pressing the [E] key near a hitching post (as shown in Figure 18c). However, the same [E] key press is the only constituting step in other actions with different semantics, like *dismount the horse* or *open*

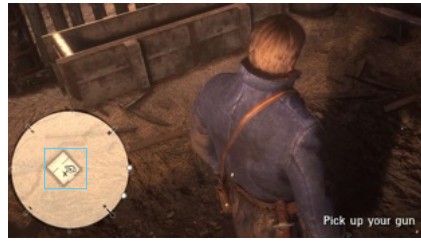 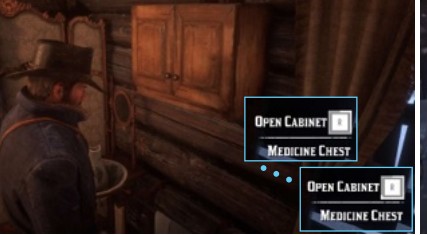 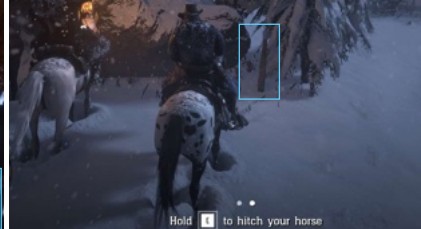

(a) 'Pick gun' unavailable       (b) 'Open cabinet' press timing       (c) 'Hitch horse' re-use of [E] key

Figure 18: Examples of action execution uncertainty. Lack of environmental feedback to actions and semantic gaps between action intent and game command can lead to challenging situations for agent reasoning.

*the door.* Wrongly triggering a horse dismount at the situation shown in the figure can lead to undesired side effects, *i.e.,* it may mislead the system about the actual effects of the action or affect the planning of which next actions to perform.

**Interference issues.** Lastly, completion of some actions requires the correct execution of multiple steps sequentially, which could be interrupted in many ways not related to the agent's own actions. Without the use of APIs that expose internal states or other forms of feedback, it is much harder for the agent to decide when to repeat sub-actions or try different strategies. For example, if the agents gets shot and loses aim while in combat, or an unrelated in-game animation is triggered mid-action, canceling it.

Since there is no direct environment feedback, the agent needs to carefully analyze the situation and try to infer if any action step needs re-execution.

### F.5. Limitations of GPT-4o and GPT-4V

Deploying CRADLE in a complex game like RDR2 requires the backbone LMM model to handle multimodal input, which revealed several limitations of both GPT-4V and GPT-4o, necessitating external tools to enhance overall framework performance. Initial tests and exploration were performed using GPT-4V, as GPT-4o was not yet available. These tests highlighted significant weaknesses in spatial perception, icon understanding, history processing, and world understanding. Upon the release of GPT-4o, further testing demonstrated some notable improvements in spatial perception. However, enhancements in other areas remained marginal, while some regressions were also observed, all indicating the need for additional tools to aid decision-making.

**Spatial Perception.** As shown in Figure 19a and 20a, GPT-4V's spatial-visual recognition capability is insufficient for precise fine-grained control, particularly in detecting whether the character is being or going to be blocked and in estimating the accurate relative positions of target objects. In contrast, GPT-4o exhibits a significant enhancement in spatial perception, capable of recognizing obstacles ahead and estimating the approximate relative positions between objects. However, both models require supplementary information, such as bounding boxes of potential target objects, to make fine-grained decisions. These led to the need to augment certain images to provide auxiliary visual clues for decision-making, *i.e.,* bounding boxes of possible target objects.

**Icon Understanding.** Both GPT-4o and GPT-4V struggle with domain-specific concepts, such as unique icons within the game, which may represent specific targets or refer to certain mouse and key actions. As shown in Figure 19b and 20b, GPT-4V and GPT-4o fail to recognize the left shift, right mouse button, and space icons. Attempts to incorporate few-shot learning to improve image understanding cannot be generalized. Therefore, we match prepared pattern templates, *e.g.,* icon images, against each screenshot to continuously detect and highlight any appearing icons.

Figure 19c and Figure 20c also demonstrate that although GPT-4o performs better than GPT-4V in understanding the mini-map, it still fails to consistently interpret this crucial information regarding the position and direction of the character. This failure in localization leads the agent to sometimes get lost in the town and miss the task target. While the aforementioned issues can be slightly alleviated by providing additional few-shot examples, a significant improvement is only achieved by cropping the image and providing GPT-4o with the exact region containing the icon to be recognized. This dependency on precise input makes the issue challenging and unreliable for decision-making. Although the above issues can be slightly alleviated by providing additional few-shot examples, it can only have an obvious effect if we crop the image and provide the GPT-4o with the region exactly containing the icon to be recognized, which makes the issue intractable.

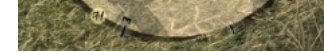
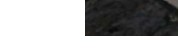

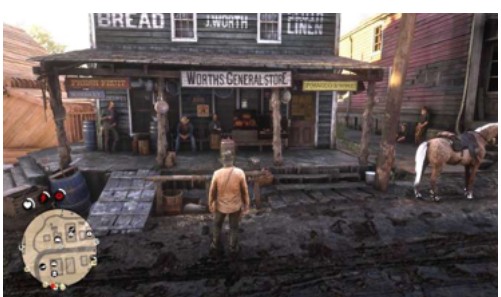
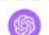
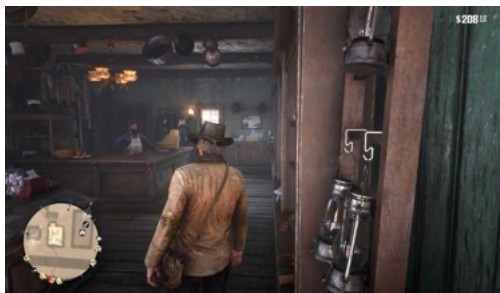
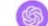

ChatGPT

There are **no immediate obstacles** directly in front of the character in the image. The character has a **clear path ahead** on a muddy street in what appears to be a Western-style town setting.

ChatGPT

The shopkeeper in the image is on the **right side**, behind the counter.

(a) GPT-4V fails to recognize obstacles in the environment and the position of the shopkeeper.

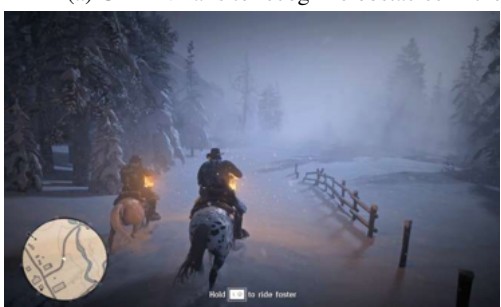
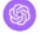
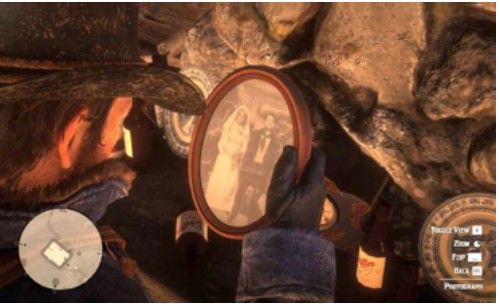
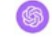

ChatGPT

Based on the provided game screenshot, I've identified the following control prompt:
- "**Hold L** ⬆️ to ride faster"
This control suggests that holding a certain key (which seems to be an **upward arrow key**, often associated with the **'W' key** for forward movement in many games) will make the character ride faster.

ChatGPT

From the provided image screenshot from the game, I can identify the following keyboard and mouse control prompts:
1. **"Toggle View"** - The key for this action is **"V"**.
2. **"Zoom"** - The key for this action is **"O"**.
3. **"Flip"** - The key for this action is **"L"**.
4. **"Back"** - The key for this action is **"ESC"**.

(b) GPT-4V struggles to recognize the icons for keys on keyboard and mouse buttons.

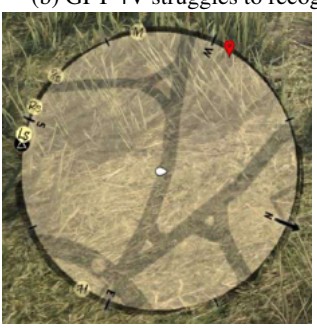
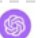
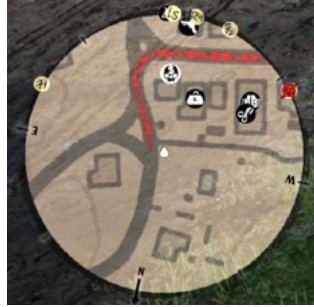
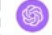

ChatGPT

The central white arrow is facing towards the **left**.

ChatGPT

The central white arrow with the white arrowhead is facing towards the **left**.

(c) GPT-4V cannot understand the correct direction of arrow points (character heading) in the mini-map.

Figure 19: Example situations of GPT-4V's limitations in understanding visual information from the game.

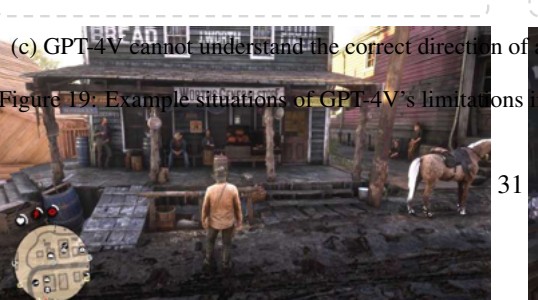
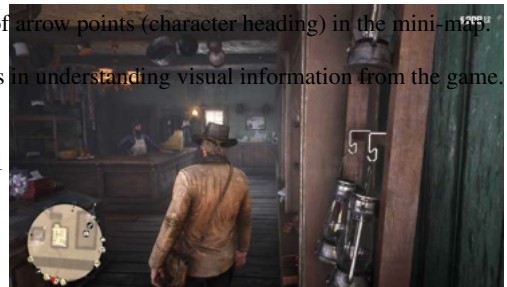

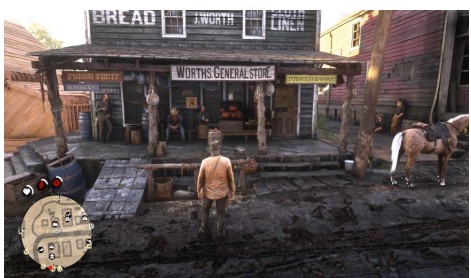
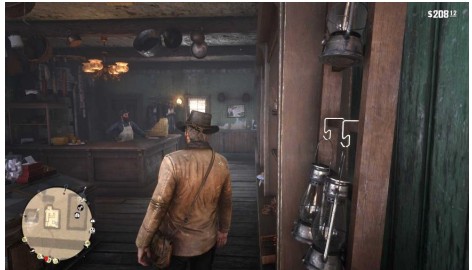

**ChatGPT**

There is **an obstacle** in front of the character. It appears to be a small wooden railing or barrier along the edge of the platform. The character would need to navigate around it or find a way over it to proceed forward.

**ChatGPT**

The shopkeeper in the image is on the **left** side

(a) GPT-4o can recognize obstacles in the environment and the position of the shopkeeper.

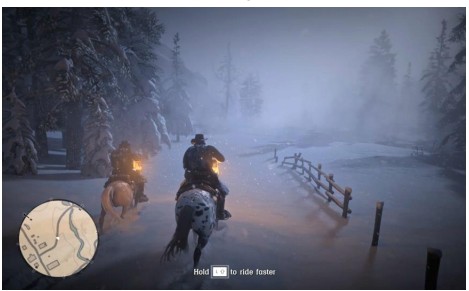
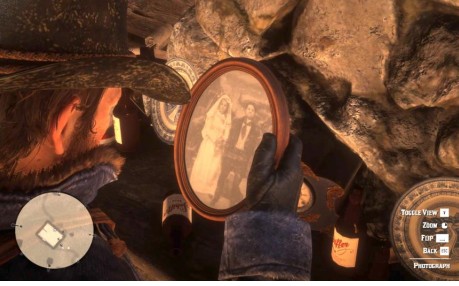

**ChatGPT**

Based on the provided image, the control prompt shown is:
• "Hold **L** to ride faster'.
The '**L**' refers to the left analog stick on a **controller**, indicating that the player should hold the left analog stick in a forward position to increase the speed of their character's movement, typically while riding a horse.

**ChatGPT**

From the provided image screenshot from the game, I can identify the following keyboard and mouse control prompts:
1. Toggle View: **V**
2. Zoom: **Mouse Scroll Wheel**
3. Flip: **R**
4. Back: **Esc**
5. Photograph: (No specific key assigned, just a label)

(b) GPT-4o struggles to recognize the icons for keys on keyboard and mouse buttons.

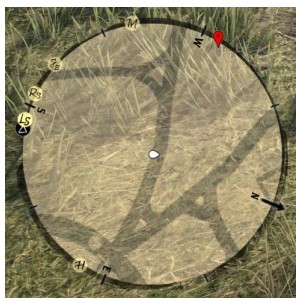
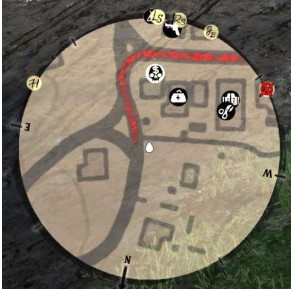

**ChatGPT**

The central white arrow in the mini-map is facing to the **right**.

**ChatGPT**

The central white arrow on the mini-map is facing **down**.

(c) GPT-4o occasionally accurately determines the direction of arrow points (character heading) in the mini-map.

Figure 20: Example situations of GPT-4o's capabilities in understanding visual information from the game.

**History Processing.** Moreover, both GPT-4o and GPT-4V can easily get distracted by irrelevant information in longer contexts, resulting in hallucinations. For example, when action planning utilizes too many historical screenshots, they may confuse past and present frames. Additionally, performance fluctuates and both model versions frequently generate output not adhering to the rules in the provided prompts. To mitigate the issue of hallucinations, we more strictly control input information by further summarizing long-term memory.

**World Understanding.** Lastly, the absence of an RDR2 world model limits GPT-4V and GPT-4o's understanding of the consequences of its actions in the game. This often results in inappropriate action selection, such as overestimating the necessary adjustments for aligning targets or misjudging the duration required for certain actions. To alleviate this problem, we introduced extra prompt rules regarding action parameters and more flexibility into the self-reflection module.

## G. Stardew Valley

### G.1. Introduction to Stardew Valley

Stardew Valley is an open-ended country-life RPG game developed by ConcernedApe, which has a 98% positive rating on Steam and is rated as Overwhelmingly Positive. Players take on the role of a character disillusioned with city life who inherits a dilapidated farm from their late grandfather. Initially, the farmland is overrun with boulders, trees, stumps, and weeds, which players must clear to make way for crops, buildings, and placeable items. The main goal is to restore and expand the farm through activities such as planting crops, raising animals, mining, fishing, and crafting. Additionally, players can interact with NPCs in town, forming relationships that can lead to marriage and children. Players complete quests for money or to restore the town's Community Center by completing "bundles," which reward items like seeds and tools and unlock new areas and game mechanics. All activities are balanced against the character's health, energy, and the game's clock. Food provides buffs, health, and energy. The game features a simplified calendar with four 28-day months representing each season, affecting crop growth and activities. Compared to RDR2, this game is more lightweight and easy to control. This game features a wealth of production and social activities, presenting a comprehensive test of an agent's abilities, which is an ideal platform to observe and evaluate agents' comprehensive behaviors and abilities, like in the Generative Agents (Park et al., 2023). We use the latest version (1.6.8) of the game to conduct all the experiments.

### G.2. Objectives

We find that GPT-4o surprisingly struggles with accurately recognizing and locating objects near the player in this 2D game. This leads to difficulties for the agent to interact with objects or people, as it requires the player to stand precisely in front of them in the grid (*e.g.,* when entering doors, using a pickaxe to break stones). Even some basic tasks are already challenging enough for current agents in this game. Therefore, as shown in Figure 21, we evaluate three essential tasks in the early stages of the game:

- **Farm Clearup.** Clear the obstacles on the farm, such as weeds, stones, and trees, as much as possible to prepare for farming. This task requires agents to move precisely to be in front of the obstacles, identify the type of obstacles correctly and select corresponding tools to deal with them.

- **Cultivation.** Use the hoe to till the soil, use a parsnip seed packet on the tilled soil to sow a crop, water the crop every day and harvest at least one parsnip. This task requires long-horizontal memory and reasoning.

- **Shopping.** Go to the general store in the town, which is on the other map, to buy more seeds and return home. This task is used to evaluate agents' long-distance navigation ability.

For each task, the maximal steps is 100.

### G.3. Implementation Details

**Visual Prompting.** As a cartoon-style pixel game, the game screen of Stardew is quite different from the real world. Although GPT-4o can observe coarse-grained information from screenshots, more fine-grained information is required to complete tasks. Therefore, as shown in Figure 22, we divide each screenshot into $3 \times 5$ grids and require GPT-4o to describe the screenshot in a grid-by-grid format. We empirically find that it can result in a more precise and accurate description. And GPT-4o can also make better control based on the grids. In addition, we also augment the image with two blue and yellow

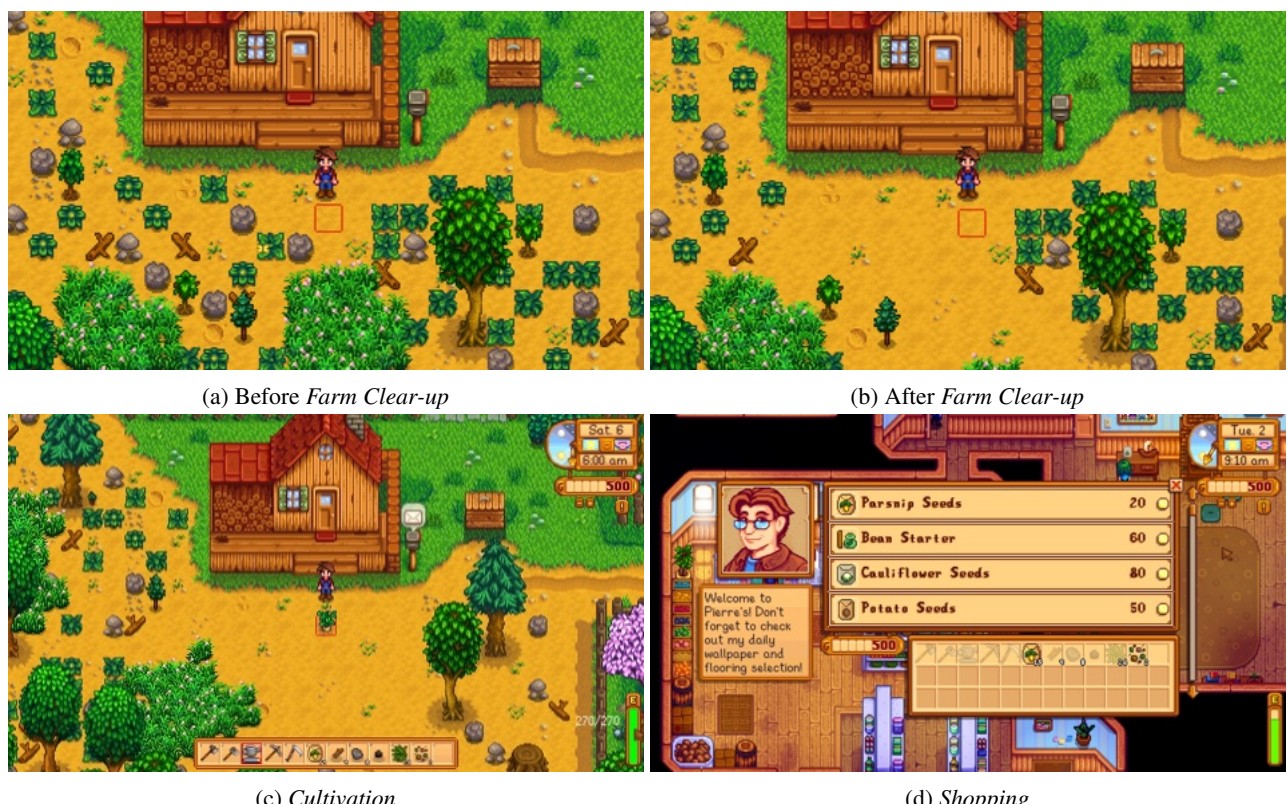

(a) Before *Farm Clear-up*

(b) After *Farm Clear-up*

(c) *Cultivation*

(d) *Shopping*

Figure 21: Three tasks in Stardew Valley.

bands on the left and right sides, respectfully, with the prompt, "The blue band represents the left side and the yellow band represents the right side". Our empirical results show that this method significantly improves GPT-4o's ability to accurately distinguish left from right.

**Information Gathering.** As mentioned in the introduction of visual prompting, we let GPT-4o describe the image grid by grid, which is helpful in locating the position of the character, surrounding objects and buildings and facilitates the understanding of the relative positions among them for GPT-4o. Besides, while compared to GPT-4V, GPT-4o is able to recognize most of the icons and their quality in the toolbar shown at the bottom of the screenshot, GPT-4o cannot output the items in the inventory sequentially one by one as it always skips a few in between. We have to clip the box for each item out of the toolbar and feed them to GPT-4o independently, augmented with template matching, for recognition, which turns out to be more accurate. The success of recognition of the tools in the toolbar is critical to tasks like **Farm Clearup** and **Cultivation**.

**Self-Reflection.** The duration of actions in Stardew is usually much shorter than in RDR2, so we only use the first and last frame from the video observation to reduce the number of tokens used per request. Additionally, we provide some helpful prior information for GPT-4o. For example, a screenshot of the inside of the store is provided to check whether the store was successfully entered. This is useful because there are many other buildings near the store, and sometimes GPT-4o controls the character to enter the wrong one. However, this is not realized if the screenshot is not provided.

**Skill Curation.** For skill curation, as mentioned in Figure 5, we mainly rely on the in-game manual to generate atomic skills, like *move_up()*, *do_action()* and *use_tool()*. In addition, to handle the challenges of locating objects, especially doors, we have a special set of composite skills specifically for Stardew. *e.g.*, go_through_door, buy_item, get_out_of_house and enter_door_and_sleep. With the restrictions of GPT-4o in fine-grained control, we designed go_through_door composite skills for the agent to control the game character to accurately reach various doors and successfully enter, such as the house and the store door. and in order to buy certain items such as parsnip seeds, we designed the composite skills buy_item to control the game character to interact with the salesman and buy parsnip seeds. similarly, we designed the get_out_of_house and enter_door_and_sleep composite skills to accurately exit the house from the bed and enter the house and walk to the bed.

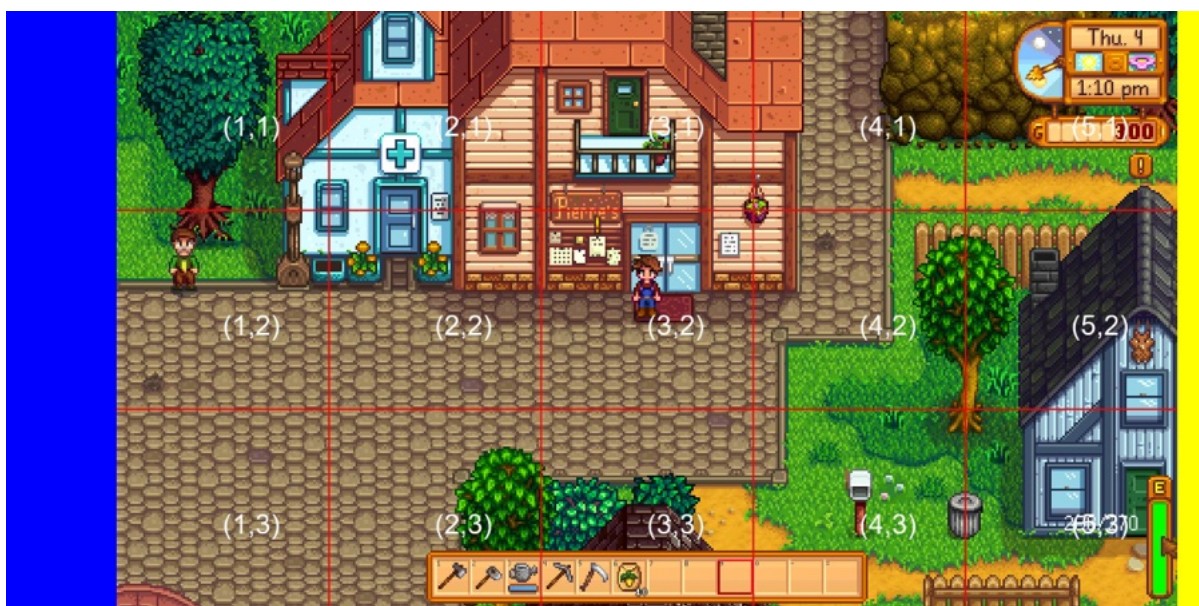

Figure 22: Augmented screenshot via visual prompting. The full screenshot is divided into $3 \times 5$ grids and each grid has a unique white coordinate. Additionally, we augment all input images with color bands, with the prompt, "The blue band represents the left side and the yellow band represents the right side", which significantly improves GPT-4o's ability to accurately distinguish left from right.

**Action Planning.** In this game, we let GPT-4o output at most two skills in a single action every time, which turns out to be efficient. The agent usually needs to select the correct tool first and then use the tool or do action.

**Procedure Memory.** Procedure Memory is used to store and retrieve skills in code form. In order for agents to quickly get started and complete some special tasks in Stardew, we have predefined skills in Procedure Memory. These skills are divided into atomic and composite skills. atomic skill consists of basic operations such as moving, selecting tools, etc. The description of all the atomic skills is listed as follows:

- *do_action()*: The function to perform a context-specific action on objects or characters.

- *use_tool()*: The function to execute an in-game action commonly assigned to using the character's current selected tool.

- *move_up(duration)*: The function to move the character upward (south) by pressing the 'w' key for the specified duration.

- *move_down(duration)*: The function to move the character downward (north) by pressing the 'w' key for the specified duration.

- *move_left(duration)*: The function to move the character left (west) by pressing the 'w' key for the specified duration.

- *move_right(duration)*: The function to move the character right (east) by pressing the 'w' key for the specified duration.

- *select_tool(key)*: The function to select a specific tool from the in-game toolbar based on the given tool number.

and the composite skills are designed for the agent to complete a variety of special tasks. The description of all the composite skills is listed as follows:

- *buy_item()*: The function to interact with the salesman and buy the item.

- *enter_door_and_sleep()*: The function to enter the house and walk to the bed.

- *get_out_of_house()*: The function to accurately exit the house from the bed

- *go_through_door( )*: The function to reach and enter all kinds of doors.

**Game Pause.** The game will pause automatically when the game window is not focused. So when the character finishes executing actions, we will activate another window, *e.g.,* code window, to pause the game and stop the passage of the time in the game.

### G.4. Case Studies

Here we present a few game-specific case studies to further discuss CRADLE's self-reflection and task-inference processes in the GCC setting.

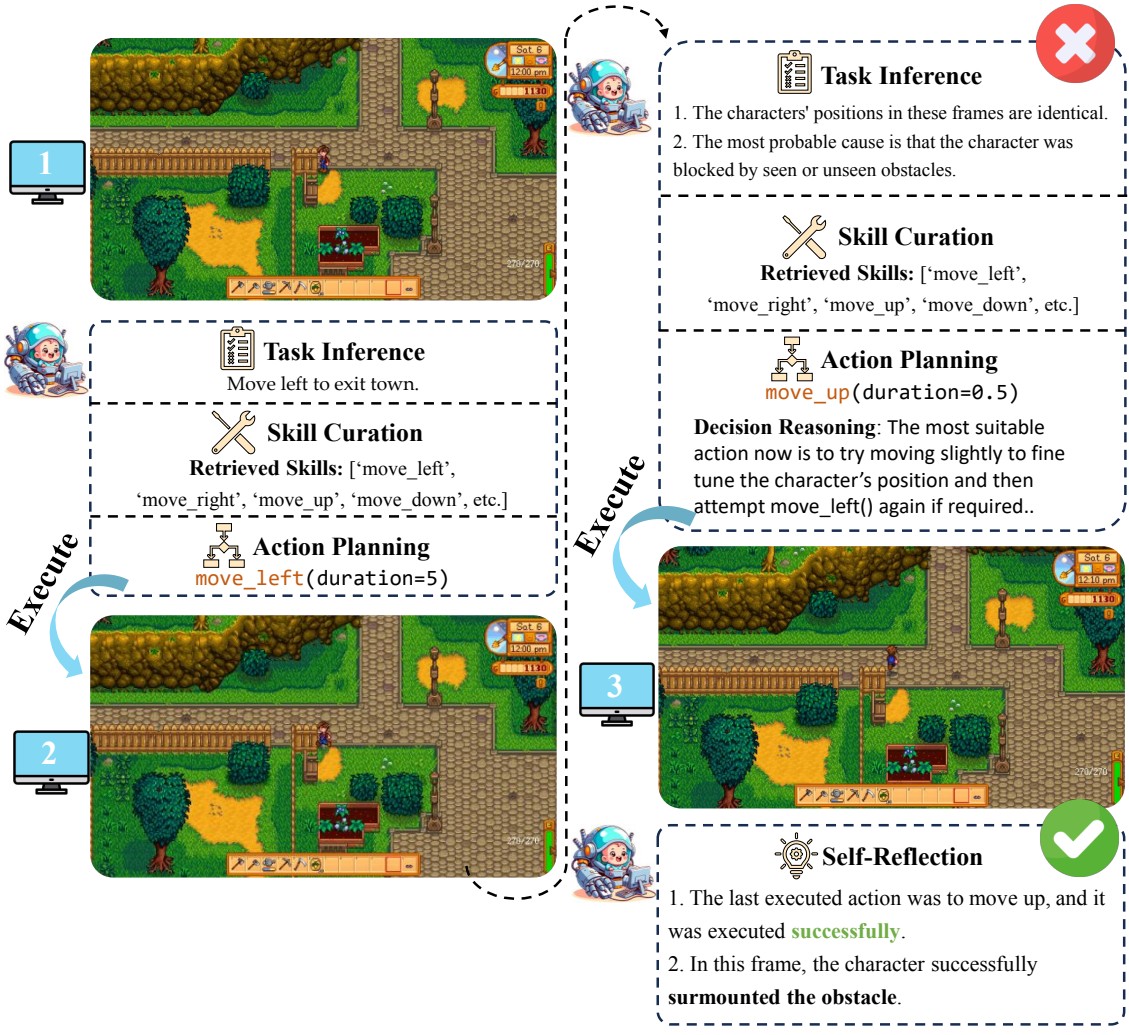

Figure 23: Case study of self-reflection on re-trying a failed task. Task instruction and context require the agent to exit town. A wrong direction is first selected, but the agent moves up after self-reflection. Only relevant modules are shown for better readability, though all modules (Figure 4) are executed per iteration.

### G.4.1. SELF-REFLECTION

The Self-reflection module plays an important role in the completion of game missions in Stardew, giving our framework the ability to determine if the actions performed are complete and effective and to correct the errors of invalid actions. In the "Purchasing Seeds" task, the Agent is asked to return home from the store after purchasing items. At the "Home is on

the left side of the store" prompt, the Agent controls the character to go left, but there are obstacles to keep going left, and the character must go up to circumnavigate the obstacles. As shown in the Fingure 23, the role will initially be stuck at the obstacle and cannot continue to the left. Through Self-Reflection, the Agent can judge that it is currently in a state of obstruction, and moving to the left cannot be implemented smoothly. Therefore, the agent can adjust the direction upward to bypass the obstacle and enable the role to continue to the left until it returns home.

### G.4.2. TASK-INFERENCE

Task Inference is a very effective module for completing game quests in Stardew. Its function is to decompose a vague and grand task into a specific sub-task, which effectively guides the Agent to complete the overall task. For example, in the Farming task, as shown in Figure 24, the task that the character needs to complete is "cultivate and harvest a parsnip." This is a complete but vague task. Through the Task Inference module, the Agent breaks down the task into (1) till the soil with the hoe, (2) plant the parsnip seeds, (3) water the planted seeds once daily for four days, (4) harvest the fully grown parsnip. This enables the Agent to know more clearly the steps needed to complete and finish the task successfully.

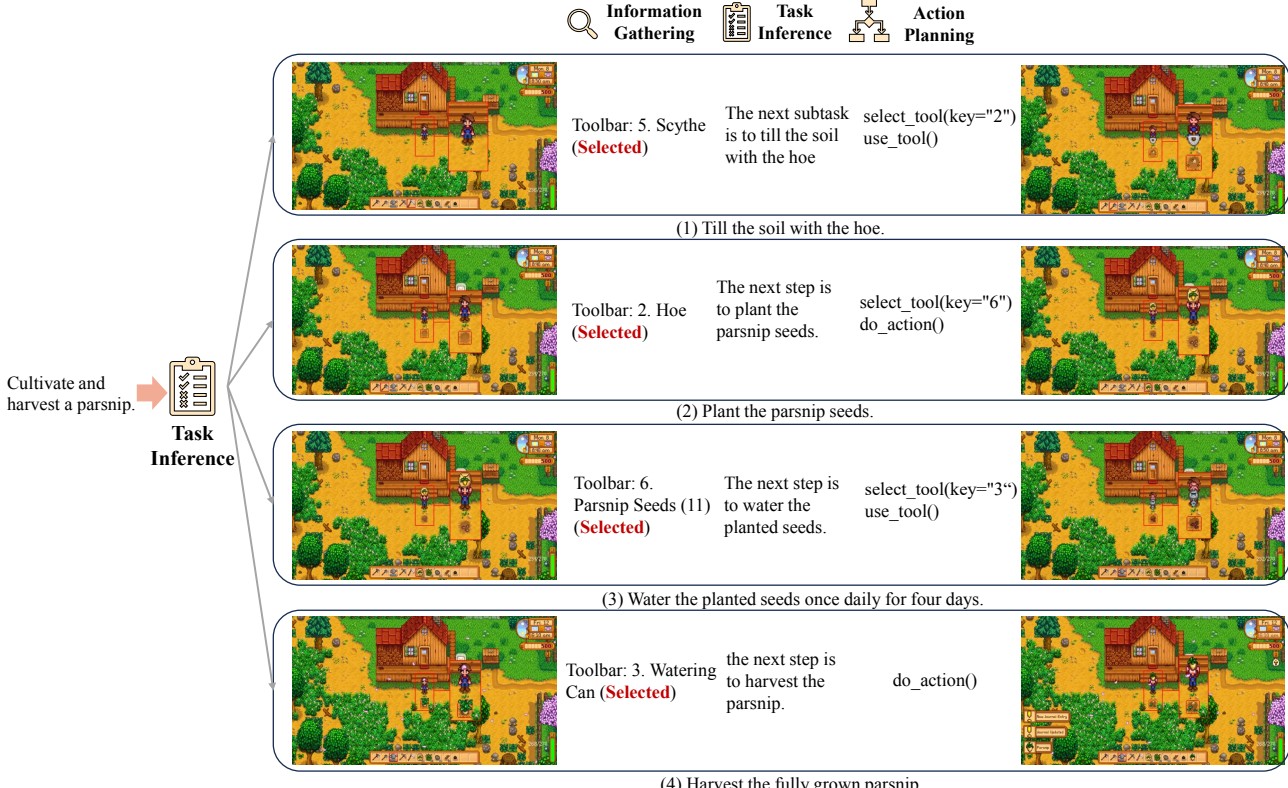

Figure 24: Case study of task inference on decomposing a task into specific sub-tasks. The complete task is to cultivate and harvest a parsnip. CRADLE decomposes the task into four sub-tasks by task inference. Only relevant modules are shown for better readability, though all modules (Figure 4) are executed per iteration.

### G.5. Limitations of GPT-4o

**Fine-grained Control.** Stardew Valley requires that players are positioned precisely to interact with objects, such as doors and NPCs. However, it is difficult for GPT-4o to take a pixel-level precise action. For example, GPT-4o can not take a precise movement even though the speed at which the figure moves is known. To alleviate this problem, we make some composite skills that use template-matching to complete some complex interaction tasks, such as purchasing items.

**Perception in a 2D virtual world .** In Stardew Valley, it's common for a character to be blocked by rocks or trees, and GPT-4o fails to tell if a character is blocked by looking at the image once, and can't predict if the next move will be blocked, which is very easy for a human to do by looking at the image. This indicates that GPT-4o is relatively weak in

perceiving the virtual world in this game. In order to solve this problem, we compare the successive frames before and after in Self-Reflection to enable GPT-4o to judge the corresponding changes.

## H. Dealer's Life 2

### H.1. Introduction to Dealer's Life 2

Dealer's Life 2 is a captivating indie simulation game developed by Abyte Entertainment. Renowned for its intricate negotiation mechanics and humorous portrayal of a pawn shop environment, the game is celebrated for its engaging gameplay that combines strategy with a quirky, cartoonish art style. As a simulation game with role-playing elements, Dealer's Life 2 is played from a first-person perspective, utilizing a mouse for point-and-click interactions and a keyboard for price inputs. This interface facilitates item appraisals, customer interactions, and comprehensive shop management.

In the game, players assume the role of a pawn shop manager, tasked with acquiring and selling various items to make a profit while managing their store's reputation and inventory. Players engage with a wide range of unique non-player characters (NPCs), each with their own distinct behaviors and negotiation styles. Whether bartering over the price of a rare collectible or managing unforeseen shop events, players must hone their haggling and strategic decision-making skills to succeed. Dealer's Life 2 operates in a closed-source format with no APIs available for accessing in-game data or automating gameplay functions. This setup ensures a hands-on experience where players are immersed in the day-to-day challenges of running a pawn shop. This game environment provides a unique and entertaining setting for testifying the GCC's haggling and strategic decision-making abilities. We run our experiments using the latest version, V. 1.013_W96 of the game.

### H.2. Objectives

We concentrate on evaluating the sustained management skills required to maximize profits through buying and selling a diverse range of items from customers. Therefore, the task in this game is defined as *Weekly shop management*, *i.e.*, managing a shop for a week automatically. This game could effectively demonstrate the negotiation ability of the LMM in a trade and bargain. For example, giving an unacceptable price to the customers, *i.e.,* a pretty low price for a seller customer or a very high price for a buyer customer, could cause the deal to fail directly, which brings no profit in this situation. The key is to carefully analyze the description of the item, *e.g.*, the rarity and condition of the item, and more importantly, the response of the customer, *i.e.*, the customer's mood changes.

Contrary to many games that feature detailed tutorials highlighting specific operations and objectives through each crucial step, Dealer's Life 2 does not provide such guidance. This absence transforms the game into a zero-shot, hard open-world task, where the LMM must directly apply its prior knowledge of haggling and strategic decision-making to a new and unfamiliar environment. To provide readers with a clear and straightforward understanding of the task, we illustrate the typical flow of a day's shop management through several key steps, presented in Table 9.

Table 9: Key points in the open-ended mission, *Weekly shop management* in Dealer's Life 2. Figure 25 showcases snapshots of key points (specific sub-figures marked in parenthesis in the table).

| Task: Weekly shop management | Description |
| --- | --- |
| Open shop (Fig. 25a) | Start a new day shop management. |
| Dialog (Fig. 25b) | Choose an option in a dialog. |
| Item Description (Fig. 25c) | View the item information |
| Haggle (Fig. 25d) | Give a price for the item. |
| Deal Result (Fig. 25e) | View the deal results. |
| Stats (Fig. 25f) | View shop stats. |

### H.3. Implementation Details

The implementation of Dealers' Life 2 also strictly follows the GCC framework, which includes Information Gathering, Self-Reflection, Task Inference, Skill Curation, Action Planning, and Action Execution. The details are described in

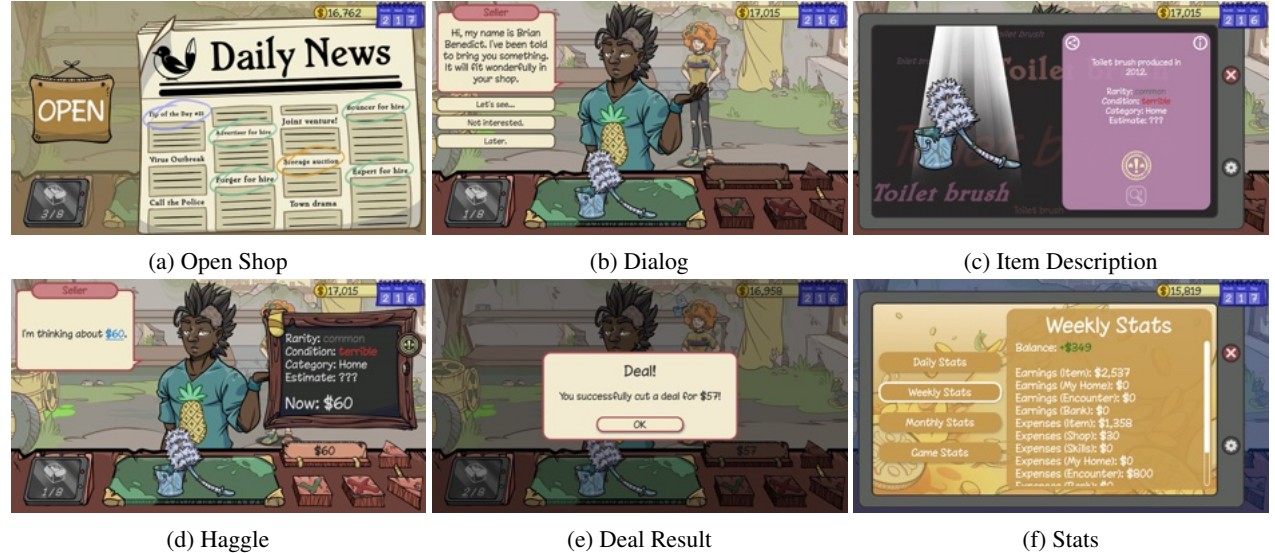

Figure 25: Image examples of key points in the open-ended task of Dealers' Life 2.

Appendix E. Therefore, we emphasize the specific implementations for Dealers' Life 2.

**Procedural Memory.** Due to the absence of a new-user guide, the LMM cannot directly and accurately know the operation method or effect of an action in the game, *e.g.*, giving the price can only use the keyboard to input an integer in an abstract box in the bottom right of the haggle screen as shown in Figure 25d, by directly observing the screen. Unless the player executes an action and observes what is happening, the player cannot know what its effect is. However, this could easily cause severe errors in an open-world environment. For example, if the player gives a price at $100,000 for an item without knowing what the box is, it could cause the player to lose all the money. Besides, this game is very simplified with finite types of screen content and fixed buttons positions for processing the deal, where we could categorize the screen types and design general atomic skills for them. Thus, with a focus on evaluating the LMM's zero-shot haggling and strategic decision-making ability in managing a shop, we believe it is reasonable to skip the skill curation by directly setting several atomic skills as the initialization of the procedural memory, such as "process_dialog()" for clicking on the option of a dialog screen to keep the deal going on as shown in Figure 25b. The description of all the atomic skills is listed as follows:

- *open_shop()*: The function to open the dealer's shop to start dealing for today.

- *give_price(price)*: The function to give a price for the item in the deal. The price must be an integer number.

- *process_dialog()*: The function to click on to choose the first option of the dialog to make the game go on.

- *close_description_page()*: The function to close a description page showing information about the item details, daily stats, or the traits of the buyer or seller.

- *accept_deal()*: The function to click on the check mark to accept the deal on the confirmation dialog.

- *reject_deal()*: The function to click on the cross mark to reject the deal on the confirmation dialog.

- *finish_buy()*: The function to click on the ok button to finish the deal on the confirmation dialog.

- *finish_sell()*: The function to click on the ok button to finish the selling on the confirmation dialog.

**Self-Reflection.** Additionally, as Dealers' Life 2 has no heavy need for a long-term reflection, so we only use the first and last frame of the video as input to reduce the number of tokens used per request. Finally, this self-reflection module could help to keep the game going, instead of sticking to the same point in the game.

**Action Planning.** In this game, we restrict GPT-4 to output only one skill per action because it is a round-based game that does not require frequent execution of actions, and the state of the next time-step after an action is executed is highly uncertain, *e.g.*, the unpredictable mood changes in a customer's response.

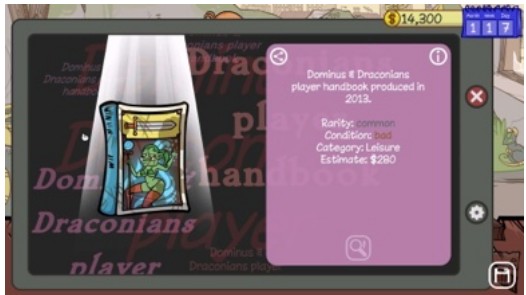

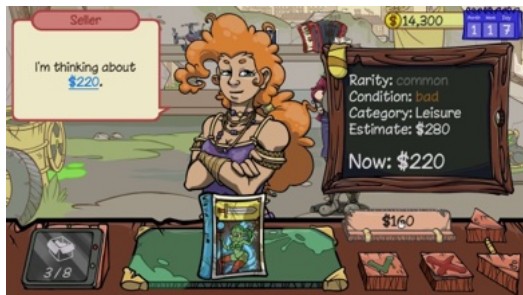

(a) Agent: Given that the customer is a seller and the item on offer is in bad condition with an estimated price of $ 280, it would be wise to offer around $150 to leave room for negotiation.

(b) Agent: Given the item's bad condition and $280 estimate, A reasonable starting counteroffer would be 50-60% of its estimated value, and $160 is a reasonable counteroffer.

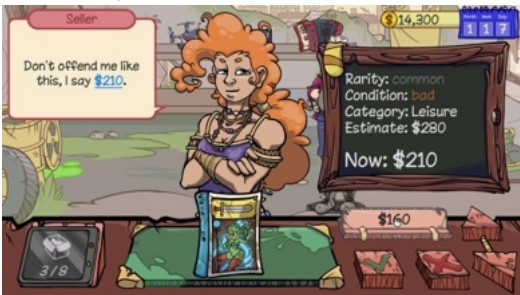

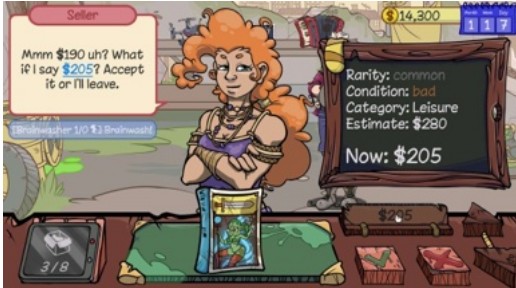

(c) Agent: A reasonable next offer could be $180 or $190, aiming to reach a middle ground. I would still risk with $160 to see if the offer is cut further.

(d) Agent: The customer is proposing her final offer at $205. It will be necessary to accept the offer as this is still a profitable margin.

Figure 26: The reasoning of **CRADLE** in a successful deal with haggling. The price provided by **CRADLE** is keyed in by keyboard and mouse operations in the digital display box in the bottom-right corner.

## H.4. Case Studies

Here we present a few game-specific case studies to further discuss **CRADLE**'s reasoning and decision-making process in the GCC setting.

### H.4.1. SUCCESSFUL NEGOTIATION

Figure 26 illustrates a successful negotiation by **CRADLE** with an NPC seller over an item valued at $280. **CRADLE** determines a strategic starting offer by considering both the item's quality and the customer's initial proposal. Throughout subsequent negotiation rounds, **CRADLE** leverages its memory to maintain an offer close to the initially assessed $160, applying pressure on the customer to reduce their expectations. However, **CRADLE** also demonstrates flexibility, adapting its strategy when faced with the customer's final offer—signaled by their incline to leave. This allows **CRADLE** to secure a final agreement that still yields a profitable deal.

### H.4.2. UNSUCCESSFUL NEGOTIATION

Figure 27 illustrates a scenario where **CRADLE** engages in an unsuccessful negotiation. The seller consistently demands a price above the estimated value of the item, while **CRADLE**, aiming to secure a profit, steadfastly offers a price below the estimated value. A common price cannot be arrived at after rounds of negotiation. Consequently, the negotiation fails to reach an agreement, resulting in the departure of the high-expectation customer.

### H.4.3. ACQUIRING AND SELLING OF A COUNTERFEIT ITEM

Figure 28 depicts a complete buying and selling cycle involving an item later identified as counterfeit by a buyer. Notably, **CRADLE** initially acquires the item without a known price estimate, basing its purchase solely on the seller's initial offer. In subsequent transactions, although **CRADLE** recognizes the item's potential forgery, it continues to trade the item in pursuit

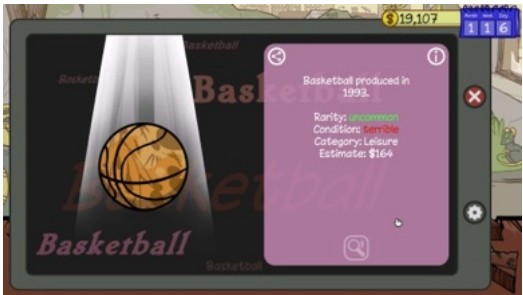
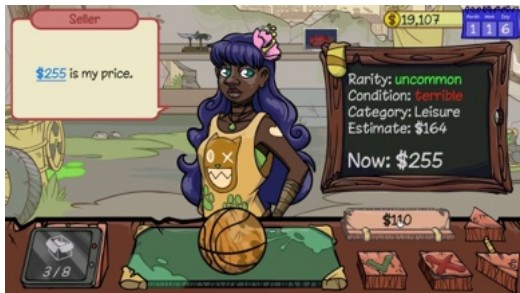

(a) Agent: I will offer an initial price of $100 to reflect the poor condition while allowing room for negotiation since the condition is poor.

(b) Agent: The buyer suggests higher than the estimated price. I should offer a price lower than the item's estimated value. I will offer $110.

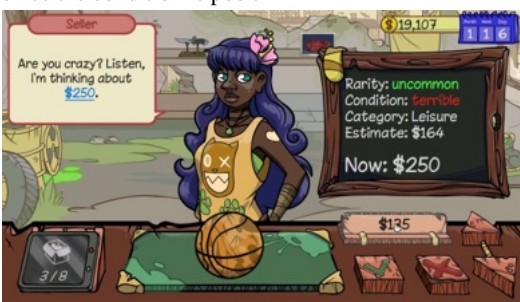
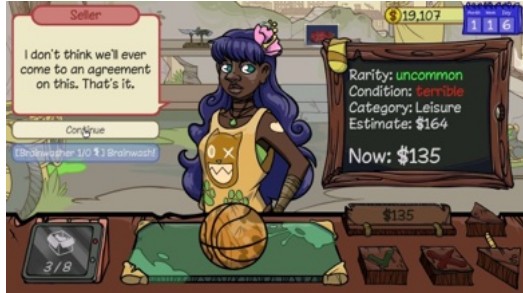

(c) Agent: The seller has countered several times already, suggesting that further negotiations could result in a mutually acceptable price, possibly just a little bit below $164.

(d) Agent: The negotiation failed. My next step is to acknowledge the failure and continue the game-play to interact with other customers to profit.

Figure 27: The reasoning of CRADLE in an unsuccessful deal with haggling. The price provided by CRADLE is keyed in by keyboard and mouse operations in the digital display box in the bottom-right corner.

of maximizing profit. This strategy yields a substantial profit margin of 145%. However, it also poses a risk to the in-game character's reputation, highlighting a strategic decision made under the directive to prioritize profit.

### H.5. Quantitative Evaluation

The effectiveness of CRADLE in terms of game performance is evaluated through experiments conducted over a 7-day gaming scenario using nine quantitative financial metrics. A comprehensive description of these metrics is provided in Section H.6.

### H.6. Evaluation Metrics

Assuming the buying price for item $i$ is denoted by $B_i$, the selling price by $S_i$, the market valuation by $V_i$, and the number of successfully traded items is $n$. To evaluate CRADLE's profitability and performance in Dealer's Life 2, we use the following evaluation metrics:

- **Turnover Rate (TR)** can be calculated as the ratio of the number of successfully traded items to the total number of items considered (both successfully and unsuccessfully traded). It reflects the Agent's ability to successfully complete transactions and can indicate operational efficiency, market competitiveness, and negotiation effectiveness. The calculation formula is $TR = \frac{n}{n+m}$.

- **Gross Profit Margin (GPM)** is the ratio of gross profit to sales revenue, reflecting the dealer's direct profit capability after selling items. The calculation formula is $GPM = \frac{\sum_{i=1}^{n} S_i - B_i}{\sum_{i=1}^{n} S_i}$.

- **Return on Investment (ROI)** is the ratio of profit to investment, used to measure the dealer's return on investment for items. The calculation formula is $ROI = \frac{\sum_{i=1}^{n} S_i - B_i}{\sum_{i=1}^{n} B_i}$.

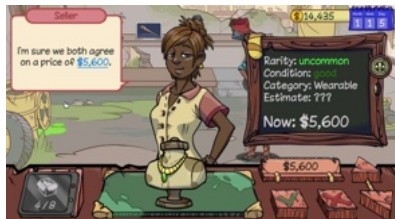 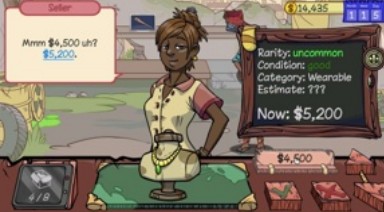 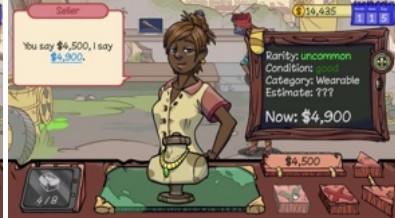

(a) Agent: Given that we do not know the estimated price, I proceed with 70%-80% of the asked price, around $4,200 to $4,480.

(b) Agent: Since the seller is willing to lower the price, the current counteroffer may still be negotiated.

(c) Agent: The counteroffer of $4,900 seems reasonable, but I can go further for the best possible price.

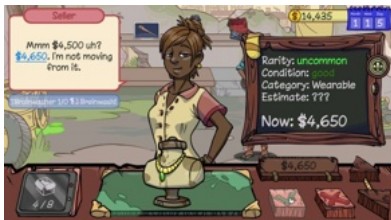 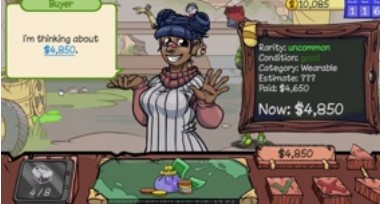 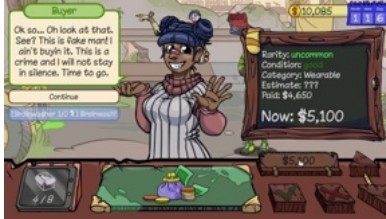

(d) Agent: The response will be final. Given this, accepting the deal is a sensible to secure the item.

(e) Agent: As I am selling the item, I'll try for a higher price for higher profits.

(f) Agent: I note that the customer rejected my counter-offer, calling the item fake.

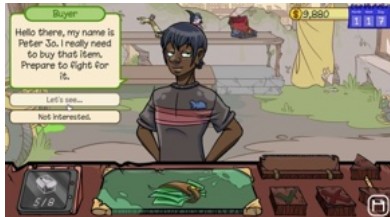 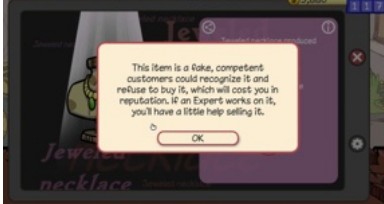 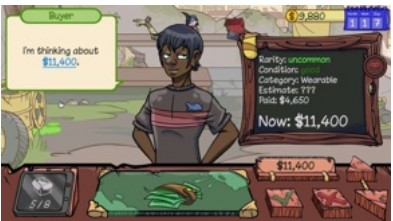

(g) Agent: The seller is in an urgent need to buy the jeweled necklace.

(h) Agent: From the warning window, I am told that this necklace is fake.

(i) Agent: Though the jeweled necklace is fake, accepting this offer provides substantial profit with the buyer's offer at $11,400.

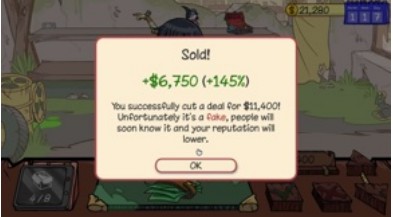

(j) Agent: Despite the successful sale at a good price, the item is revealed as a fake. The added profit is good (+145%)

Figure 28: Case in acquiring and selling an item for multiple attempts with reasoning, and dealing with unexpected information on the authenticity. The price provided by CRADLE is keyed in by keyboard and mouse operations in the digital display box in the bottom-right corner.

- **Valuation Deviation (VD)** reflects the difference between the selling price and the market valuation, used to evaluate the reasonableness of the pricing strategy. It is denoted as $VD = \frac{\sum_{i=1}^{n} S_i - V_i}{\sum_{i=1}^{n} V_i}$.

- **Buying Price to Valuation Ratio (BPVR)** can help determine whether the buying price is lower than the market valuation, reflecting the success of the procurement. The calculation formula is $BPVR = \frac{\sum_{i=1}^{n} B_i}{\sum_{i=1}^{n} V_i}$.

- **Selling Price to Valuation Ratio (SPVR)** reflects the selling price relative to the market valuation, helping to assess the success of the sales. The calculation formula is $SPVR = \frac{\sum_{i=1}^{n} S_i}{\sum_{i=1}^{n} V_i}$.

- **Average Profit Rate (APR)** reflects the overall profitability of the dealer on items. Assuming the return rate for item $i$ is $\frac{S_i - B_i}{B_i}$, the calculation formula of average return rate is denoted as $APR = \frac{1}{n} \sum_{i=1}^{n} \frac{S_i - B_i}{B_i}$.

- **Maximum Return Rate (MRR)** is the highest return rate among all items. The calculation formula is $MRR = \max(\frac{S_1 - B_1}{B_1}, \frac{S_2 - B_2}{B_2}, \ldots, \frac{S_n - B_n}{B_n})$.

- **Minimum Return Rate (mRR)** is the lowest return rate among all items. The calculation formula is $mRR = \min(\frac{S_1 - B_1}{B_1}, \frac{S_2 - B_2}{B_2}, \ldots, \frac{S_n - B_n}{B_n})$.

Table 10: Performance of CRADLE with GPT-4o in Dealer's Life 2 gameplay. "# attempts" represents the total number of all negotiation attempts on items, including both successful and unsuccessful transactions.

| Exp | # attempts | TR↑ | GPM↑ | ROI↑ | VD↑ | BPVR↓ | SPVR↑ | APR↑ | MRR↑ | mRR↑ |
|-----|-----------|-------|-------|-------|-------|--------|--------|-------|--------|-------|
| 01 | 13 | 92.86 | 20.38 | 25.60 | 13.17 | 90.10 | 113.17 | 42.97 | 105.56 | 0.00 |
| 02 | 12 | 91.67 | 18.89 | 23.30 | 23.30 | 100.00 | 123.30 | 17.98 | 97.76 | 0.00 |
| 03 | 12 | 83.33 | 26.81 | 36.63 | 34.39 | 98.36 | 134.39 | 38.68 | 127.27 | -8.06 |
| 04 | 9 | 100.00 | 49.35 | 87.45 | 80.69 | 93.53 | 165.74 | 66.45 | 145.16 | 0.00 |
| 05 | 12 | 100.00 | 20.61 | 25.25 | 25.25 | 100.00 | 125.25 | 23.08 | 44.33 | 0.00 |
| Avg. | 11.6 | 93.57 | 27.21 | 39.65 | 35.36 | 96.40 | 132.37 | 37.83 | 104.02 | -1.61 |

# I. Cities: Skylines

## I.1. Introduction to Cities: Skylines

Cities: Skylines is a single-player open-ended city-building simulation game developed by Colossal Order. In the game, players assume the role of a city planner, tasked with building and managing various aspects of a city to ensure its growth and prosperity. Players engage with a wide range of urban challenges, from managing traffic flow to balancing the budget, and from providing essential services to fostering a vibrant economy. Each decision impacts the city's development, requiring players to hone their planning and strategic decision-making skills to succeed. Effective city management leads to thriving neighborhoods, a growing economy, and high citizen satisfaction, while mismanagement can result in traffic congestion, service shortages, and a decline in population and reputation. Proper planning and responsive governance are crucial for a city that flourishes and remains appealing to its residents and visitors.

As the city's infrastructure and various supporting resources are well-developed, it can attract more people. And a larger population brings more tax revenue and also brings greater expenses to the city's operations. If operated properly, the increasing population can continuously unlock richer urban facilities; if operated improperly, such as road congestion, insufficient services, housing shortage, water and electricity shortage, noise pollution, water pollution, excessive garbage, disease, fire Situation, etc., will all lead to population decline.

This game could be used to evaluate agents' strategies in managing urban development and resource allocation. By simulating different scenarios, agents can experiment with various policies and infrastructural changes to see their impacts on the city's growth and sustainability. Effective strategies may involve optimizing public transportation systems to reduce road congestion, investing in renewable energy sources to prevent power shortages, and implementing comprehensive waste management programs to handle excessive garbage. It offers a risk-free environment to test innovative ideas and learn from the consequences of their actions, ultimately promoting a deeper understanding of sustainable urban development.

Though this game is ranked very positive on Steam, it is notorious for its extremely high difficulty for beginners, as it lacks a detailed tutorial in the beginning, which introduces more challenges for CRADLE to deal with. On the other side,

Although the successor, Cities: Skylines 2, simplified the controls and provided a detailed tutorial for beginners, it became notorious for poor optimization and frequent crashes that caused computer blue screens. As a result, we had to back to using Cities: Skylines 1 instead of 2. And we do not apply any modes to the game. We use the latest version of the game (version 1.17.1-f4).

## I.2. Objectives

Our mission is to build cities so that they can support as many people as possible. Maps in this game are usually very large, which usually costs human players dozens of hours to cover all areas. Besides, the technology tree unlocks as the population grows, which requires multiple turns of planning and building. In this work, we simplified the problem by starting the game near the water and fixing the viewpoint (as shown in Figure 29a), so that CRADLE can leverage the pixel position in the screenshot to locate the position of placed buildings and facilities. Agents start with a plot of land, which is equipped with an entry and an exit from a major highway, providing crucial access for future traffic flow, and proximity to the water source, which is essential for the city's water supply needs. And we focus on the first turn of planning, i.e., pause the game and stop the passage of the in-game time, use the initial starting funds of ℂ70,000 and the most basic road, water, and electricity facilities provided at the beginning of the game, which is enough to achieve the first milestone, *Little Hamlet* with the population of 440 in the game. Then what kind of city can CRADLE create? Can this city ensure water and electricity supply to keep functioning normally while reasonably dividing residential, commercial, and industrial zones? A run is terminated when it reaches the maximal steps, 1000, or the budget is used up (less than ℂ 1000).

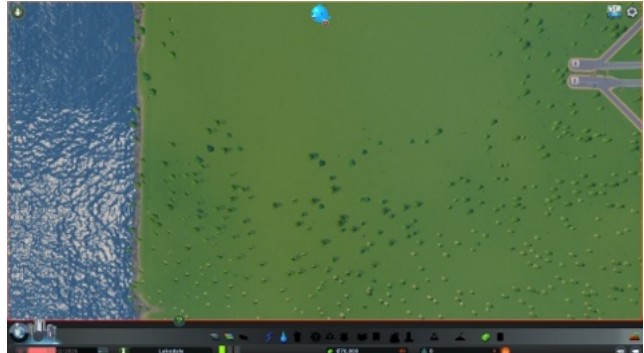

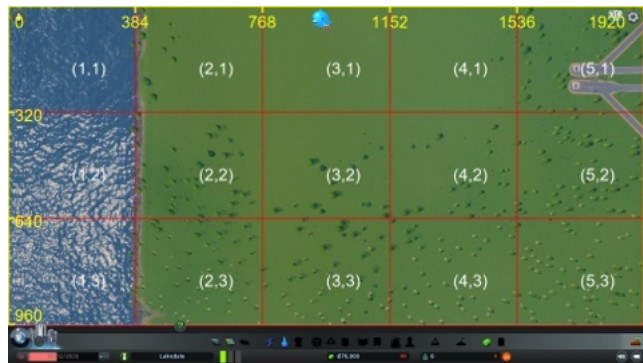

(a) Demonstration for the initialization location of our mission in City: Skylines, which is near the river and contains the entry and exit of the highways.

(b) Visual prompting methods used in Cities: Skylines. The full screenshot is divided into $3 \times 5$ grids and each grid is assigned a unique white coordinate.

## I.3. Evaluation Metric

To measure the completeness of the city built by the agent, we design the following preliminary metrics:

- **Roads in closed loop**: Whether the road is a closed loop, which is crucial for ensuring smooth traffic flow and is beneficial for the city's future development.

- **Sufficient water supply**: To ensure a sufficient water supply, the player needs to construct a water pumping station at the shoreline and then use water pipes to cover every district along the roads. To manage the effluent effectively, the other end of the water pipe network must be equipped with the water drain pipe which is also required to be placed near the shoreline.

- **Sufficient electricity supply**: Both zones and water facilities need electricity to power. To provide sufficient electricity supply, the player can build a coal power plant or wind turbine. Considering coal power plants cost too much and will create heavy pollution, wind turbines combined with the power lines are a better choice at the beginning. The electricity area extends automatically based on the presence of buildings and infrastructure that consume electricity.

- **Zones Coverage > 90%**: The built two-lane road will provide empty space for the development of zones, *i.e.,* residential zone, commercial zone and industrial zone. Residential zones provide houses for people to live in, which is the most essential zone to increase the population. Commercial zones provide places for small businesses, shops,

and services produced in the industrial zones or imported. Industrial zones provide jobs for the residents and products for commercial buildings, which is also important to attract more people to move to the city. This metric is used to evaluate whether 90% of the available areas are covered by the zones. The agent needs to reasonably allocate the areas and proportions of various zones to achieve better city development and attract a larger population.

- **Maximal population**: After CRADLE finishes building, we will unpause the game and start the simulation. Then houses start to be built and residents start to move in. We will record the maximal population during the simulation as the value for this metric.

- **Maximal population with assistance**: We find that cities built by CRADLE manage to meet most of the requirements but suffer a significant population loss due to a few easy-to-fix mistakes. So after CRADLE finishes the design of the city, we apply human assistance that attempts to address these small mistakes within 3 unit operations (building or removing a road/facility/a place of zones is counted as one unit operation). We will also record the maximum population during the simulation in the city with human assistance.

### I.4. Implementation Details

The implementation of Cities: Skylines also strictly follows the GCC framework, which includes Information Gathering, Self-Reflection, Task Inference, Skill Curation, Action Planning and Action Execution. The details are described in Appendix E. Therefore, we emphasize the specific design for Cities: Skylines.

**Pause.** Since the game is stopped before starting the simulation, there is no need to unpause and pause the game while executing actions.

**Visual Prompting.** As shown in Figure 29b, similar to Stardew Valley, we divide each screenshot into $3 \times 5$ grids with an axis based on the resolution of the game screen. Then CRADLE can utilize the pixel-level position in the screenshot to locate the building and facility. We empirically find that this visual prompting method can result in a more precise control of GPT-4o.

**Information Gathering.** In Cities: Skylines, the game's perspective is typically adjustable, allowing players to zoom in and out, rotate, and pan across their cityscape to get a detailed view of their urban development. To ensure consistency and ease of navigation for GPT-4o, we have locked the camera angle and applied a visual prompting method to enhance GPT-4o's visual understanding. Besides, we use GPT-4o to extract key information, such as budget, population, construction information and error messages, in the game.

It is worth noting that in this module, we feed the original screenshot to GPT-4o, rather than the augmented screenshot with axis and coordinates. We find that the numbers and lines may cover some key information and result in wrong OCR recognition. For example, the construction information, "Estimated Production: 120,000m$^3$/week" may be mistakenly interpreted as "Estimated Production: 000,000m$^3$/week" by GPT-4o, due to interference from the lines and numbers. This construction information is a key signal for the suitable place of the water pumping station. For the other modules, we feed GPT-4o with the augmented screenshots.

**Self-Reflection.** Since actions in this game are very short, and each of them has a significant effect shown in the last screenshot. We only use the first screenshot and the last screenshot of the video clip as input to this module, which is proved to be enough for not missing any important information.

**Task Inference.** Due to the lack of a detailed tutorial, we have to provide a draft blueprint for the GPT-4o as the plan at the beginning to help GPT-4o to determine the next step to do. This plan provides guidance to the orders of building each facility and how to build a closed road, how to ensure water and electricity supply and zone placement. Even so, we find that GPT-4o failed frequently to follow the plan, resulting in the lack of building some important facilities, like water pumping stations.

**Skill Curation.** Due to the lack of detailed tutorials in the game, we generate the skills through self-exploration in this game. The skill generation basically involves manipulating the toolbar to understand the items on it. The pseudo-code for skill generation is described in Algorithm 1. This process leverages SAM for objective grounding and GPT-4o to gather information about the objects provided by the game, subsequently generating skills based on a predefined template. An example of the process is shown in Fig 30, 31, 32a, 32b, 33a and 33b.

**Action Planning.** In this game, we only let GPT-4o output one skill for each action since we observe that GPT-4o tends to

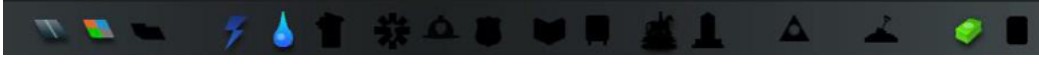

Figure 30: The toolbar in Cities: Skylines

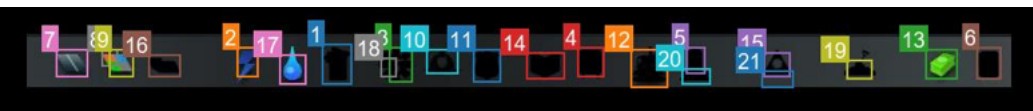

Figure 31: The grounding result of the toolbar in Cities: Skylines

output *try_place* and *confirm placement* together if we allow it to output and execute multiple skills in one action, which is against the intention of our design for the *try_place* action.

**Procedure Memory.** Skills generated through self-exploration are listed below:

- *open_roads_menu()*: The function to open the roads options in the lower menu bar for further determination of which types of roads to build.

- *open_electricity_menu()*: The function to open the electricity options in the lower menu bar for further determination of which types of power facility to build.

- *open_water_sewage_menu()*: The function to open the water and sewage options in the lower menu bar for further determination of which types of water and sewage to build.

- *open_zoning_menu()*: The function to open the zoning options in the lower menu bar for further determination of which types of zonings to build.

- *try_place_two_lane_road*$(x_1, y_1, x_2, y_2)$: Previews the placement of a road between two specified points, $(x_1, y_1)$ and $(x_2, y_2)$, with $x_1, y_1$ being the coordinate of start point of the road, and $(x_2, y_2)$ being the coordinate of end point of the road. This function does not actually construct the road, but rather displays a visual representation of where the road would be placed if confirmed.

- *try_place_wind_turbine*$(x, y)$: Previews the placement of a wind turbine on point, $(x, y)$. This function does not actually construct the wind turbine, but rather displays a visual representation of where the wind turbine would be placed if confirmed.

- *try_place_water_pumping_station*$(x, y)$: Previews the placement of a water pumping station on point, $(x, y)$. This function does not actually construct the water pumping station, but rather displays a visual representation of where the water pumping station would be placed if confirmed.

- *try_place_water_pipe*$(x_1, y_1, x_2, y_2)$: Previews the placement of a water pipe between two specified points, $(x_1, y_1)$ and $(x_2, y_2)$. This function does not actually construct the water pipe, but rather displays a visual representation of where the water pipe would be placed if confirmed.

- *try_place_water_drain_pipe*$(x, y)$: Previews the placement of a water drain pipe on point, $(x, y)$. This function does not actually construct the water drain pipe, but rather displays a visual representation of where the water drain pipe would be placed if confirmed.

- *try_place_commercial_zone*$(x_1, y_1, x_2, y_2)$: Previews the placement of a commercial zone within a rectangular region with diagonal corners at $(x_1, y_1)$ and $(x_2, y_2)$. This function does not actually construct the commercial zone, but rather displays a visual representation of where the commercial zone would be placed if confirmed.

- *try_place_industrial_zone*$(x_1, y_1, x_2, y_2)$: Previews the placement of a industrial zone within a rectangular region with diagonal corners at $(x_1, y_1)$ and $(x_2, y_2)$. This function does not actually construct the industrial zone, but rather displays a visual representation of where the industrial zone would be placed if confirmed.

- *try_de_zone*$(x_1, y_1, x_2, y_2)$: The function to remove the zone in the game. The zone must cover the road.

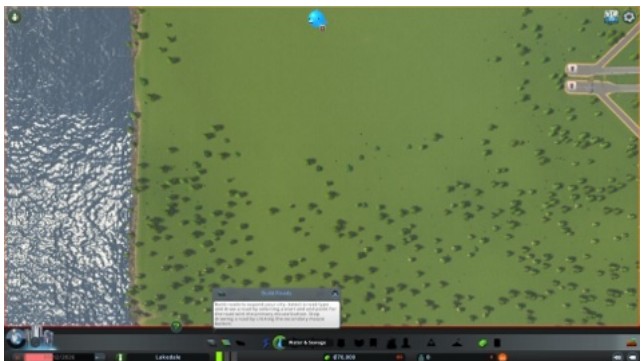

(a) When hovering the mouse over a toolbar item, the pop-up description is "Water & Sewage". The skill generated is then called "open_water_sewage_menu".

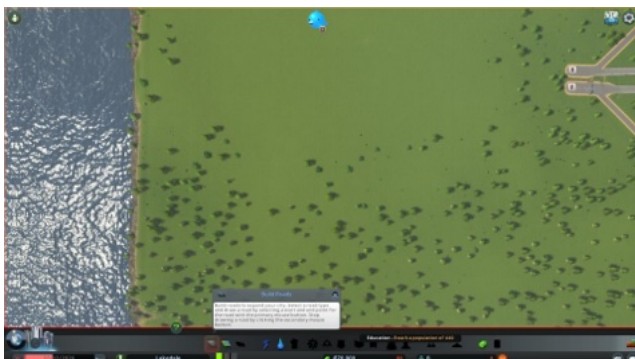

(b) When hovering the mouse over a toolbar item, the pop-up description is "Education - Reach a population of 440". As this is not selectable for now, GPT-4o does not generate a new skill for it.

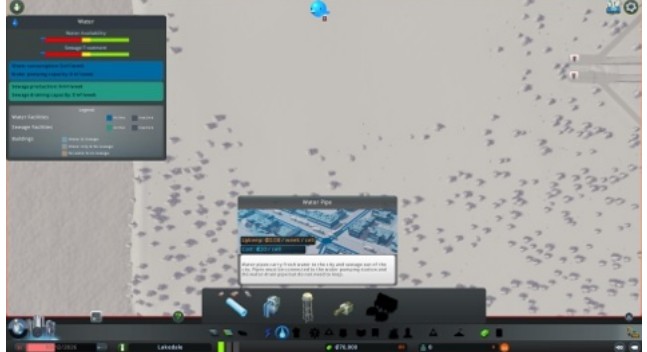

(a) The Water & Sewage menu is opened by executing the new skill "open_water_sewage_menu". The Agent then hovers the mouse over a second-level toolbar item, the pop-up description is "Water Pipe", and the generated skill is called "try_place_water_pipe".

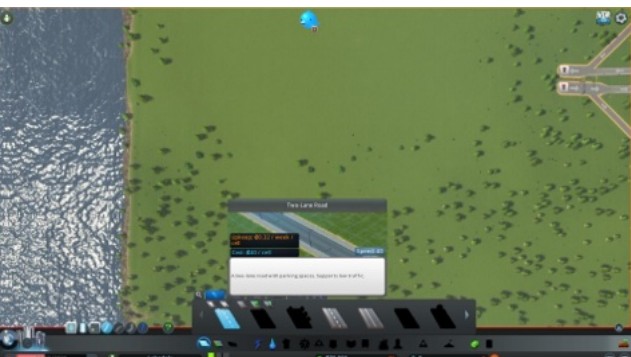

(b) The Roads menu is opened by executing the new skill "open_roads_menu". The Agent then hovers the mouse over a second-level toolbar item, the pop-up description is "Two-Lane Road", and the generated skill is called "try_place_two_lane_road".

- *confirm_placement()*: The function to confirm the placement and build the object after the *try_place_[object]* function.

- *cancel_placement()*: The function to cancel the placement of the object after the *try_place_[object]* function.

**Episodic Memory.** Besides the common information to store in the episodic memory. We initialize the memory with the coordinates of the entry and exit of the highway. Then CRADLE is able to extend the roads according to these two points at the beginning. When a road or a facility such as wind turbine, water pumping station, water drain pipe and water pipe is placed on the map, the corresponding coordinates will also be stored in the memory for future development of the city.

### I.5. Case Studies

#### I.5.1. FAILURE FOR ROAD BUILDING.

As shown in Figure 34, sometimes GPT-4o will build a long road, which ends on the top of water. The recorded endpoint of the road is actually the projection of the road on the sea level, resulting in the offset from the projection point and the real endpoint of the road. It leads to the failure of extending the road to the other places.

Figure 34b, 34c, 34d and 34e tells a story that GPT-4o sometimes forgets to confirm the placement (from 34c to 34d) and directly moves to the next step of building the next road (from 34d to 34e), resulting in the disconnection of the roads.

#### I.5.2. FAILURE FOR SUFFICIENT WATER SUPPLY.

Figure 35 displays three cases where CRADLE fails to ensure the water supply due to the disconnection of water pipes and the missing water pumping station. All of them can be fixed within three unit operations. As shown in Figure 35b and 35f,

---

**Algorithm 1** Skill Generation

---

**Input:** Toolbar with objects, Skill template
**Output:** Procedure memory with generated skills

1 Initialize procedure memory
2 **for** *each object in the toolbar* **do**
3     Hover the mouse on the object to get the description  Generate skill using GPT-4o based on the object description and the skill template  Store generated skill in procedure memory  Execute the generated skill to enter the second-level toolbar
4     **for** *each object in the second-level toolbar* **do**
5         Hover the mouse on the object to get the description  Generate skill using GPT-4o based on the object description and skill template  Store generated skill in procedure memory

6 **return** *procedure memory*

---

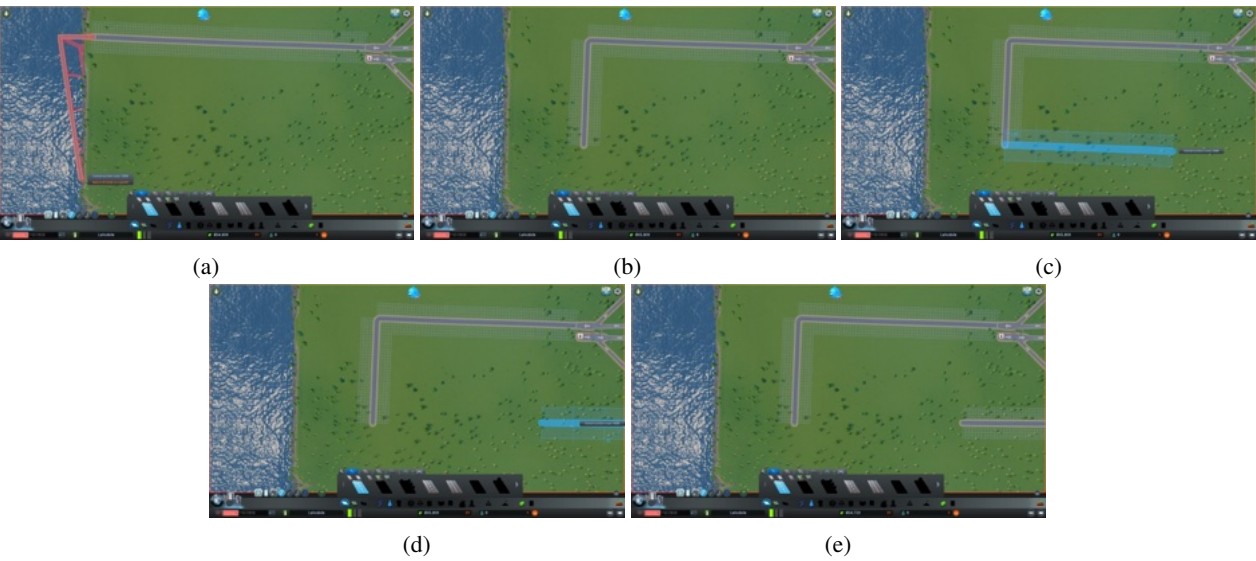

Figure 34: Failure cases of building roads in a closed loop. Figure 34a shows that the road is built over the water and is difficult to continue. Figure 34b, 34c, 34d and 34e tells a story that GPT-4o sometimes forgets to confirm the placement (from 34c to 34d) and directly moves to the next step of building (from 34d to 34e), resulting in the disconnection of the roads.

we observe a significant increase in the population if these mistakes are fixed, which proves that **CRADLE** already has the ability to build a reasonable city but some minor adjustments are needed.

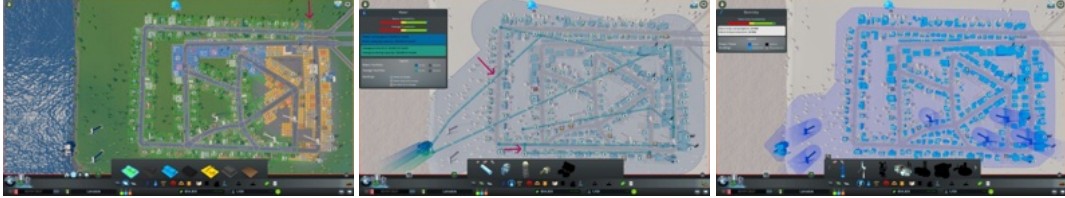

(a) **CRADLE**'s craftwork I. The upper left corner of the city is experiencing a severe local water shortage since the water pipes there are not connected. **Population: 800+**.

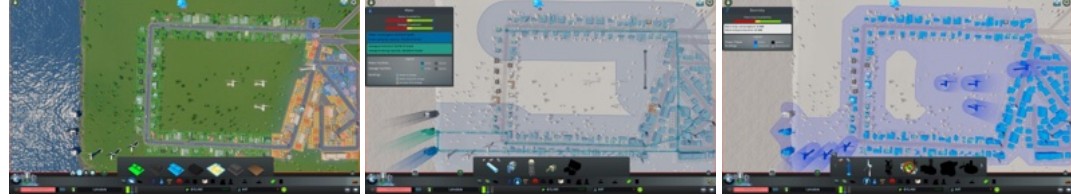

(b) **CRADLE**'s craftwork I with assistant within three unit operations to develop the idle area in the upper right corner of the city into a residential zone and put two water pipes to ensure all the water pipes connected and cover the whole city. **Population: 1150+**.

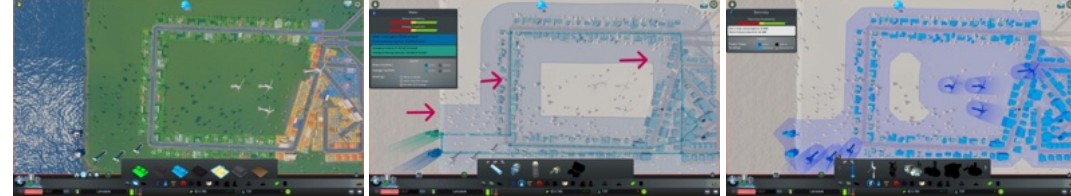

(c) **CRADLE**'s craftwork II. The left side of the city a localized area on the right suffers from water shortage because of the water pipes connected issues. **Population: 640+.**

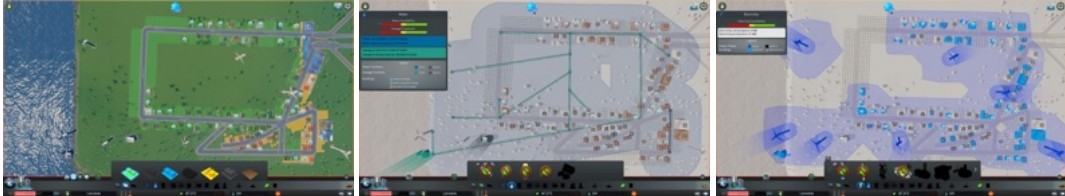

(d) **CRADLE**'s craftwork II with assistant within three unit operations by selling the redundant water pumping station and the independent water pipe on the right to get some budget and using the budget to get the water pipes connected. **Population: 730+**.

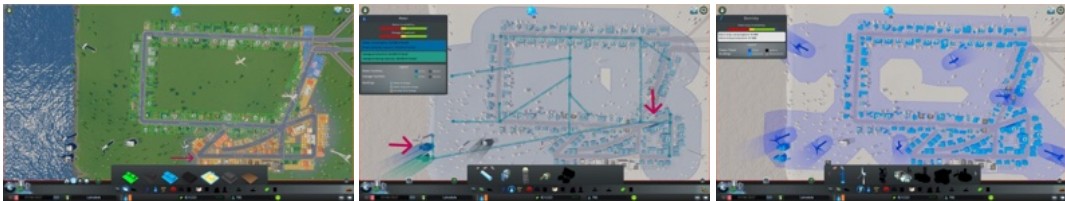

(e) **CRADLE**'s craftwork III. The entire city is experiencing a severe water shortage due to the lack of the water pumping station. **Population: 200+**.

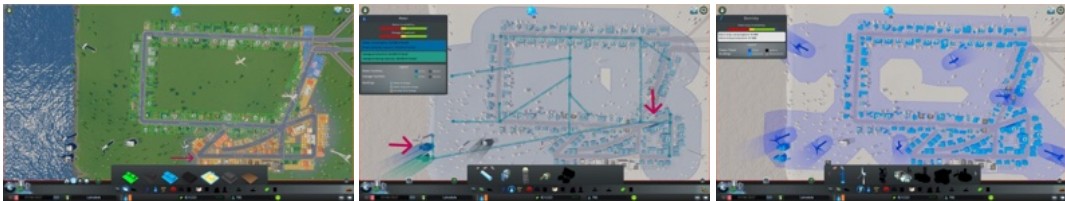

(f) **CRADLE**'s craftwork III with assistant within three unit operations to place the water pumping station, lay water pipe on the right side and develop the bottom area with industrial zones. **Population: 780+**.

Figure 35: Demonstrations of three cities built by **CRADLE** in zoning view (left), water view (middle) and electricity view (right). Figures 35b, 35d, 35f show the cities with human assistance to address construction issues (shown in red arrow). Populations shown in the figures are close to but not exactly the maximal population since they are changed dynamically.

## J. Software Applications

### J.1. Selected Software Applications

Besides targeting complex digital games, CRADLE also includes an initial benchmark task set across diverse software applications. The selected applications include Chrome, Outlook, Feishu, CapCut, and Meitu. These applications cover popular applications for daily tasks in different usage categories, such as web browsing, communication, work, and media manipulation. Table 11 shows the exact application versions benchmarked in this paper. Five distinct tasks were designed for each application to represent their target domains and explore the difficulties posed to LMM-based agents and analyze their limitations. Figure 3 shows an overview of all tasks across applications and Tables 12 and 13 detail each task.

Chrome and Outlook were selected as common representatives for web browsing and e-mail, with well-known functionality and UI design. CapCut and Meitu are two popular media editing applications for video/image editing with their own interaction styles. Lastly, Feishu (also known as Lark) is an office collaboration and productivity application, which includes messaging, calendar/meetings, and approval workflows. It represents a complex business application that doesn't strictly follow OS-specific UI guidelines. To the best of our knowledge, this is the **first agent** targeting applications like CapCut, Meitu, and Feishu.

#### J.1.1. BRIEF DESCRIPTIONS

**Chrome** is a web browser developed by Google. It allows users to access and utilize online resources through activities such as browsing websites, streaming videos, and using web applications. Additionally, users can customize their browsing experience with various extensions, manage bookmarks and passwords, and synchronize their data across multiple devices for seamless access.

**Outlook** is an application by that allows users to manage emails, calendars, contacts, and tasks. It includes tools for communication and scheduling through features such as sending and receiving emails, setting up meetings, and keeping track of appointments. Additionally, users can customize their experience and integrate Outlook with other Microsoft Office applications.

Table 11: Exact software versions utilized in the described experiments. Similar versions should behave similarly.

| Software | Version |
|---|---|
| Chrome | 125.0.6422.142 |
| Outlook | 1.2024.529.200 |
| CapCut | 4.0.0 |
| Meitu | 7.5.6.1 |
| Feishu | 7.19.5 |

**CapCut** is a popular video editing application developed by ByteDance. It provides easy-to-use editing tools and and enables users to create quality videos with a range of advanced features. CapCut offers a set of editing tools, including trimming, cutting, merging, and splitting video clips; the application of various effects, filters, and transitions; as well as adjusting speed, and adding music or text overlays.

**Meitu** is a photo editing application. It is designed to cater to a broad audience and enables users to enhance and transform their photos with minimal effort. Meitu offers editing tools, including basic adjustments like cropping, rotating, and resizing, as well as advanced features such as beauty retouching, filters, and special effects. Additionally, Meitu offers a wide range of stickers, frames, and text options to further personalize photos.

**Feishu**, also known as Lark, is a business communication and collaboration platform by ByteDance. It integrates various tools for office workflows and project management. Feishu offers a wide array of functionalities, including instant messaging, video conferencing, file sharing, and collaboration within the app. It also includes an integrated calendar, which helps users schedule and manage meetings and events, and task management tools that allow users to assign and track tasks.

### J.2. Software Tasks

For each of the five applications, we selected a set of representative tasks for their respective domains. For example, search, navigation, and settings tasks on Chrome; sending, searching, and deleting emails, plus changing settings on Outlook; basic video and image editing operations on CapCut and Meitu (*e.g.,* adding special effects and creating a collage); and communication and organization operations on Feishu. Tables 12 and 13 describe in detail the 25 tasks CRADLE performs and analyzes on the five selected applications; also illustrated in Figures 36, 37, 38, 39, 40, and 3.

It is worth noting that we add a *special* task on CapCut to demonstrate the agent's ability for tool use. In this task, a

Table 12: Task Descriptions for Chrome, Outlook, and CapCut. *Difficulty* refers to how hard it is for our agent to accomplish the corresponding tasks. Figures 36, 37, and 38 illustrate each task (specific sub-figures marked in parenthesis in the left-most column along with task name).

| Software | Description | Difficulty |
|---|---|---|
| **Chrome** | | |
| Download Paper (Fig. 36a) | Search for an article with a title like *{paper_title}* and download its PDF file. | Hard |
| Post in Twitter (Fig. 36b) | Post "It's a good day." on my Twitter. | Hard |
| Open Closed Page (Fig. 36c) | Open the last closed page. | Easy |
| Go to Profile (Fig. 36d) | Find and navigate to *{person_name}*'s homepage on GitHub. | Medium |
| Change Mode (Fig. 36e) | Customize Chrome to dark mode. | Medium |
| **Outlook** | | |
| Send New E-mail (Fig. 37a) | Create a new e-mail to *{email_address}* with subject "Hello friend" and send it. | Medium |
| Empty Junk Folder (Fig. 37b) | Open the junk folder and delete all messages in it, if any. | Medium |
| Reply to Person (Fig. 37c) | Open an e-mail from *{person_name}* in the inbox, reply to it with "Got it. Thanks.", and click send. | Medium |
| Find Target E-mail (Fig. 37d) | Find the e-mail whose subject is "Urgent meeting" and open it. | Easy |
| Setup Forwarding (Fig. 37e) | Set up email forwarding for every email received to go to *{email_address}*. | Medium |
| **CapCut** | | |
| Create Media Project (Fig. 38a) | Create a new project, then import *{video_file_name}* to the media, click the "Audio" button to add music to the timeline, and finally export the video. | Hard |
| Add Transition (Fig. 38b) | Open the first existing project. Switch to Transitions panel. Drag a transition effect between the two videos, and then export the video. | Medium |
| Crop by Timestamp (Fig. 38c) | Delete the video frames after five seconds and then before one second in this video, and then export the video. | Medium |
| Add Sticker (Fig. 38d) | Open the first existing project. Switch to Stickers panel. Drag a sticker of a person's face to the video, and then export the video. | Hard |
| Crop by Content (Fig. 38e) | Crop the video when the ball enters the goal, and then export the video. | Very hard |

Table 13: Task Descriptions for: Meitu, and Feishu. *Difficulty* refers to how hard it is for our agent to accomplish the corresponding tasks. Figures 39, and 40 illustrate each task (specific sub-figures marked in parenthesis in the left-most column along with task name).

| Software | Description | Difficulty |
|---|---|---|
| **Meitu** | | |
| Apply Filter (Fig. 39a) | Apply a filter from Meitu to *{picture_file_name}* and save the project. | Easy |
| Cutout (Fig. 39b) | Cutout a person from *{picture_file_name}* and save the project. | Easy |
| Add Sticker (Fig. 39c) | Add a flower sticker to *{picture_file_name}* and save the picture. | Middle |
| Create Collage (Fig. 39d) | Make a collage using 3 pictures and save the project. | Hard |
| Add Frame (Fig. 39e) | Add a circle-shaped frame to *{picture_file_name}* and save the picture. | Hard |
| **Feishu** | | |
| Create Appointment (Fig. 40a) | Create a new appointment in my calendar anytime later today with title "Focus time". | Hard |
| Message Contact (Fig. 40b) | Please send a "Hi" chat message to *{contact_name}*. | Easy |
| Send File (Fig. 40c) | Send the AWS bill file at *{pdf_path}* in a chat with *{contact_name}*. | Hard |
| Set User Status (Fig. 40d) | Open the user profile menu and set my status to "In meeting". | Medium |
| Start Video Conference (Fig. 40e) | Create a new meeting and meet now. | Easy |

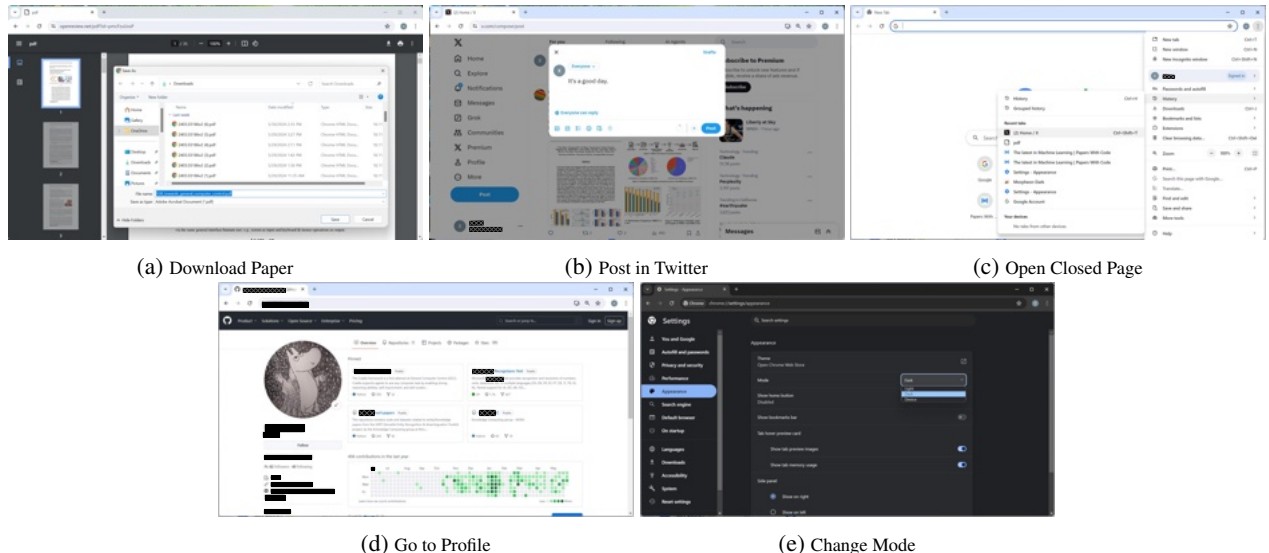

(a) Download Paper          (b) Post in Twitter          (c) Open Closed Page

(d) Go to Profile          (e) Change Mode

Figure 36: Screenshots of Chrome tasks.

pre-defined skill uses GPT-4o as a tool for video understanding capabilities. The skill can be selected to answer content-based questions about a video (*e.g.,* "when the ball enters the goal") and the response be used during task completion. This task is illustrated in detail in Figure 47.

### J.3. Quantitative Evaluation

We calculate **CRADLE**'s performance over the 25 tasks in the applications set. Each task is executed five times and performance is measured in three metrics: success rate, average number of steps taken by the agent (and variance over the five runs), and efficiency. *Efficiency* is defined as the ratio between the expected number of steps in a given task and the total number of steps taken by the agent. The expected number of steps per task is calculated by having humans perform each task.

Table 14 and Figure 41 show the details of the evaluation. **CRADLE** presents overall good performance over the diverse tasks and applications (compared to Expected Steps, **CRADLE** achieves an overall efficiency of 50%). However, performance for certain tasks can vary considerably due to different factors. The main reason for the higher number of task step during agent execution is the frequent incorrect positioning decisions for the mouse, *i.e.,* the backbone model chooses a position of bounding box tag that does not correspond to the UI item described in the model reasoning. We discuss examples of task-specific issues in Sections J.5 and J.6 below.

It is worth noting that in Chrome's task 3 ("Open the last closed page"), **CRADLE** knows how to use the shortcut key directly, calling the key_press skill directly with the correct keyboard shortcut: *'Ctrl + Shift + T'*, whereas humans typically do not know this.

To further evaluate the performance of **CRADLE** in diverse software applications scenarios, we provide quantitative results over OSWorld, a new contemporaneous benchmark with similar characteristics to our settings. More details in Appendix K and overview of the results in Table 15.

### J.4. Implementation Details

The implementation of **CRADLE** targeting all five software applications follows the GCC setting and framework modules (which include Information Gathering, Self-Reflection, Task Inference, Skill Curation, Action Planning, and Action Execution). Implementation details of the overall framework are described in Appendix E. Therefore, here we emphasize any application-specific differences or customization.

To apply **CRADLE** to the target application set described in this appendix, we start with base common prompts, and customize those prompts for specific modules, if necessary, to handle application-specific characteristics. For example, for

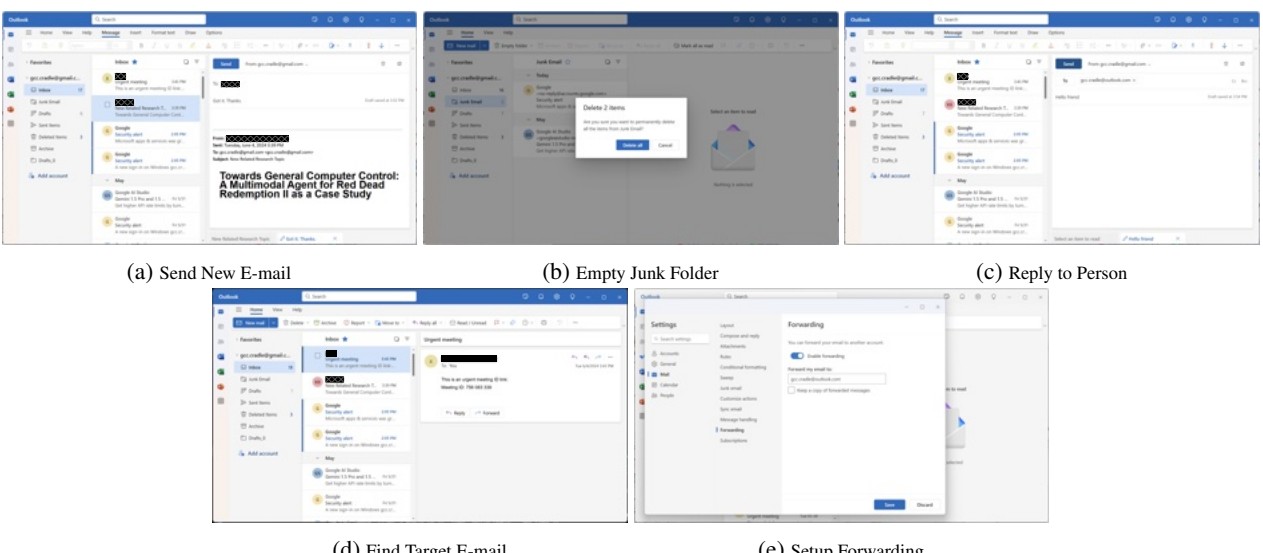

(a) Send New E-mail        (b) Empty Junk Folder        (c) Reply to Person

(d) Find Target E-mail        (e) Setup Forwarding

Figure 37: Screenshots of Outlook tasks.

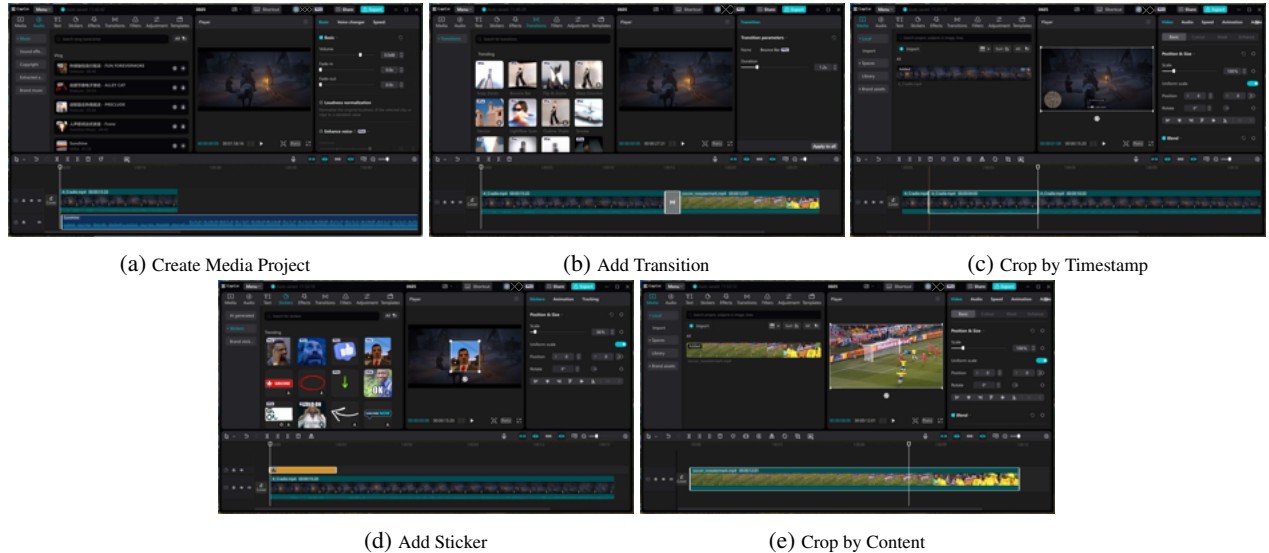

(a) Create Media Project        (b) Add Transition        (c) Crop by Timestamp

(d) Add Sticker        (e) Crop by Content

Figure 38: Screenshots of CapCut tasks.

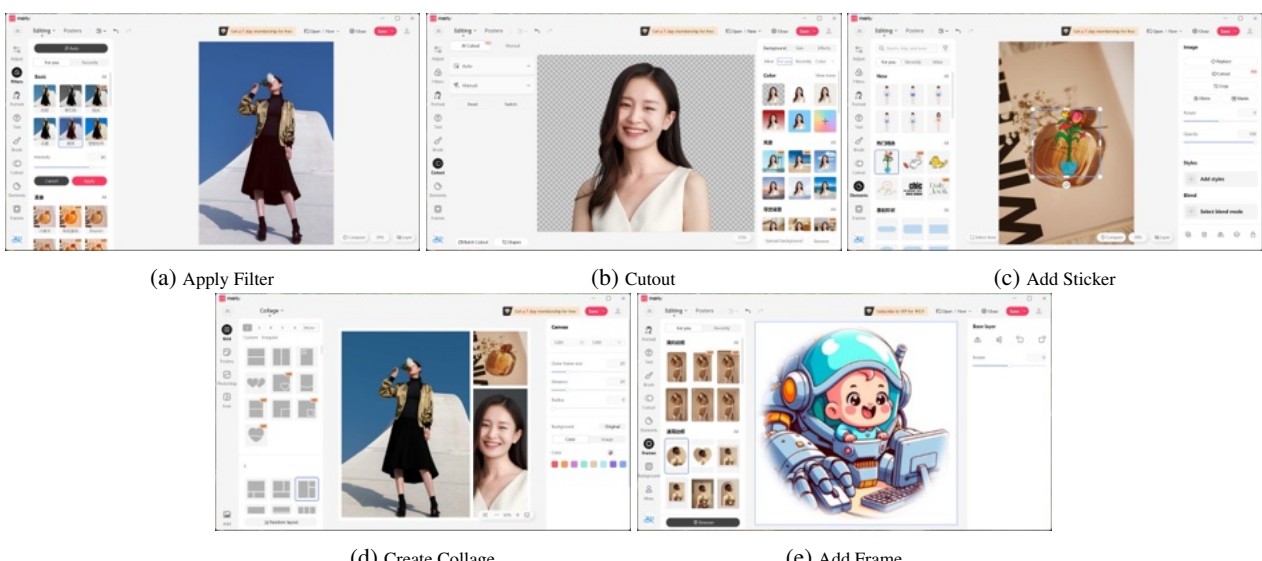

(a) Apply Filter      (b) Cutout      (c) Add Sticker

(d) Create Collage      (e) Add Frame

Figure 39: Screenshots of Meitu tasks.

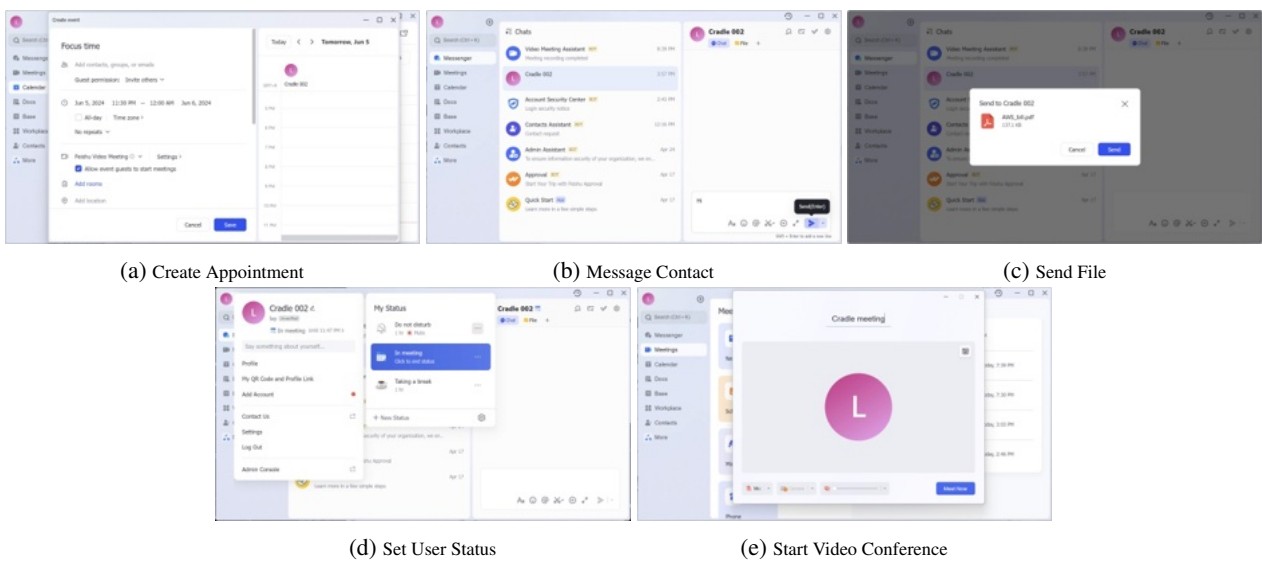

(a) Create Appointment      (b) Message Contact      (c) Send File

(d) Set User Status      (e) Start Video Conference

Figure 40: Screenshots of Feishu tasks.

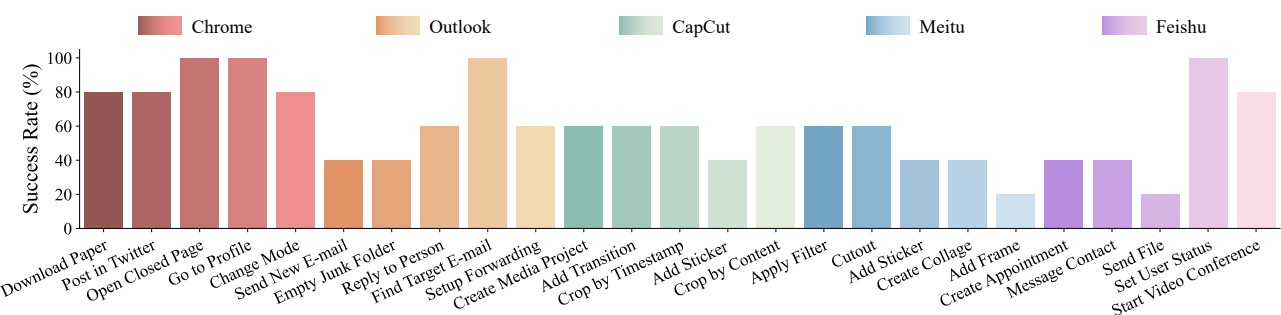

Figure 41: Success rates for tasks in software applications

CapCut we add few-shot examples for Self-Reflection, to let it properly perform success detection, as the application UI by itself is non-standard and sometimes provides little post-action feedback to users, making it harder for the backend model to determine action success.

**Information Gathering.** Noticeably, GPT-4o presents the same limitations in both spatial reasoning (*e.g.,* confusing up/down, left/right) and image understanding identifying specific UI items or the state of the forefront GUI, across all applications.

To help mitigate such issues, we perform augmentation on the captured screenshots similarly to the Set-of-Mark (SoM) approach (Yang et al., 2023a), by only utilizing SAM (Kirillov et al., 2023) to generate potential UI items bounding boxes and assign them numerical tags. Our SoM-like augmentation *differs* from recent agent-related work (*e.g.,* (Zhang et al., 2024; Xie et al., 2024)), which use OS-specific APIs to draw ground-truth bounding boxes for interactable elements (plus UI structure info, like types and element tree) to the results, while CRADLE relies only on image input and the segmentation output as augmentation. To make this distinction explicit, we call our augmentation approach SAM2SOM [10]. Figure 45 illustrates the difference. While our approach produces many more potential bounding boxes, it is more general by relying only on a screenshot (or video frame).

To ensure all bounding box labels are consistently positioned, CRADLE's SAM2SOM implements two rendering styles, as shown in Figure 43 first and second rows. In the *standard* style, we pad the SAM2SOM-enhanced image when showing the label IDs in the upper left corner of the bounding boxes (to prevent labels from hiding the contents of small areas), so no numerical label ID is drawn outside the image area). In the *uniform* style, all bounding boxes utilize single-color borders with labels in black text over white background, placed within the bounding box area (top left corner).

Moreover, in specific situations we may still need to refine SAM2SOM's output further. For example, in the Feishu case, we observe that watermarks generated by the software affect the segmentation negatively, complicating GPT-4o's selection of the correct bounding boxes to interact with. Therefore, we implement a simple filtering method for such watermarks. This filter is enabled only in the Feishu benchmark and, as shown in Figure 44, can greatly reduce the number of unnecessary bounding boxes (from 216 to 166, in this example).

In addition to using the SAM2SOM method for image augmentation, we also redraw the mouse pointer not present in captured screenshots in a more prominent magenta color based on its screen position, to emphasize both its presence and position for image understanding

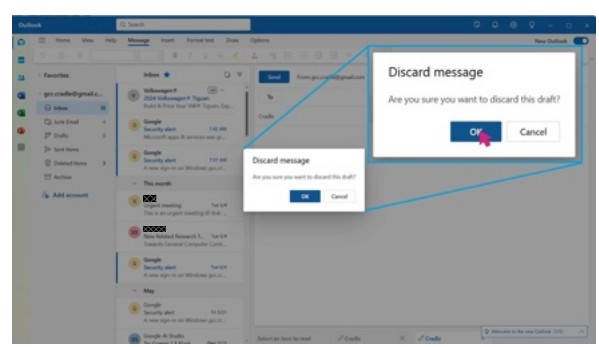

Figure 42: Sample augmented image w/ drawn mouse pointer. Zoom overlay shows the image difference.

(*e.g.,* Figure 42). The augmentation process in Information Gathering can then result in four versions of a screenshot: a) base image, b) SAM2SOM image, c) base image with mouse pointer, and d) SAM2SOM image with mouse pointer.

---

[10]We do not claim the method itself as a core contribution. SAM2SOM is used to illustrate a possible extra capability of the backend model, as mitigation for current spatial reasoning issues.

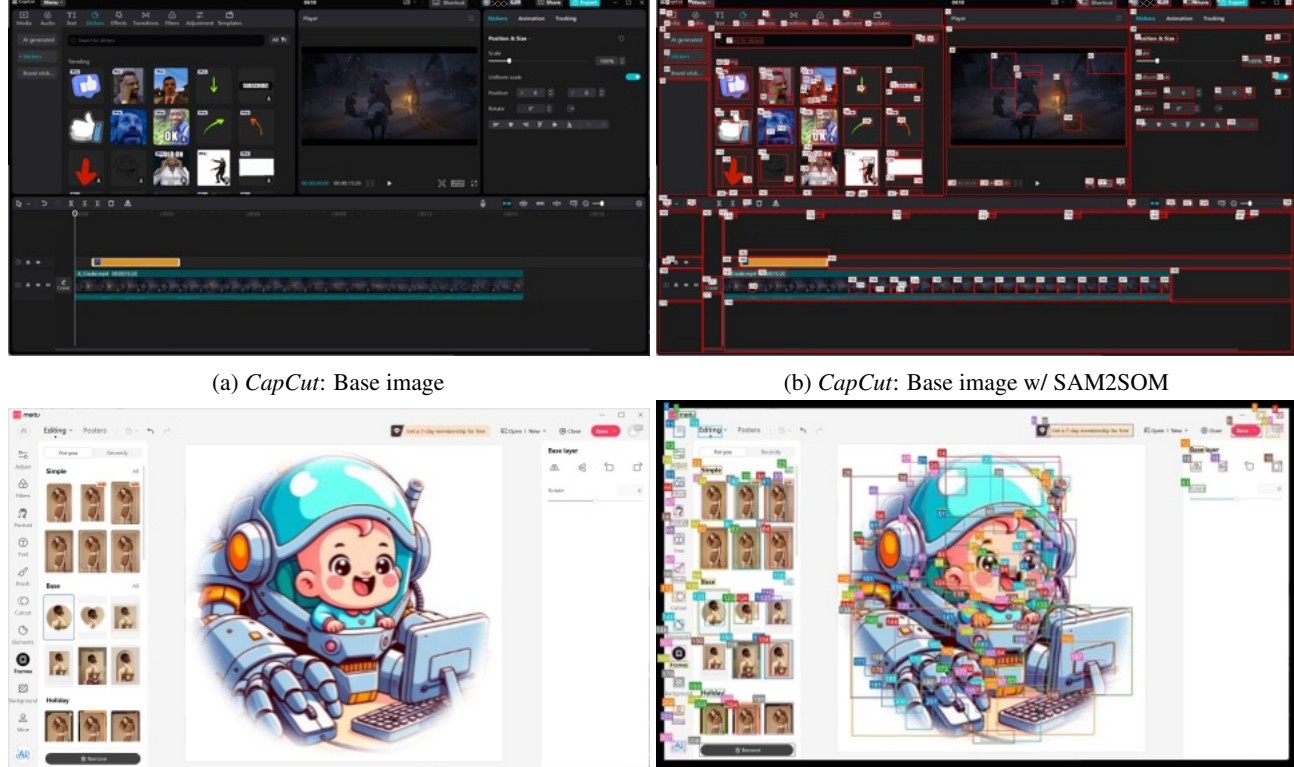

(a) *CapCut*: Base image

(b) *CapCut*: Base image w/ SAM2SOM

(c) *Meitu*: Base image

(d) *Meitu*: Base image w/ SAM2SOM

Figure 43: Image examples of the two SAM2SOM augmentation styles. As CapCut's UI (top row) has very dark background, we utilize single-color borders with IDs in black text over white background, placed within the bounding box area. Other application software and OSWorld use the "standard" SAM2SOM multi-color style, as shown for Meitu (bottom row).

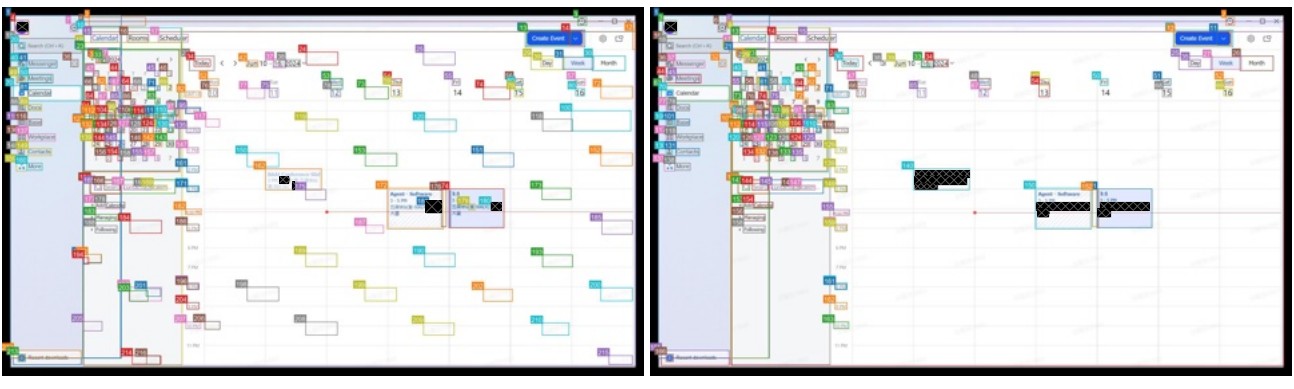

(a) SAM2SOM image w/ watermarks

(b) SAM2SOM image w/o watermarks

Figure 44: Examples of filtering watermark in Feishu. The number of labels is greatly reduced from 216 to 166.

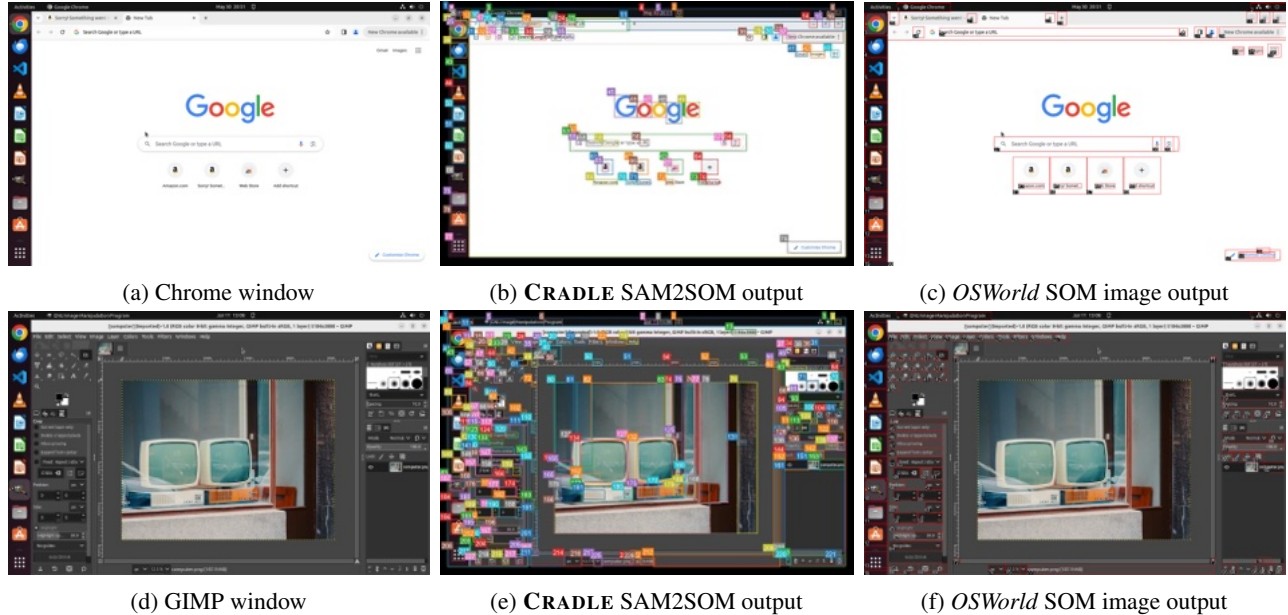

| (a) Chrome window | (b) CRADLE SAM2SOM output | (c) *OSWorld* SOM image output |
| (d) GIMP window | (e) CRADLE SAM2SOM output | (f) *OSWorld* SOM image output |

Figure 45: Comparison of CRADLE's visual-only SAM2SOM and *OSWorld*'s API-based SOM image results. Chrome: 78 vs. 53 bounding boxes; GIMP: 227 vs. 98 bounding boxes.

**Self-Reflection.** As the applications in the software set are much less dynamic than complex games, there is no need to send multiple video frames to Self-Reflection. For the software applications, pre- and post-action screenshot usually suffice, *i.e.,* one image before and one image after an action is executed. Digital games often have continuous and dynamic environments that require multiple frames to properly capture the full context and thus help the backbone LMMs understand what happened. In contrast, software operations are typically more discrete and static, where the state before and after an action provides sufficient information for most analysis.

Nonetheless, we find that irrespective of images used, GPT-4o sometimes can have difficulty determining the success of certain tasks. For example, when downloading a file on Chrome, after either pressing 'Ctrl + S', or using a 'Save' menu, the agent must also press 'Enter' or click the 'Save' button to complete the task. However, GPT-4o often assumes the task is complete when the dialog opens and before this final step. Similar cases of incorrect conclusion happen when an action correctly closes a new panel or dialog. To address this category of issues, we add mandatory reasoning rules in the prompt for the Self-Reflection module to help mitigate such mistakes. If for specific applications this still remains an issue, we can use few-shot image examples to reinforce how the backend model should correctly judge success.

Table 14: Application Software results. *Success Rate* determines the ratio of successful completions over five runs. *Average Steps* refers to the number of actions the agent takes to fulfil a task, if successful. *Expected Steps* represents the number of steps as estimated by humans performing the task. *Efficiency* represents the ratio between the expected number of steps and the total number of steps taken by the agent.

| Software | Success Rate | Average Steps | Expected Steps | Efficiency |
|---|---|---|---|---|
| **Chrome** | 88% | $8.23 \pm 6.75$ | 4.20 | 48.05% |
| Download Paper | 80% | $16.00 \pm 5.52$ | 6 | 37.50% |
| Post in Twitter | 80% | $11.75 \pm 5.26$ | 7 | 61.14% |
| Open Closed Page | 100% | $1.00 \pm 0$ | 3 | 300.00% |
| Go to Profile | 100% | $4.00 \pm 0.63$ | 1 | 25.00% |
| Change Mode | 80% | $11.25 \pm 4.71$ | 4 | 35.56% |
| **Outlook** | 60% | $7.13 \pm 5.61$ | 4 | 48.48% |
| Send New E-mail | 40% | $11.00 \pm 4$ | 5 | 45.45% |
| Empty Junk Folder | 40% | $8.50 \pm 3.50$ | 3 | 35.29% |
| Reply to Person | 60% | $8.33 \pm 4.71$ | 4 | 48.02% |
| Find Target E-mail | 100% | $1.40 \pm 0.80$ | 1 | 71.43% |
| Setup forwarding | 60% | $12.00 \pm 4.90$ | 7 | 58.33% |
| **CapCut** | 56% | $10.87 \pm 5.56$ | 4.80 | 44.16% |
| Create Media Project | 60% | $13.67 \pm 5.25$ | 7 | 51.20% |
| Add transition | 60% | $10.67 \pm 4.03$ | 4 | 37.49% |
| Crop by Timestamp | 60% | $11.00 \pm 5.66$ | 5 | 45.45% |
| Add Sticker | 40% | $12.00 \pm 8.00$ | 4 | 33.33% |
| Crop by Content | 60% | $7.00 \pm 1.41$ | 4 | 57.14% |
| **Meitu** | 44% | $12.36 \pm 3.34$ | 8.00 | 64% |
| Apply Filter | 60% | $14.67 \pm 2.36$ | 7 | 47.72% |
| Cutout | 60% | $9.33 \pm 1.89$ | 5 | 53.59% |
| Add Sticker | 40% | $9.50 \pm 0.50$ | 8 | 84.21% |
| Create Collage | 40% | $16.00 \pm 2.00$ | 12 | 75.00% |
| Add Frame | 20% | $13.00 \pm 0.00$ | 7 | 53.85% |
| **Feishu** | 56% | $7.50 \pm 4.50$ | 4.00 | 46.07% |
| Create Appointment | 40% | $8.00 \pm 1.00$ | 4 | 50.00% |
| Message Contact | 40% | $6.00 \pm 1.00$ | 3 | 50.00% |
| Send file | 20% | $11.00 \pm 0.00$ | 7 | 63.64% |
| Set User Status | 100% | $14.60 \pm 7.50$ | 3 | 20.55% |
| Start Video Conference | 80% | $4.50 \pm 2.60$ | 3 | 46.15% |

**Skill Curation.** In software tasks, direct skill generation was not necessary, as UI operations generally map closely to specific mouse or keyboard actions, making them more straightforward. In contrast, digital game environments involve continuous interactions and decision-making, raising new previously undiscovered information, and requiring the development of new skills to handle novel scenarios and adapt to changing contexts.

However, we do add some additional pre-defined skills, on a per-application basis, for specific knowledge like less-widely known keyboard shortcuts which could be learnt from the application. For example, CapCut's shortcuts screen, shown in Figure 46, or toolbar/icon processing output similarly to the process described for Cities: Skylines. Moreover, we also introduce pre-defined complex skills to demonstrate CRADLE's capability to leverage tools into novel functionality, such as using GPT-4o as a tool to extract information from a video to complete task 5 in CapCut.

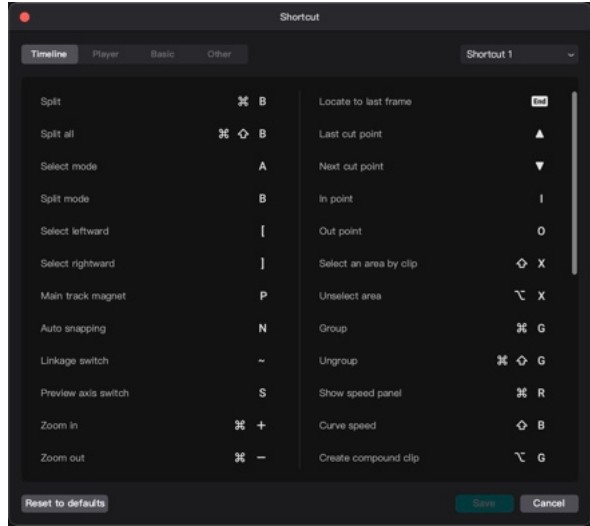

Figure 46: Shortcuts screen in CapCut.

When dealing with shortcuts, *e.g.,* as alternatives to mouse operations, it may be the case that specific shortcuts require "calibration". For example, using the keyboard to navigate the timeline in CapCut (as seen in the bottom area of Figure 43b) requires mapping the keyboard shortcut ('Alt + arrow keys') to pixels or time, which we perform a priori and use the mapping in the pre-defined skill go_to_timestamp(seconds).

**Task Inference.** During the execution of an application task, we let GPT-4o decompose the execution strategy for the next step based on the overall task description and the subtask description. If the previous task decomposition is found to be unreasonable, a new decomposition plan should be proposed and this is evaluated at each iteration round.

**Action Planning.** To enable usage of SAM2SOM, for Action Planning, we insert new mouse skills, which mirror existing coordinates-based mouse skills (*i.e.,* that use x,y coordinates), but take a bounding box numerical label as an argument.

Furthermore, unlike in game playing, which focuses on performing one action per turn, when manipulating software CRADLE can be configured to perform two actions in sequence and thus lower interaction frequency requirements to the backend model. We find that GPT4-o can usually correctly output two-step compound actions. For example, when performing a search in the browser, it can typically output two consecutive action steps, *e.g.,* type_text(text='{user_query}'), followed by the required press_key(key='enter').

**Action Execution.** While atomic and composite skills can involve complex operations, Action Execution happens over the regular CRADLE action space, as shown in Table 6. For example, during Action Execution, a post-processing step converts the bounding box calls into regular mouse actions, using the centroid of a given bounding box as its coordinates for regular mouse operations.

Tool usage, like calling GPT-4o separately to analyze the contents of a media file, is not considered as an action, as tools do not operate on the environment, only as code steps inside a composite skill.

### J.5. Case Studies

#### J.5.1. TASK HARDNESS

It is well known that the difficulty of task completion can vary widely between humans and agents. The results in Table 14 help illustrate some such cases. While many application operation issues may be attributed to UI variety or non-conformity, that is not necessarily the main source of task hardness (*i.e.,* how unexpectedly complex performing an operation is).

Here we use Outlook, a well-known e-mail client, as a case study to discuss how different factors affect CRADLE task completion in real-world application situations (the exact version used is listed in Table 11). Taking task 1 ("Create a new e-mail to {email_address} with the subject 'Hello friend' and send it.") as an example, a success rate of 40% and efficiency of 45.45% may seem lower than expected.

Such a task could be reasonably broken down into steps like: a) Create new e-mail, b) Add recipient, c) Write title, and d) Send e-mail. And the Task Inference module performs such decomposition consistently. However, Action Planning needs to define specific actionable operations with mouse and keyboard to execute each step.

Firstly, CRADLE needs to decide based on the knowledge and visual understanding capabilities available to it to either use a known keyboard shortcut (*e.g.,* 'Ctrl + N') or to click at the "New mail" button. In our experiments, CRADLE tends to chose clicking on the button, which is then affected by the previously discussed issues that led to the integration of SAM2SOM into the framework. Issues in spatial reasoning issues or icon/image understanding may cause a few incorrect click attempts.

Adding the recipient to the e-mail requires typing an address at the appropriate location, *i.e.,* the typical "To" field. This can be accomplished in multiple ways, mainly by typing the address on the UI next to the "To" item or choosing a pre-existing contact.

Clicking on the "To" button triggers the UI to search and select a pre-existing contact e-mail address (with no option of adding a new contact entry, which requires first accessing the "Contacts" menu, outside of "Mail"). Moreover, the UI interaction sequence to select an existing contact can be unintuitive even to experienced users, requiring a minimum of four steps, at each step offering multiple UI options that go away from contact selection. Attempting this flow usually leads CRADLE to exceed the maximum number of allowed step as it gets confused by the UI design.

Nonetheless, choosing the simpler alternative of typing the e-mail address (assuming the correct text field is selected) triggers assistive UI pop-ups (as shown in Figure 48), which lead GPT-4o to falsely conclude the e-mail address is either already typed at the correct location or that it is duplicated and needs to be edited/removed. Furthermore, the pop-ups partially hide the subject area, making it harder for CRADLE to choose the next UI item to interact with for the next task step.

Similar issues with positioning and correctly identifying the typed subject text can also occur, but at a much smaller frequency.

Lastly, completing the task and sending the e-mail requires step similar to creating a new message. But determining send success requires additional attention/reflection as not all cases of the "Send mail" interface disappearing indicate a successful send (*e.g.,* clicking on an unrelated e-mail on the Inbox or closing the current window pop-up).

The Self-Reflection module plays a key role in moving task completion forward by detecting failed attempts at executing each sub-task and providing rationale for failures, even if Information Gathering and Action Planning make repeated mistakes. Such feedback from Self-Reflection and allows Action Planning to tune its process and move ahead.

### J.5.2. TOOL USE IN CAPCUT

Some general computer control tasks may require additional capabilities during execution preparation that can benefit from external tools to enhance agent abilities.

When performing video editing, like in CapCut, a user may need to determine the precise frames to operate on based on video content. For such scenarios, CRADLE can easily leverage tool-using skills, like the LMM's ability to understand actions in a sequence of video frames, enabling it to comprehend video content and identify the exact frames for editing.

We exemplify such tasks with task 5 ("Crop the video when the ball enters the goal, and then export the video") for CapCut, as illustrated in Figure 47. This means our agent can effectively execute tool usage to find the specific frame where "the ball enters the goal". After the first round of Task Inference, CRADLE decomposes the task into three subtasks: 1. Identify the exact frame, 2. Crop the video, and 3. Export the video. Action Planning can then plan to execute 'get_information_from_video(event)' from our curated skills and generate "ball enters the goal" as its required argument for execution.

In this skill, we input a frame set of the video at 1 fps to identify the specific frame where the event occurs. The response is then recorded in Episodic Memory to ensure that subsequent operations can accurately utilize it and target the moment when the action occurs. Across subsequent iterations, CRADLE can then correctly plan and execute the remaining necessary actions for task completion: 'go_to_timestamp(seconds=8)', 'delete_right()', and 'export_project()'].

We have integrated few-shot learning into Self-Reflection to ensure CRADLE recognizes that following export_project(), the expected screen is the CapCut application main window. This information allows it to verify the successful execution of the task, leading to success detection for the overall task.

Figure 47: Showcase of Task 5 ("Crop the video when the ball enters the goal, and then export the new video") in CapCut.

## J.6. Limitations of GPT-4o

Besides the previously discussed limitations of GPT-4o, it is important to highlight a couple other GUI grounding issues.

**Non-standard UI and Noise.**

Non-standard UI, be it in visual style or in behaviour, can lead GPT-4o to misinterpret UI item functionality and application context state. The same applies to visual noise in the form of update pop-up, external contents (*e.g.,* ads), new e-mail/chat messages, etc.

CapCut is affected by both factors, as further illustrated in Figure 49. Moreover, its UI includes non-standard layouts involving precise positioning and drag/dropping. Lack of such prior knowledge by GPT4-o and differences in behaviour between similar functions, may also lead to mistakes in trying to decompose actions to perform. E.g., "Add an effect" requires very different UI-interaction depending on details. Users can add effects in three different ways: i) dragging an effect to the timeline; ii) click the plus sign in a given effect in the effects panel, which adds the effect to the current place on the timeline; and iii) drag an effect directly onto a video and apply the effect to the entire video.

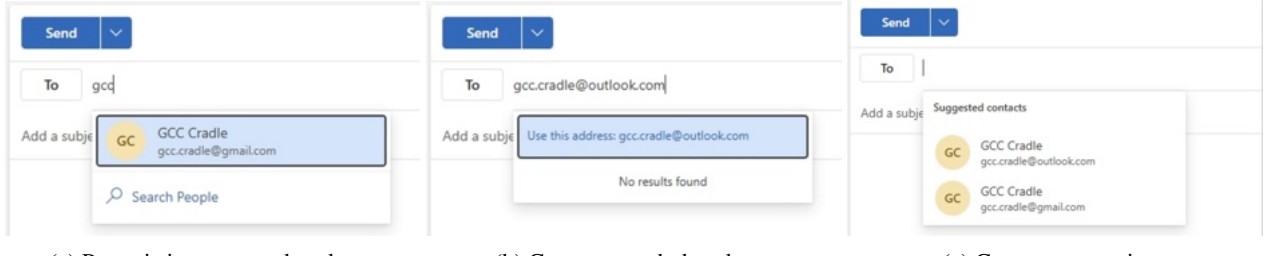

(a) Pre-existing contact dropdown    (b) Contact search dropdown    (c) Contact suggestions

Figure 48: Visual behaviour in Outlook that may lead GPT-4o to visual understanding mistakes.

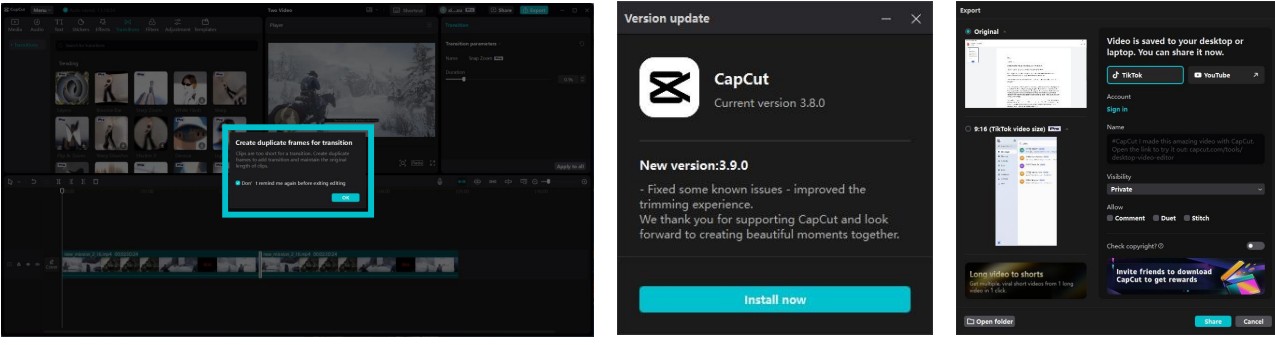

Figure 49: Different *CapCut* pop-ups

**Visual Context Detail.**

GPT-4o still struggles with detailed visual understanding and over-relies on textual information or hallucinations, which results in insufficient attention to visual context and leads to understanding and reasoning mistakes.

One such common example is GPT-4o declaring a dialog state to be ready to press a button like "Save", while ignoring no file name was provided, even if GPT-4o has been prompted to check for such situations. The same applies to it suggesting keyboard shortcuts to open menus that do not exist in the image being interpreted, *e.g.,* trying to press 'Alt + F' to open the "File" menu on a screenshot that has no "File" menu.

Lastly, this lack of attention to context details can also affect understanding the outcome of operations over visual content, leading to incorrect estimation of operation success, *e.g.,* when retouching an image or deciding between a circle and a heart for a shape form.

# K. OSWorld

### K.1. Introduction to OSWorld

OSWorld is a scalable, computer environment designed for multimodal agents. This platform provides a unified environment for assessing open-ended computer tasks involving various applications.

### K.2. OSWorld Tasks

OSWorld is a benchmark suite of 369 real-world computer tasks (mostly on an Ubuntu Linux environment, but including a smaller set on Microsoft Windows) collected from authors and diverse sources such as forums, tutorials, guidelines. Each task is annotated with a natural language instruction and a manually crafted evaluation script for scoring.

### K.3. Implementation Details

The OSWorld environment uses a virtual machine that takes in Python scripts based on PyAutoGUI for actions and provides screenshots and an accessibility tree for observations. We strictly follow the GCC settings. Our agent only uses the screenshot

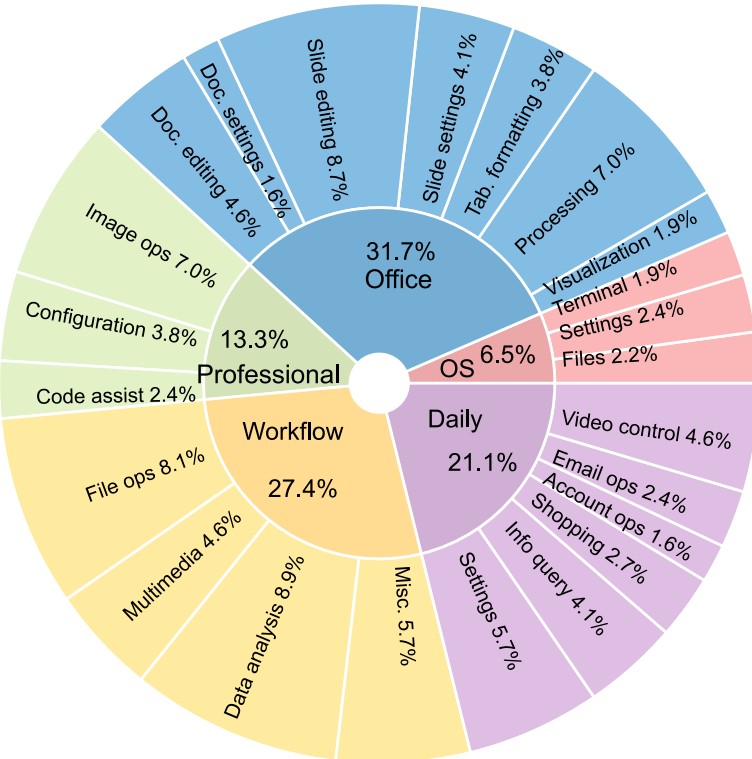

Figure 50: Task instructions distribution in OSWorld (Xie et al., 2024)

as input and outputs Python scripts using PyAutoGUI methods to control the keyboard and mouse (these operations are analogous to the regular action space for **CRADLE**). All 369 tasks use a same set of prompt templates.

We employ GPT-4o as the framework's backbone model. We use the default experimental settings, as in OSWorld's baseline agent. The executable action space is the same as the OSWorld setting, the atomic skills are as follows:

- **Mouse Actions**
    - *move_mouse_to_position(x, y)*: Moves the mouse to a specified position on the screen.
    - *click_at_position(x, y)*: Performs a click at a specified position.
    - *mouse_down(button)*: Presses the specified mouse button.
    - *mouse_up(button)*: Releases the specified mouse button.
    - *right_click(x, y)*: Right-clicks at the specified position.
    - *double_click_at_position(x, y)*: Double-clicks at the specified position.
    - *mouse_drag(x, y)*: Drags the cursor to the position.
    - *scroll(direction, amount)*: Scrolls the mouse wheel up or down by a specified amount.

- **Keyboard Actions**
    - *type_text(text)*: Types the specified text.
    - *press_key(key)*: Presses and releases the specified key.
    - *key_down(key)*: Holds a specified key.
    - *key_up(key)*: Releases a specified key.
    - *press_hotkey(keys)*: Presses a combination of keys and releases them in the opposite order (e.g., Ctrl+C), useful for shortcuts.

- **Task Status**

   – *task_is_not_feasible()*: Indicates that the task cannot be completed, providing feedback for scenarios where the agent encounters infeasible tasks.

Many of these basic skills require GPT-4o to directly output an (x,y) position based on a screenshot. Given that the current GPT-4o is not able to achieve such precise control, we use a grounding tool to augment the screenshot. This way, GPT-4o only needs to choose an object ID. With the object ID and the bounding box of the object, we automatically convert it to the (x,y) position needed for skill execution. Instead of having GPT-4o directly choose the executable skills that require (x,y) position input, we provide several skills that only require a label ID as input for GPT-4o.

- **Actions with Grounding Tools**

  – *click_on_label(label_id)*: Clicks on a specified label in the grounding result.
  – *double_click_on_label(label_id)*: Double-clicks on a specified label in the grounding result.
  – *hover_over_label(label_id)*: Moves the mouse to hover over a specified label in the grounding result.
  – *mouse_drag_to_label(label_id)*: Drags the mouse to a specified label in the grounding result.

**Information Gathering.** Tasks in OSWorld require pixel-level mouse control. While GPT-4 exhibits grounding ability, using tools like SAM can further augment the screenshot with the grounding of icons in complex computer control tasks. The bounding box is helpful for GPT-4 to understand the occurrence of objects on the screen and can also be used to calculate the precise position for mouse control.

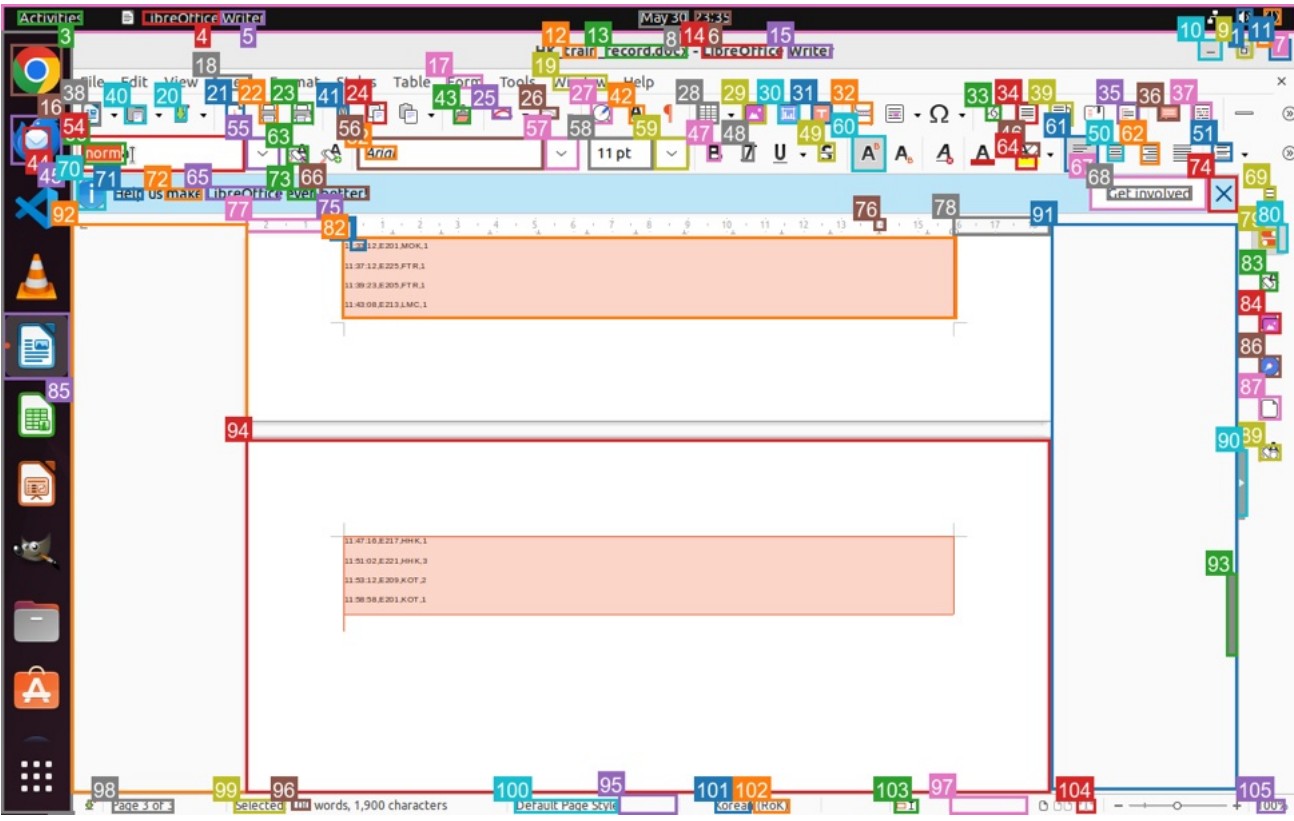

Figure 51: Augmented screenshot using CRADLE's SAM2SOM

**Self-Reflection.** The reflection module evaluates whether previous actions have been successfully executed and determines if the entire task was successful. The self-reflection module is important for tasks in OSWorld, which are sequential decision-making problems that require re-planning based on the current state and previous actions. The self-reflection module also helps to identify infeasible tasks.

### K.4. Application Target and Setting Challenges

Evaluations within OSWorld reveal notable challenges in agents' abilities, particularly in GUI understanding and operational knowledge (Xie et al., 2024). To further complete tasks in OSWorld, the agent needs advanced visual capabilities and robust GUI interaction abilities. Furthermore, the agents face challenges in leveraging lengthy raw observation and action records. The next-level approach encompasses designing more effective agent architectures that augment the agents' abilities to explore autonomously and synthesize their findings.

### K.5. Case Studies

#### K.5.1. INFORMATION GATHERING

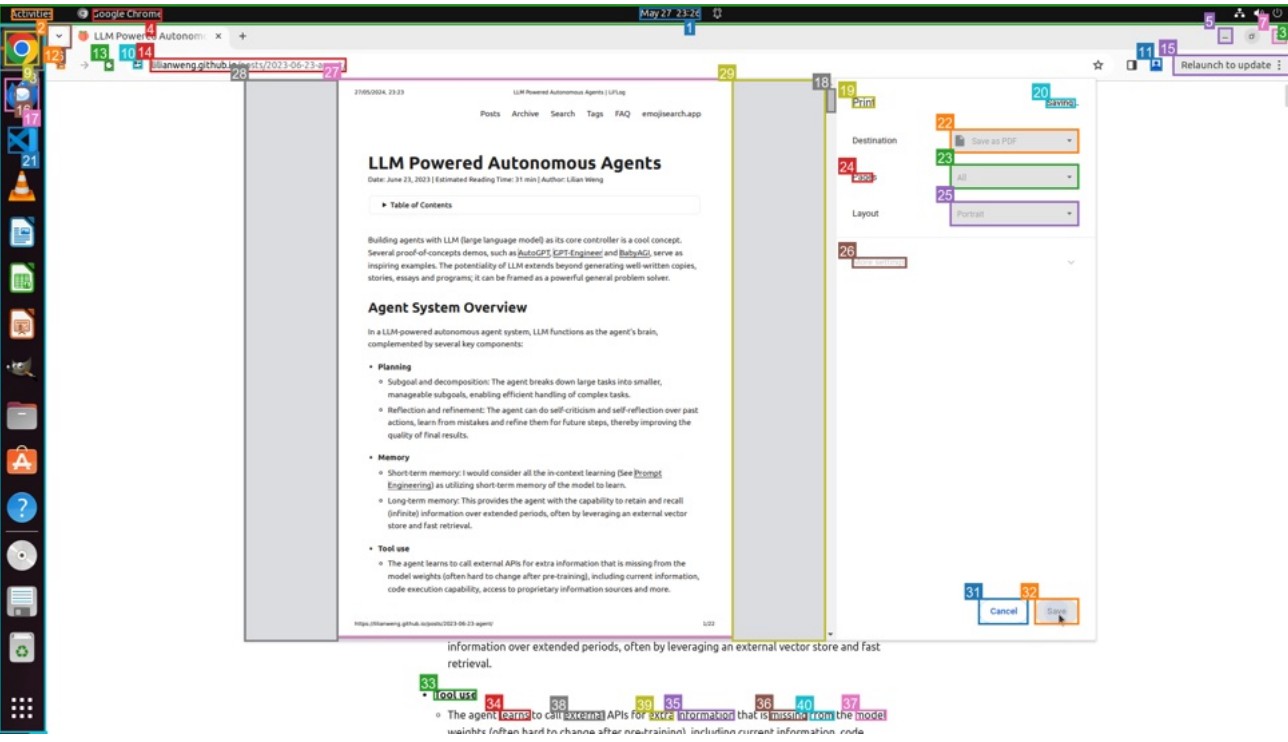

Figure 52: Case Study of robust and precise GUI interaction via information gathering

With SAM as the grounding tool, we prompt the agent to identify the objects in each bounding box to determine the exact position of each object. As shown in Figure 52, the agent recognized the GUI element in box 32 as the Save button. In the planner, the agent chose to click on box 32 to save the PDF, resulting in success.

#### K.5.2. PLANNING WITH SELF-REFLECTION

We showcase how self-reflection combined with planning helps the agent complete a task by coming up with an alternative plan and validating its success.

The current task instruction is "Copy the file 'file1' to each of the directories 'dir1', 'dir2', 'dir3'." As shown in Figure 53, the agent made two attempts at implementing the command but encountered errors and warnings.

As shown in Figure 54, after observing the errors and warnings in the previous steps, the agent checked the files in the directory to debug. After confirming the file structure, the agent tried different commands.

As shown in Figure 55, after executing the new command without receiving an error message, the agent checks whether the files have been copied to the folders. After observing the result, it marks this task as a success.

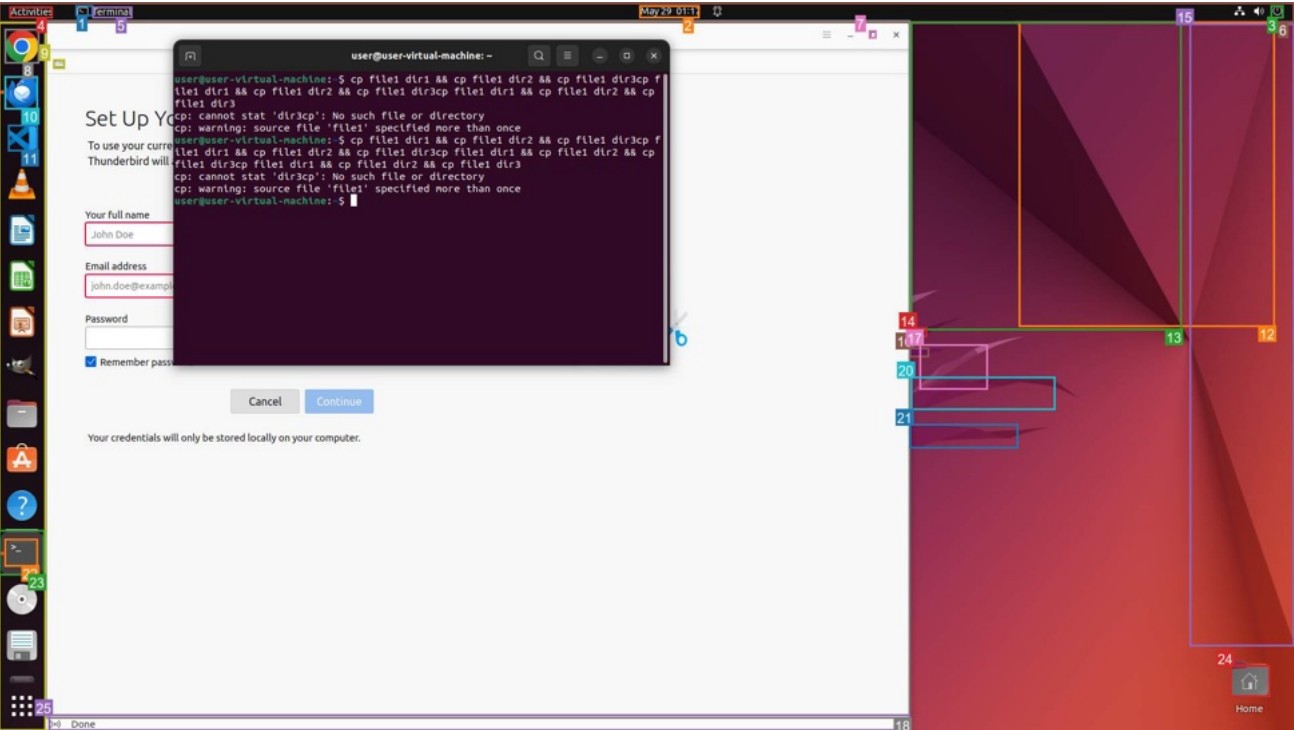

Figure 53: The agent fails to copy the files due to using incorrect commands

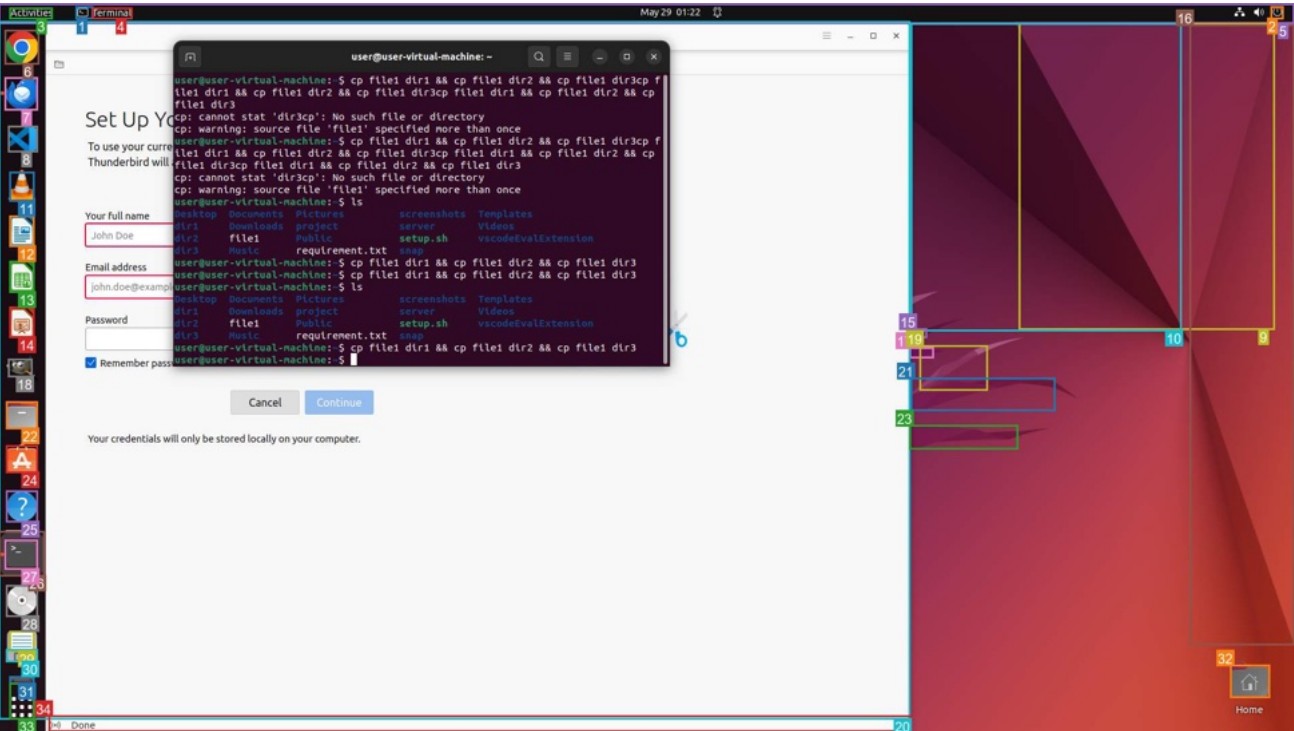

Figure 54: The agent reflects on the errors, checks the file structure and tries to debug

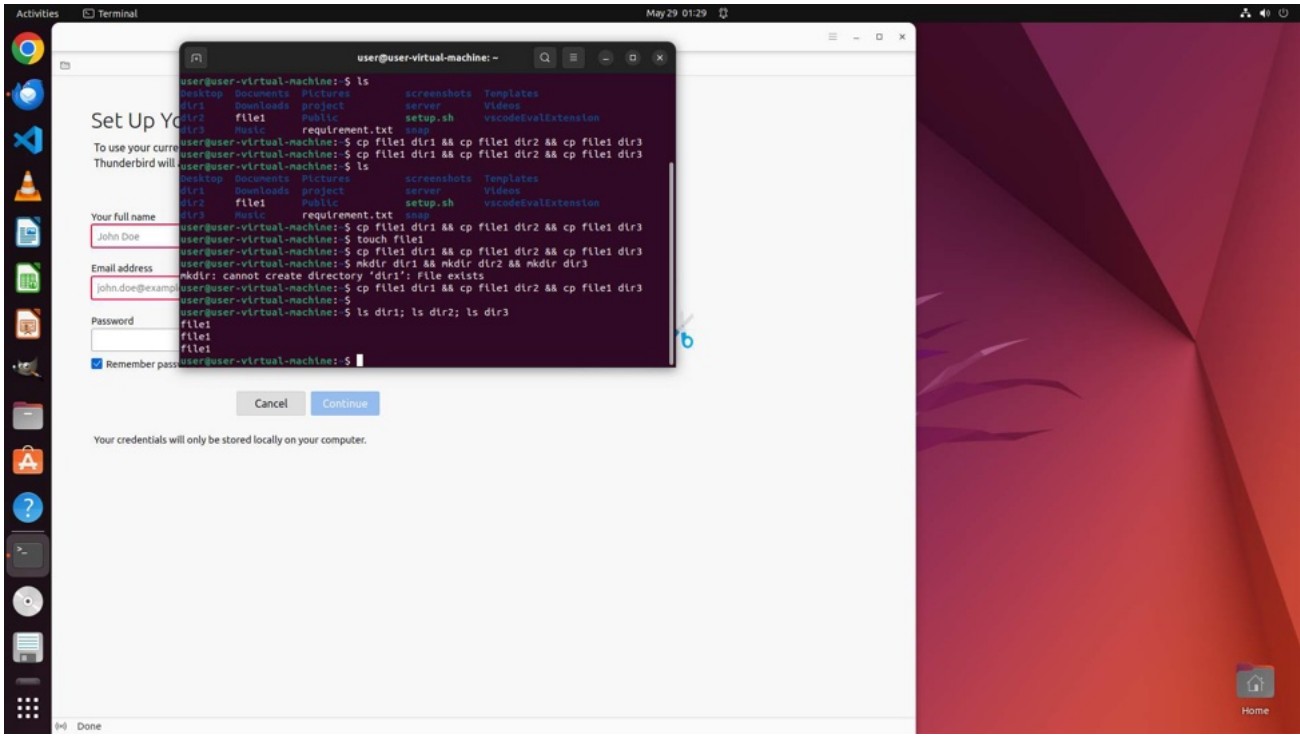

Figure 55: The agent checks if the files have already been copied

Table 15: Detailed success rates divided by domains: OS, LibreOffice Calc, LibreOffice Impress, LibreOffice Writer, Chrome, VLC Player, Thunderbird, VS Code, GIMP, and Workflow (*i.e.,* involves multiple applications).

| Method | OS (24) | Calc (47) | Impress (47) | Writer (23) | VLC (17) | TB (15) | Chrome (46) | VSC (23) | GIMP (26) | Workflow (101) |
|---|---|---|---|---|---|---|---|---|---|---|
| GPT-4o | 8.33 | 0.00 | 6.77 | 4.35 | 16.10 | 0.00 | 4.35 | 4.35 | 3.85 | 5.58 |
| GPT-4o+SoM | 20.83 | 0.00 | 6.77 | 4.35 | 6.53 | 0.00 | 4.35 | 4.35 | 0.00 | 3.60 |
| **CRADLE** | 16.67 | 0.00 | 4.65 | 8.70 | 6.53 | 0.00 | 8.70 | 0.00 | 38.46 | 5.48 |

## K.6. Quantitative Evaluation

The detailed success rates for each application are listed in Table 15. We followed the same experimental settings as the OSWorld paper, running the experiment only once. Our results show that our agent performs better in the Chrome and GIMP domains. However, the difference in performance in the OS, Writer, and VSC domains is less statistically significant due to the smaller number of tasks. While improved information gathering and self-reflection empowered the agent in these domains, the complex pipeline and limitations of current grounding tools and GPT-4 hindered performance in domains like VLC and VSC. We identify these limitations as future directions for implementing the agent in real-world scenarios.

