# OpenReview forum: "Cradle: Empowering Foundation Agents towards General Computer Control"
_ICML.cc/2025/Conference — ICML 2025 poster_

### Official Review · Reviewer_Wqig · 2025-03-14

**Overall Recommendation:** 2

**Summary:**

The paper presents CRADLE, framework that leverages LMMs designed for General Computer Control (GCC). CRADLE operates directly through visual observations (screenshots) and generates keyboard  and mouse commands, enabling it to interact with diverse software environments without relying on specialized APIs. Its architecture comprises six core modules: 1. Information Gathering; 2. Self-Reflection; 3. Task Inference; 4. Skill Curation; 5. Action Planning, and 6. Memory. These facilitate effective interaction, learning, and adaptability.

Experimentally, CRADLE states that it demonstrated notable generalization and strong performance across challenging tasks in 4 complex commercial video games (including RDR2 and Stardew Valley) and 5 real world software applications (Chrome, Outlook, Feishu, Meitu and CapCut).

Key findings include CRADLE’s ability to complete extended missions in an AAA game environment, generate sophisticated procedural skills, and achieve performance comparable to or surpassing human players in several tasks, thus validating the proposed GCC setting. The authors mention being the first to evaluate and showcase a framework that can interacti with both commercial complex games and software applications.

**Claims And Evidence:**

Work clearly demonstrates feasibility and promising generalization in complex commercial video games & sofwatware application, including initial successes with extended storyline tasks (e.g., RDR2).

However, the claims made by the authors regarding CRADLE's strong generalization and high performance across previously unexplored complex commercial video games and software applications are only partially supported. While results in games like RDR2 and Cities: Skylines are convincing, the performance in common software applications (e.g., Chrome, Outlook, CapCut) is less impressive and insufficiently analyzed quantitatively. Additionally, comparisons are limited to inexperienced human players, omitting valuable amateur-level comparisons, which could better contextualize the agent's effectiveness, especially given possible prior LLM knowledge about these games. Most importantly, evaluation results on established software benchmarks (e.g., OSWorld) are absent from the main paper, weakening the generalization claim (are only present in appendix). Finally, relevant comparisons to stronger existing software use frameworks are missing, e.g. AGUVIS - Xu et. al. 2024.

**Essential References Not Discussed:**

Not specifically.

**Experimental Designs Or Analyses:**

The experiments and applications are generally appropriate for the proposed GCC framework chosen.

Experimental design issues:

- comparison baseline was conducted by inexperienced human players; this limits insights into the agent's relative performance, especially since the used LMM  (GPT-4o) might have prior knowledge of certain games.
- the evaluation of software tasks laks detailed quantitative analyses, such as ablation studies or stronger baseline comparisons on more common benchmarks. These omissions make it hard to fully assess CRADLE’s claimed superiority and generalization capabilities.

**Methods And Evaluation Criteria:**

Yes, the methods and evaluation criteria proposed in the paper generally align well with their stated goal of addressing the General Computer Control (GCC) setting. However, some evaluation choices limit the comprehensiveness of their claims. Notably, Minecraft, a well-established benchmark for agent generalization and lifelong learning would have been a valuable addition but is missing. Additionally, the 5 software applications they chose to analyse are not common for computer use benchmarks, and neither are they properly introduced or evaluated. Nonetheless, the chosen environments and methods generally align well with their proposed GCC setting.

**Other Comments Or Suggestions:**

No

**Other Strengths And Weaknesses:**

**Strengths:**

- Effectively combines multimodal inputs, reflection, skill generation, and memory into a modular and coherent framework.
- Demonstrates practical significance and general-purpose computer-use capability through challenging commercial video games and everyday software tasks, practical significance.

**Weaknesses:**

- Limited quantitative evaluation, particularly in common software tasks, weakens the generalization claims.
- Insufficient comparison against strong, existing benchmarks (e.g., OSWorld, AGUVIS, Voyager) diminishes the strength of the results.
- Evaluations conducted only against inexperienced humans neglect expert-level comparisons, potentially inflating agent effectiveness.
- Excessive focus on video games over common software tasks limits broader applicability insights.
- Limited algorithmic novelty

**Questions For Authors:**

1. Why did you choose to evaluate CRADLE against only human players who had never played the corresponding games before?

**Relation To Broader Scientific Literature:**

The paper builds  extends recent advances in foundation agents, multimodal learning, and embodied AI. It positions CRADLE within the broader context of LMM-powered agents like Voyager (Wang et al. 2023), which demonstrated lifelong in-context learning in Minecraft, and other multimodal web agents such as WebVoyager (He et al., 2024) and Mind2Web (Deng et al., 2023). Unlike these prior works, which typically rely on domain-specific APIs or simpler interaction spaces, CRADLE introduces the General Computer Control (GCC) setting, utilizing general-purpose screenshots and executable mouse and keyboard actions without API dependence. This paper also relates closely to recent LMM-based agents designed for GUI interaction tasks (ScreenAgent, Niu et al., 2024; Voyager, Wang et al., 2023) but aims for broader generalization across both gaming environments and practical software. Given the limited algorithmic novelty, maybe the paper could be adapted and focus on strictly proposing a general purpose benchmark.

**Theoretical Claims:**

The paper does not contain any theoretical claims.

---

> ### Author Rebuttal · Authors · 2025-03-31
>
> We sincerely appreciate the reviewers for their valuable feedback and insightful comments. We hope our following answers will clear up the doubts about our work, and please let us know if there is any other clarification we can provide.
>
> ---
> **Q1**: About the selection of the video game and software application for evaluation.
>
> **A1**: We would like to clarify that Cradle primarily focuses on demonstrating the effectiveness of the GCC setting, enabling agents to interact with software in a unified manner. Thus, we deliberately selected four representative games that do not provide API access, so that they have not been explored in prior studies, to clearly distinguish Cradle from existing approaches. As the reviewer rightly pointed out, Minecraft is already a well-established benchmark with rich API access. Many agents have demonstrated impressive performance on it. Evaluating Cradle on Minecraft would not further strengthen our main claims. Benchmarking against specialized domain-specific agents or models is not the primary objective of our current study.
>
> On the other side, the selection of software applications tells the same story. We deliberately selected several challenging productivity software (e.g., Feishu, Meitu and Cupcut) that are seldom explored before. Compared to video games, there are already many previous agentic works on software and software tasks are usually closer to daily life and easy to understand. Due to the limitation of pages, it is regretful that we have to put the introduction of these software and tasks, and the quantitative results on OSWorld in the appendix. We will move them back to the main paper in the camera-ready version as one more page will be provided.
>
> ---
> **Q2**: Insufficient comparison against strong, existing benchmarks (e.g., OSWorld, AGUVIS, Voyager) diminishes the strength of the results.
>
> **A2**: We would like to kindly remind the reviewers that both OSWorld and Voyager are evaluated in our paper. As for AGUVIS, what they proposed is a trained VLM model instead of an agentic framework. The work is released within two months before the deadline for ICML 2025. According to the ICML policy (https://icml.cc/Conferences/2025/ReviewerInstructions), “Authors cannot expect to discuss other papers that have only been made publicly available within four months of the submission deadline. Such recent papers should be considered as concurrent and simultaneous. ” Nevertheless, we agree on the importance of these works and will include additional discussion in the related work section.
>
> ---
> **Q3**: Evaluations conducted only against inexperienced humans neglect expert-level comparisons, potentially inflating agent effectiveness.
>
> **A3**: We deeply understand the reviewers’ concerns and provide supplementary comparisons with expert-level players in the following link: https://drive.google.com/file/d/1NUgtjCFhrV3B8RdCvw65LMr5NJX8pJm7/view?usp=sharing.
>
> There indeed still a gap between Cradle and expert players, however we argue that comparing these agents to expert-level humans might be inherently unfair. Although LMMs have some high-level gameplay knowledge from large-scale internet pretraining, they lack explicit training on the low-level actions needed for specific game tasks evaluated in our study. Thus, LMMs are closer to novice players who gather some information from the internet or game wikis but have no practical experience. Inexperienced players are introduced to basic gameplay and provided the same prompts used by agents to ensure fairness.
>
> Moreover, Cradle is designed to emulate the experience of a fresh player by progressively acquiring new skills and capabilities as gameplay unfolds. At the start of the game, Cradle possesses only a limited set of atomic control skills. In contrast, expert human players already have complete mastery over all necessary skills from the beginning, providing them with a significant initial advantage. Thus, we believe the comparisons made in our study accurately reflect the agent’s true learning capabilities in relation to novice human players.
>
> ---
> **Q4**: About algorithmic novelty
>
> **A4**: We deeply understand reviewers’ high criteria for improving our work to be better. We kindly remind that the selected Primary Area of this submission is Applications. Cradle focues on showing that the challenging GCC setting can be properly handled by current techniques, thus motivating more researchers and developers engaged in this setting. Limited algorithmic novelty is unnecessary to be a weakness of our submission.

---

### Official Review · Reviewer_GCyf · 2025-03-14

**Overall Recommendation:** 3

**Summary:**

This paper focuses on building a framework based on a multimodal model, specifically OpenAI’s GPT-4o, for computer use through keyboard and mouse inputs. The proposed framework consists of six distinct modules: information gathering, self-reflection, task inference, skill curation, action planning, and memory, which are employed in a prompting and agentic manner to interact with games and software applications.

The authors evaluate their proposed framework on four different games and common software applications. The experimental results in gaming show that, except for Stardew Valley, the framework achieves high completion rates in three of the tested games. However, in the case of general software applications, while the framework demonstrates the ability to accomplish specific tasks, it does not exhibit a consistently high level of performance across different applications.

### update after rebuttal:
I thank the authors for their response and have taken into account the perspectives of the other reviewers. My concerns have been partially addressed, and I have accordingly raised my score.

**Claims And Evidence:**

I believe the paper’s main claim, which asserts that the proposed framework (CRADLE) can effectively operate a computer in complex environments and exhibits strong generalization capabilities, is not sufficiently supported by evidence.

The validation of the framework primarily relies on experiments in four games, where it demonstrates promising results in three of them. While this provides some support for the claim, I am concerned that the scale of the evaluation is too small to convincingly establish the framework’s generalization ability. Moreover, the framework does not demonstrate strong performance in general software applications, which further weakens the generalization claim.

Currently, OSWorld is the most widely recognized benchmark for evaluating computer-use agents. The paper reports that the proposed framework achieves a score of 7.81, which is significantly lower than the state-of-the-art models on this benchmark. Even compared to the much smaller UI-TARS 7B model (which scores 18.7), the proposed framework falls short.

**Essential References Not Discussed:**

The paper adequately covers the key related works.

**Experimental Designs Or Analyses:**

Yes, I reviewed the experimental design and analysis used to evaluate the proposed framework.

As mentioned earlier, one major issue with the experiment is the limited scale of evaluation, particularly in the gaming domain.

Another experimental design issue is the lack of strong baseline comparisons. The authors state in Section 4.2 that there are no existing models capable of performing computer operations like their proposed framework. However, several computer-use agents have already been evaluated on the OSWorld benchmark, including Claude and UI-TARS. These models have publicly reported performance scores on OSWorld and could have been used for comparison. Furthermore, these existing agents could have been adapted for gameplay evaluations, allowing for a more rigorous comparison.

**Methods And Evaluation Criteria:**

I find the proposed framework reasonable, but a major shortcoming is the lack of clarity in its description.

In Section 3, the authors provide a high-level overview of each module in the framework. However, they do not clearly explain how these modules interact or provide sufficient details on the end-to-end process of the framework. Even though the authors mention in the appendix that there was limited space to include more details, I believe this does not justify the absence of a structured description of the full framework—such as presenting its workflow in an algorithmic format.

The lack of detailed algorithmic descriptions makes it difficult to assess the novelty and effectiveness of the proposed approach. Additionally, many components of the framework, such as episodic memory and self-reflection, have already been widely explored in recent agentic frameworks, further raising concerns about its level of innovation.

**Other Comments Or Suggestions:**

The authors should consider revising Section A.5 in the appendix. It appears that this section contains responses from a previous submission, as seen in line 764, where the text states:"The work mentioned by the reviewer…"

**Other Strengths And Weaknesses:**

As mentioned earlier, many of the modules used in the proposed framework—such as self-reflection—have already been widely implemented in existing agentic frameworks. However, the paper does not provide a clear explanation of its algorithmic contributions or highlight its specific innovations. Based on the current description, the proposed framework appears to be a combination of existing techniques.

**Questions For Authors:**

Why not compare the proposed framework with existing models that have been evaluated on OSWorld?

**Relation To Broader Scientific Literature:**

The paper’s focus on GUI-based computer-use agents for gaming applications is novel, as most existing computer-use agents are primarily designed for browser-based tasks or basic software operations.

**Theoretical Claims:**

The paper does not make any theoretical claims.

---

> ### Author Rebuttal · Authors · 2025-03-31
>
> We sincerely appreciate the reviewers for their valuable feedback and insightful comments. We hope our following answers will clear up the doubts about our work, and please let us know if there is any other clarification we can provide.
>
> ---
>
> **Q1**:  The scale of evaluation in the gaming domain is limited.
>
> **A1**: We appreciate the reviewer’s concern about the scale of evaluation. However, the selected four representative games already cover a broad spectrum of game types and gameplay, from 2D to 3D, from RPG to simulation game, from comic style to realistic style, from first-person/third-person perspective to top-down perspective, etc. To our best knowledge, no previous works were evaluated on video games with so many diverse game types and gameplay before.  Additionally, based on GCC setting, Cradle does not rely on any assumptions of playing these games, which is sufficient to show that Cradle also has the potential to be developed to thousands of games with the same types and even different types. We would like also to note that one of our key related work and baseline methods, Voyager, is only evaluated in one game.
>
> Finally, we also provide some preliminary results of applying Cradle to the extremely challenging Action RPG game, Black Myth: Wukong. Cradle still manages to defeat boss and enemies in the early stages of the game.
>
> |              Task             |   Cradle  |
> |:-----------------------------:|:---------:|
> |       Defeat Erlang           |    100%    |
> |       Defeat WolfScout      |    30%    |
> |       Defeat WolfSwornScout      |    20%    |
> |       Defeat Croaky       |    40%    |
>
> ---
>
> **Q2**: The framework does not demonstrate strong performance in general software applications, which is significantly lower than the SOTA model like UI-TARS and Claude. These baselines are not compared in the paper.
>
> **A2**: We deeply understand that compared to the very recent SOTA models, the performance reported by Cradle in OSWorld is not impressive enough.  However, we would like to kindly remind that all the models mentioned by the reviewers emerged very close to the ICML 2025 submission deadline.  UI-TARS is even after the ICML abstract deadline. According to the ICML policy (https://icml.cc/Conferences/2025/ReviewerInstructions), “Authors cannot expect to discuss other papers that have only been made publicly available within four months of the submission deadline. Such recent papers should be considered as concurrent and simultaneous. ” Nevertheless, we agree on the importance of these works and will include additional discussion in the related work section.
>
> Importantly, we would like to clarify that our primary claim is that Cradle represents the first framework capable of achieving strong performance across both video games and software applications. The domain-specific model, UI-TARS, which primarily demonstrates effectiveness only in software tasks, does not undermine Cradle’s broader contribution. Furthermore, UI-TARS benefits from direct training on data collected from OSWorld, naturally resulting in good performance in this specific benchmark. It still struggles with less common software applications like Cupcut, Feishu and Meitu, further validating Cradle's generalization across diverse software tasks.
> Additionally, comparing models directly with Cradle may not be entirely appropriate. Cradle is a framework instead of a model, which needs to be initialized with a base model. Compared to UI-TARS, the base model used by Cradle (gpt-4o-0513) inherently exhibits much weaker performance on GUI tasks. Cradle's capability to significantly enhance performance relative to its base model clearly illustrates its strong generalizability and adaptability. As the base model improves, the performance of Cradle will also improve. Therefore, the improvement of general-purpose models like GPT and Claude does not have a direct competition relationship with Cradle, but a mutually reinforcing relationship instead.
>
> ---
>
> **Q3**: Lack of structured description of the full framework such as presenting its workflow in an algorithmic format.
>
> **A3**: We would like to clarify that Cradle is a flexible framework, which can be customized to different tasks and environments. The main workflow has been illustrated in Figure 3. To further address the reviewers’ concerns, we provide a pseudocode to show the workflow in the following anonymous link: https://drive.google.com/file/d/15r4lveEyGaMEFDhOfZrJ8PT1DXhC6M4i/view .
>
> ---
>
> We also thank the reviewers for catching our minor oversight in the appendix, which will be fixed in the latest version.

---

### Official Review · Reviewer_Cfph · 2025-03-15

**Overall Recommendation:** 3

**Summary:**

The paper proposes the General Computer Control (GCC) setting where the input is restricted to screenshots and the output to keyboard and mouse actions. To address this setting, the paper proposes Cradle, an LMM-based framework with six components:  Information
Gathering, Self-Reflection, Task Inference, Skill Curation, Action Planning, and Memory. Given a screenshot as input, Information Gathering parses visual and textual information using LMM. Self-Refelction then reasons about what happened based on the extracted information. Task Inference plans a next task from the reflected result to achieve a desired goal. Given the predicted task, Skill Curation generates necessary skills to complete it and Action Planning retrieves relevant skills to take the next action for the goal. The proposed approach is validated by four video games, five software applications, and the OSWorld benchmark with noticeable margins over the baslines.

**Claims And Evidence:**

It seems the claims made in the submission are supported by convincing evidence.

**Essential References Not Discussed:**

It seems essential references are cited in this paper.

**Experimental Designs Or Analyses:**

- For Table 2, Section 4.2 describes the baselines as models using a subset of the six components used in Cradle. It is unclear whether we can say they are indeed prior work. And why not just providing an ablation study of each component instead?

**Methods And Evaluation Criteria:**

- Why do we need these six steps? Can this be reduced to fewer steps? The necessity of each step seems not well justified.
- A naive approach to the GCC setting is to directly ask LMM to output the next step. Why not directly ask LMM to generate the next action?
- The proposed multi-staged approach can be easily affected by even a single failure in intermediate steps. Can the proposed approach address this? This may be particularly important for some tasks, which are often irrecoverable, such as bank accounts, privacy issues, etc.

**Other Comments Or Suggestions:**

I have no other comments.

**Other Strengths And Weaknesses:**

Strengths
- The paper is generally written well and easy to follow.
- The proposed GCC setting sounds reasonable and well-motivated. Addressing the general I/O framework seems important and necessary.
- The paper provides extensive analyses on its method.

Weaknesses
- While agreeing to address the GCC setting, it would be better to see how much Cradle can be improved if it has access to APIs and their documentation for a target program.

**Questions For Authors:**

All questions are made in the sections above.

**Relation To Broader Scientific Literature:**

The GCC setting addresses the general format of I/O for computer control. I believe that addressing this setting can be extended to many other domains such as robotics that usually require different I/O formats for different robot bodies.

**Theoretical Claims:**

No theoretical claims are made.

---

> ### Author Rebuttal · Authors · 2025-03-31
>
> We sincerely appreciate the reviewers for their valuable feedback and insightful comments. We hope our following answers will clear up the doubts about our work, and please let us know if there is any other clarification we can provide.
>
> ---
>
> **Q1**: About the necessity of each module of Cradle. Why not provide an ablation study?
>
> **A1**: Thanks for pointing it out. We indeed provide a comprehensive ablation study by systematically removing each module of Cradle to show the effectiveness in Appendix E3 and Table 5 (page 18-19). The ablation study shows that removing any of the modules will result in a significant performance loss. Due to the limitation of pages, we have to put the result in the appendix. As the camera-ready version will allow one more page, we will move the ablation study to the main paper for better readability.
>
> ---
>
> **Q2**: Why not directly ask LMM to generate the next action? The proposed multi-staged approach can be easily affected by even a single failure in intermediate steps.
>
> **A2**: One of our baselines, ReAct, is exactly kind of a one-stage approach. It lets LMM generate the next action with CoT in one step. This method shows much worse performance than Cradle. Additionally, according to our ablation study, removing any of the modules will result in a significant performance loss. From the perspective of designing complex systems [1], systems with no redundancy or error-detection mechanisms are susceptible to single-point failures. A module dedicated to a single function is less likely to cause serious system-wide issues than one that tries to handle everything. For the multi-staged approach, the error still has the potential to be corrected by the following stages, however, the error in the single-staged approach will directly be executed into the environment and cause unrecoverable loss.
>
> [1] Blanchard, Benjamin S., Wolter J. Fabrycky, and Walter J. Fabrycky. Systems engineering and analysis. Vol. 4. Englewood Cliffs, NJ: Prentice hall, 1990.
>
> ---
>
> **Q3**: While agreeing to address the GCC setting, it would be better to see how much Cradle can be improved if it has access to APIs and their documentation for a target program.
>
> **A3**: We thank the reviewers for acknowledging our GCC setting. We want to clarify that Cradle is designed to solve the challenges that GCC presents. If provided with APIs, the agent can directly have access to the internal state and the full action space with the meaning of each action. Modules like information gathering and skill curation are less effective. The agent can even complete tasks with textual observation without visual ability. One of our baselines, Voyager, is a good example to show the performance of a multi-staged agent with API access. Since most of the software applications and video games do not provide API access, this kind of method has limited application scenarios. The setting with API and documentation will largely violate the GCC setting. We expect that the APIs and documentation access can improve the performance of Cradle, but this is beyond the scope of this paper.

---

> > ### Comment · Reviewer_Cfph · 2025-04-04
> >
> > Thank you for your response. The answer addressed my concerns and therefore I'd like to keep my accept rating for now. I believe this work still has some value but I'll make the final rating based on the other reviews as well.

---

### Decision · Program_Chairs · 2025-05-01

**Decision:**

Accept (poster)

**Comment:**

This paper focuses on the interesting and ambitious application of general computer control, and demonstrated solid experimental results across a variety of games and software applications that these challenging general computer control settings can be properly handled by current VLMs. Concerns about comparison with other methods and concurrent work were addressed by the authors in the rebuttal with no additional follow ups from the reviewer. Despite the work is largely application focused, discussing any algorithmic insights with some additional algorithmic innovations can improve the quality of the work. Overall, the work makes meaningful contributions. Please be sure to open source the framework upon acceptance of the work.